# Shortcuts and Identifiability in Concept-based Models from a Neuro-Symbolic Lens

**Samuele Bortolotti**[*]
DISI, University of Trento
Italy
samuele.bortolotti@unitn.it

**Emanuele Marconato**[*]
DISI, University of Trento
Italy
emanuele.marconato@unitn.it

**Paolo Morettin**
DISI, University of Trento
Italy
paolo.morettin@unitn.it

**Andrea Passerini**
DISI, University of Trento
Italy
andrea.passerini@unitn.it

**Stefano Teso**
CIMeC and DISI, University of Trento
Italy
stefano.teso@unitn.it

## Abstract

Concept-based Models are neural networks that learn a concept extractor to map inputs to high-level ***concepts*** and an ***inference layer*** to translate these into predictions. Ensuring these modules produce interpretable concepts and behave reliably in out-of-distribution is crucial, yet the conditions for achieving this remain unclear. We study this problem by establishing a novel connection between Concept-based Models and ***reasoning shortcuts*** (RSs), a common issue where models achieve high accuracy by learning low-quality concepts, even when the inference layer is *fixed* and provided upfront. Specifically, we extend RSs to the more complex setting of Concept-based Models and derive theoretical conditions for identifying both the concepts and the inference layer. Our empirical results highlight the impact of RSs and show that existing methods, even combined with multiple natural mitigation strategies, often fail to meet these conditions in practice.

## 1 Introduction

Concept-based Models (CBMs) are a broad class of self-explainable classifiers [1, 2, 3, 4, 5, 6, 7] designed for high performance and *ante-hoc* interpretability. Learning a CBM involves solving two conjoint problems: acquiring high-level ***concepts*** describing the input (e.g., an image) and an ***inference layer*** that predicts a label from them. In many applications, it is essential that these two elements are "high quality", in the sense that: *i*) the concepts should be ***interpretable***, as failure in doing so compromises understanding [8, 9] and steerability [10, 11], both key selling points of CBMs; and *ii*) the concepts and inference layer should behave well also ***out of distribution*** (OOD), e.g., they should not pick up spurious correlations between the input, the concepts and the output [12, 13, 14].

This raises the question of when CBMs can acquire "high-quality" concepts and inference layers. While existing studies focus on concept quality [15, 16, 17], they neglect the role of the inference

---

[*]Equal contribution. Correspondence to samuele.bortolotti@unitn.it

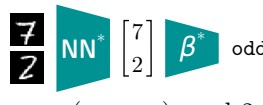 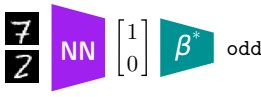 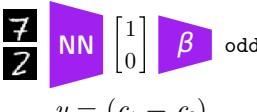

$$y = (c_1 + c_2) \bmod 2 \qquad y = (c_1 + c_2) \bmod 2 \qquad y = (c_1 - c_2)$$

Figure 1: **Joint reasoning shortcuts**. The goal is to predict whether the sum of two MNIST digits is odd (as in Example 2.1) from a training set of all possible unique (even, even), (odd, odd), and (odd, even) pairs of MNIST digits. **Green** elements are fixed, **purple** ones are learned. **Left**: ground-truth concepts and inference layer. **Middle**: NeSy-CBMs with given knowledge can learn reasoning shortcuts, i.e., concepts with unintended semantics. **Right**: CBMs can learn joint reasoning shortcuts, i.e., both concepts and inference layer have unintended semantics.

layer altogether. In contrast, we cast the question in terms of whether it is possible to ***identify*** from data concepts and inference layers with the intended semantics, defined formally in Section 3.1. We proceed to answer this question by building a novel connection with ***reasoning shortcuts*** (RSs), a well-known issue in Neuro-Symbolic (NeSy) AI whereby models achieve high accuracy by learning low-quality concepts ***even if the inference layer is fixed*** [18, 19, 20, 21]. For instance, a NeSy model for autonomous driving whose inference layer encodes the traffic laws might confuse pedestrians with red lights, as both entail the same prediction (the car has to stop) [22]. We generalize RSs to CBMs in which ***the inference layer is learned***, and ***concept supervision may be absent***. Our analysis shows that shortcuts can be exponentially many, even more than RSs, (we count them explicitly in Appendix B.5) and that maximum likelihood training is insufficient for attaining intended semantics. This hinders both interpretability and OOD behavior. On the positive side, we also specify conditions under which (under suitable assumptions) CBMs *cannot* be fooled by RSs, proving that ***the ground-truth concepts and inference layer can be identified*** (see Theorem 3.9).

Our evaluation on several learning tasks suggest that CBMs can be severely impacted by reasoning shortcuts in practice, as expected, and also that the benefits of popular mitigation strategies do not carry over to this more challenging problem. These results cast doubts on the ability of these models to identify concepts and inference layers with the intended semantics unless appropriately nudged.

**Contributions**: In summary, we: (i) generalize reasoning shortcuts to the challenging case of CBMs whose inference layer is learned (Section 3.1), (ii) study conditions under which maximum likelihood training can identify good concepts and inference layers (Section 3.2), and (iii) present empirical evidence that well-tested mitigations fail in this challenging setting (Section 5).

## 2 Preliminaries

**Concept-based Models** (CBMs) first map the input $\mathbf{x} \in \mathbb{R}^n$ into $k$ discrete categorical concepts $\mathbf{c} = (c_1, \ldots, c_k) \in \mathcal{C}$ via a neural backbone, and then infer labels $\mathbf{y} \in \mathcal{Y}$ from this using a white-box layer, e.g., a linear layer. This setup makes it easy to figure out what concepts are most responsible for any prediction, yielding a form of *ante-hoc* concept-based interpretability. Several architectures follow this recipe, including approaches for converting black-box neural networks into CBMs [23, 24, 25, 26].

A key issue is how to ensure the concepts are interpretable. Some CBMs rely on ***concept annotations*** [3, 27, 4, 5, 28, 29]. These are however expensive to obtain, prompting researchers to replace them with (potentially unreliable [30, 31]) annotations obtained from foundation models [32, 33, 34, 35] or ***unsupervised*** concept discovery [1, 2, 6, 36].

**Neuro-Symbolic CBMs** (NeSy-CBMs) specialize CBMs to tasks in which the prediction $\mathbf{y} \in \mathcal{Y}$ ought to comply with known safety or structural constraints. These are supplied as a formal specification – a logic formula K, *aka **knowledge*** – tying together the prediction $\mathbf{y}$ and the concepts $\mathbf{c}$. In NeSy-CBMs, the inference step is a ***symbolic reasoning layer*** that steers [37, 38, 39] or guarantees [40, 41, 42, 43, 44] the labels and concepts to be logically consistent according to K. Throughout, we will consider this example task:

**Example 2.1** (MNIST-SumParity). *Given two MNIST digits [45], we wish to predict whether their sum is even or odd. The numerical values of the two digits can be modelled as concepts* $\mathbf{C} \in \{0, \ldots, 9\}^2$*, and the inference layer is entirely determined by the prior knowledge:* K = ((y =

$1) \Leftrightarrow (C_1 + C_2) \text{ is odd}$). *This specifies that the label* $y \in \{0, 1\}$ *ought to be consistent with the predicted concepts* **C**. *See Fig. 1 for an illustration.*

Like CBMs, NeSy-CBMs are usually trained via ***maximum likelihood*** and gradient descent, but ***without concept supervision***. The reasoning layer is typically imbued with fuzzy [46] or probabilistic [47] logic semantics to ensure differentiability. Many NeSy-CBMs require K to be provided upfront, as in Example 2.1, hence their inference layer has no learnable parameters. Starting with Section 3, we will instead consider NeSy-CBMs that – just like regular CBMs – ***learn the inference layer*** [48, 49, 50, 51, 52].

## 2.1 Reasoning Shortcuts

Before discussing reasoning shortcuts, we need to establish a clear relationship between concepts, inputs, and labels. The RS literature does so by assuming the following ***data generation process*** [19, 21, 53]: each input $\mathbf{x} \in \mathbb{R}^n$ is the result of sampling $k$ ***ground-truth concepts*** $\mathbf{g} = (g_1, \ldots, g_k) \in \mathcal{G}$ (e.g., in MNIST-SumParity two numerical digits) from an unobserved distribution $p^*(\mathbf{G})$ and then $\mathbf{x}$ itself (e.g., two corresponding MNIST images) from the conditional distribution $p^*(\mathbf{X} \mid \mathbf{g})$.[2] Labels $\mathbf{y} \in \mathcal{Y}$ are sampled from the conditional distribution of $\mathbf{g}$ given by $p^*(\mathbf{Y} \mid \mathbf{g}; \mathsf{K})$ consistently with K (e.g., $y = 1$ if and only if $g_1 + g_2$ is odd). The ground-truth distribution is thus:

$$p^*(\mathbf{X}, \mathbf{Y}) = \mathbb{E}_{\mathbf{g} \sim p^*(\mathbf{G})}[p^*(\mathbf{X} \mid \mathbf{g}) p^*(\mathbf{Y} \mid \mathbf{g}; \mathsf{K})] \tag{1}$$

Intuitively, a ***reasoning shortcut*** (RS) occurs when a NeSy-CBM with *fixed* knowledge K attains high or even perfect label accuracy by learning concepts **C** that differ from the ground-truth ones **G**.

**Example 2.2.** *In* MNIST-SumParity*, a NeSy model can achieve perfect accuracy by mapping each pair of MNIST images* $\mathbf{x} = (\mathbf{x}_1, \mathbf{x}_2)$ *to the corresponding ground-truth digits, that is,* $\mathbf{c} = (g_1, g_2)$. *However, it would achieve the same accuracy if it were to map it to* $\mathbf{c} = (g_1 \bmod 2, g_2 \bmod 2)$*, as doing so leaves the parity of the sum unchanged, see Fig. 1. Hence, a NeSy model cannot distinguish between the two based on label likelihood alone during training.*

RSs *by definition* yield good labels in-distribution, yet they compromise out-of-distribution (OOD) performance. For instance, in autonomous driving tasks, NeSy-CBMs can confuse the concepts of "pedestrian" and "red light", leading to poor decisions for OOD decisions where the distinction matters [22]. The meaning of concepts affected by RSs is unclear, affecting understanding [8], intervenability [54, 55, 56], debugging [57, 14] and down-stream applications that hinge on concepts being high-quality, like NeSy formal verification [58, 59, 60]. Unfortunately, existing works on RSs do not apply to CBMs and NeSy-CBMs where the inference layer is learned.

As commonly done, we work in the setting with equal discrete predicted $\mathcal{C}$ and ground-truth $\mathcal{G}$ concept sets, i.e., $\mathcal{G} = \mathcal{C}$ [19, 21]. Notice that, we make no assumption on how the set $\mathcal{G}$ is made; multiple concept vocabularies at different levels of abstraction may be valid for a given task. At this stage, different choices of $\mathcal{G}$ are allowed but may lead to distinct results for RSs, depending on the number of ground-truth concepts and how they are related to the labels. Both these two aspects will be made clear in light of the data generation process as per Assumptions 3.1 and 3.2.

## 3 Reasoning Shortcuts in CBMs

Given a finite set $\mathcal{S}$, we indicate with $\Delta_{\mathcal{S}} \subset [0, 1]^{|\mathcal{S}|}$ the simplex of probability distributions $P(Q)$ over items in $\mathcal{S}$. Any random variable $Q \in \mathcal{S}$ defines a point in the simplex via its distribution $P(Q)$. Notice that the set of the simplex vertices $\mathsf{Vert}(\Delta_{\mathcal{S}})$ contains all point mass distributions $\mathbb{1}\{Q = q\}$ for all $q \in \mathcal{S}$. All relevant notation we will use is reported in Table 6.

**CBMs as pairs of functions**. CBMs and NeSy-CBMs differ in how they implement the inference layer, hence to bridge them we employ the following unified formalism. Any CBM can be viewed as a pair of learnable functions implementing the concept extractor and the inference layer, respectively, cf. Fig. 2. Formally, the former is a function $\mathbf{f} : \mathbb{R}^n \to \Delta_{\mathcal{C}}$ mapping inputs $\mathbf{x}$ to a conditional distribution $p(\mathbf{C} \mid \mathbf{x})$ over the concepts, however it can be better understood as a function $\boldsymbol{\alpha} : \mathcal{G} \to \Delta_{\mathcal{C}}$ taking ground-truth concepts $\mathbf{g}$ as input instead, and defined as:

$$\boldsymbol{\alpha}(\mathbf{g}) := \mathbb{E}_{\mathbf{x} \sim p^*(\mathbf{X} \mid \mathbf{g})}[\mathbf{f}(\mathbf{x})] \tag{2}$$

---

[2]This distribution subsumes stylistic factors, e.g., calligraphy.

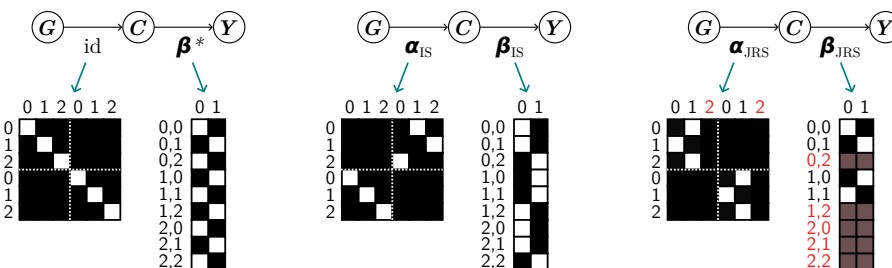

Figure 2: **Examples of semantics in** `MNIST-SumParity` restricted to $\mathbf{g}, \mathbf{c} \in \{0, 1, 2\}^2$ for readability. **Left**: ideally, $\boldsymbol{\alpha}$ should be the identity (i.e., $\mathbf{C}$ recovers the ground-truth concepts $\mathbf{G}$) and the inference layer should learn $\boldsymbol{\beta}^*$. **Middle**: $(\boldsymbol{\alpha}_{\mathrm{IS}}, \boldsymbol{\beta}_{\mathrm{IS}}) \neq (\mathrm{id}, \boldsymbol{\beta}^*)$ has intended semantics (Definition 3.3), i.e., the ground-truth concepts and inference layer can be recovered and generalize OOD. **Right**: $(\boldsymbol{\alpha}_{\mathrm{JRS}}, \boldsymbol{\beta}_{\mathrm{JRS}})$ affected by the Joint Reasoning Shortcuts in Fig. 1. Elements (predicted concepts $\mathbf{C}$ and entries in $\boldsymbol{\beta}$) in red are never predicted nor used, highlighting *simplicity bias*. Maps are visualized as matrices.

In contrast, the inference layer is a function $\boldsymbol{\omega} : \Delta_{\mathcal{C}} \to \Delta_{\mathcal{Y}}$ mapping the concept distribution output by the concept extractor into a label distribution $p(\mathbf{Y} \mid \mathbf{f}(\mathbf{x}))$. For clarity, we also define $\boldsymbol{\beta} : \mathcal{C} \to \Delta_{\mathcal{Y}}$, which is identical to the former except it works with concept *values* rather than *distributions*, that is:

$$\boldsymbol{\beta}(\mathbf{c}) := \boldsymbol{\omega}(\mathbb{1}\{\mathbf{C} = \mathbf{c}\}) \tag{3}$$

Hence, a CBM entails both a pair $(\mathbf{f}, \boldsymbol{\omega}) \in \mathcal{F} \times \Omega$ and a pair $(\boldsymbol{\alpha}, \boldsymbol{\beta}) \in \mathcal{A} \times \mathcal{B}$.[3] Later on, we will make use of the fact that $\mathcal{A}$ and $\mathcal{B}$ are *simplices* [61, 62], i.e., each $\boldsymbol{\alpha} \in \mathcal{A}$ (resp. $\boldsymbol{\beta} \in \mathcal{B}$) can be written as a convex combination of vertices $\mathsf{Vert}(\mathcal{A})$ (resp. $\mathsf{Vert}(\mathcal{B})$).

As mentioned in Section 2, supplying prior knowledge $\mathsf{K}$ to a NeSy-CBM is equivalent to *fixing the inference layer* to a corresponding function $\boldsymbol{\omega}^*$ (and $\boldsymbol{\beta}^*$). Note that whereas $\boldsymbol{\omega}^*$ changes based on how reasoning is implemented – e.g., fuzzy vs. probabilistic logic – $\boldsymbol{\beta}^*$ does not, as both kinds of reasoning layers behave identically when input any point-mass concept distribution $\mathbb{1}\{\mathbf{C} = \mathbf{c}\}$.

**Standard assumptions**. The maps $\boldsymbol{\alpha}$ and $\boldsymbol{\beta}$, induced respectively by the concept extractor and inference layer, are especially useful for analysis provided the following two standard assumptions about how data are distributed [19, 21, 53]:

**Assumption 3.1** (Extrapolability). *The ground-truth distribution $p^*(\mathbf{G} \mid \mathbf{X})$ is induced by a function $\mathbf{f}^* : \mathbf{x} \mapsto \mathbf{g}$, i.e., $p^*(\mathbf{G} \mid \mathbf{X}) = \mathbb{1}\{\mathbf{G} = \mathbf{f}^*(\mathbf{X})\}$.*

This means that the ground-truth concepts $\mathbf{g}$ can always be recovered for all inputs $\mathbf{x}$ by a sufficiently expressive concept extractor; the $\boldsymbol{\alpha}$ it induces is the identity $\mathrm{id}(\mathbf{g}) := \mathbb{1}\{\mathbf{C} = \mathbf{g}\} \in \mathsf{Vert}(\mathcal{A})$.

**Assumption 3.2** (Deterministic knowledge). *The ground-truth distribution $p^*(\mathbf{Y} \mid \mathbf{G}; \mathsf{K})$ is induced by the knowledge via a map $\boldsymbol{\beta}^* \in \mathsf{Vert}(\mathcal{B})$, such that $p^*(\mathbf{Y} \mid \mathbf{G}; \mathsf{K}) = \boldsymbol{\beta}^*(\mathbf{g})$.*

This ensures that the labels $\mathbf{y}$ can be predicted without any ambiguity from the ground-truth concepts $\mathbf{g}$. Notice that, not all choices of $\mathcal{G}$ guarantee that these assumptions are met. Small concept spaces may not give a deterministic knowledge, i.e., labels can be confused for one another with only few concepts, or too-arbitrary choices of the constituents may not guarantee their extrapolation from the input, i.e., ground-truth concepts of the input are ambiguous. Nonetheless, both assumptions hold in several NeSy tasks [22] and underlie many works in RSs [19, 63, 20], but formulating a theory that relaxes them is not straightfoward and is technically challenging. In fact, Marconato et al. [19] showed that upon relaxing Assumption 3.2 it might not be possible to deal with RSs.

The maximum log-likelihood objective is then written as:

$$\max_{(\mathbf{f}, \boldsymbol{\omega}) \in \mathcal{F} \times \Omega} \mathbb{E}_{(\mathbf{x}, \mathbf{y}) \sim p^*(\mathbf{X}, \mathbf{Y})} \log(\boldsymbol{\omega}_{\mathbf{y}} \circ \mathbf{f})(\mathbf{x}) \tag{4}$$

Here, $\boldsymbol{\omega}_{\mathbf{y}}$ is the conditional probability of the ground-truth labels $\mathbf{y}$. Notice that under the above assumptions CBMs attaining maximum likelihood perfectly model the ground-truth data distribution $p^*(\mathbf{X}, \mathbf{Y})$, see Lemma C.1.

---

[3]We work in the non-parametric setting, hence $\mathcal{F}$ and $\Omega$ contain all learnable concept extractors $\mathbf{f}$ and inference layers $\boldsymbol{\omega}$, and similarly $\mathcal{A}$ and $\mathcal{B}$ contain all learnable maps $\boldsymbol{\alpha}$ and $\boldsymbol{\beta}$.

## 3.1 Intended Semantics and Joint Reasoning Shortcuts

We posit that a CBM $(\boldsymbol{\alpha}, \boldsymbol{\beta}) \in \mathcal{A} \times \mathcal{B}$ has "high-quality" concepts and inference layer if it satisfies two desiderata: (i) **disentanglement**: each learned concept $C_i$ should correspond to a single ground-truth concept $G_j$ up to an invertible transformation; (ii) **generalization**: the combination of $\boldsymbol{\alpha}$ and $\boldsymbol{\beta}$ must always yield correct predictions.

In our setting, without concept supervision and prior knowledge, the learned concepts are *anonymous* and users have to figure out which is which in a *post-hoc* fashion, e.g., by aligning them to dense annotations [3, 4, 50, 51]. Doing so is also a prerequisite for understanding the learned inference layer [52, 50, 4, 55]. When **disentanglement** holds the mapping between $\mathbf{G}$ and $\mathbf{C}$ can be recovered *exactly* using Hungarian matching [64]; otherwise it is arbitrarily difficult to recover, hindering interpretability. This links CBMs with many works that treat model representations' *identifiability* in Independent Component Analysis and Causal Representation Learning [65, 66, 67, 68, 69], where disentanglement plays a central role. **Generalization** is equally important, as it entails that CBMs generalize beyond the support of training data and yields sensible OOD behavior, i.e., output the same predictions that would be obtained by using the ground-truth pair. Therefore, the pairs $(\boldsymbol{\alpha}, \boldsymbol{\beta})$ which satisfy both desiderata will be **equivalent** to the ground-truth pair $(\mathrm{id}, \boldsymbol{\beta}^*)$ and equally valid solutions for the NeSy task. Since we want $\boldsymbol{\alpha}$ to be disentangled, this implies, in turn, a specific form for the map $\boldsymbol{\beta}$, as shown by the next definition, which formalizes these desiderata:

**Definition 3.3** (Intended Semantics). *A CBM $(\mathbf{f}, \boldsymbol{\omega}) \in \mathcal{F} \times \Omega$ entailing a pair $(\boldsymbol{\alpha}, \boldsymbol{\beta}) \in \mathcal{A} \times \mathcal{B}$ possesses the intended semantics if there exists a permutation $\pi : [k] \to [k]$ and $k$ element-wise invertible functions $\psi_1, \ldots, \psi_k$ such that:*

$$\boldsymbol{\alpha}(\mathbf{g}) = (\boldsymbol{\psi} \circ \mathbf{P}_\pi \circ \mathrm{id})(\mathbf{g}) \qquad \boldsymbol{\beta}(\mathbf{c}) = (\boldsymbol{\beta}^* \circ \mathbf{P}_\pi^{-1} \circ \boldsymbol{\psi}^{-1})(\mathbf{c}) \qquad (5)$$

*Here, $\mathbf{P}_\pi : \mathcal{C} \to \mathcal{C}$ is the permutation matrix induced by $\pi$ and $\boldsymbol{\psi}(\mathbf{c}) := (\psi_1(c_1), \ldots, \psi_k(c_k))$. In this case, we say that $(\boldsymbol{\alpha}, \boldsymbol{\beta})$ is equivalent to $(\mathrm{id}, \boldsymbol{\beta}^*)$, i.e., $(\boldsymbol{\alpha}, \boldsymbol{\beta}) \sim (\mathrm{id}, \boldsymbol{\beta}^*)$*

Intended semantics holds if the learned concepts $\mathbf{C}$ match the ground-truth concepts $\mathbf{G}$ modulo simple invertible transformations – like reordering and negation (Eq. (5), left) – and the inference layer *undoes* these transformations before applying the "right" inference layer $\boldsymbol{\beta}^*$ (Eq. (5), right). In particular, Eq. (5) (left) guarantees disentanglement of the concepts, ensuring that each learned concept corresponds to a distinct ground-truth concept up to an invertible transformation, matching the notion of [68]. A similar equivalence relation was analyzed for continuous representations in energy-based models, including supervised classifiers, by Khemakhem et al. [65]. CBMs satisfying these conditions are **equivalent** – specifically by the equivalence relation $\sim$, see Appendix C.1 – to the ground-truth pair $(\mathrm{id}, \boldsymbol{\beta}^*)$; see Fig. 2 (middle) for an illustration. In Lemma C.4, we prove that models with the intended semantics yield the same predictions of the ground-truth pair for all $\mathbf{g} \in \mathcal{G}$.

Training a (NeSy) CBM via maximum likelihood does not guarantee it will embody intended semantics. We denote these failure cases as ***joint reasoning shortcuts*** (JRSs):

**Definition 3.4** (Joint Reasoning Shortcut). *Take a CBM $(\mathbf{f}, \boldsymbol{\omega}) \in \mathcal{F} \times \Omega$ that attains optimal log-likelihood (Eq. (4)). The pair $(\boldsymbol{\alpha}, \boldsymbol{\beta})$ entailed by it (Eqs. (2) and (3)) is a JRSs if it does not possess the intended semantics (Definition 3.3), i.e., $(\boldsymbol{\alpha}, \boldsymbol{\beta}) \nsim (\mathrm{id}, \boldsymbol{\beta}^*)$.*

JRSs can take different forms. First, even if the learned inference layer $\boldsymbol{\beta}$ matches (modulo $\mathbf{P}_\pi$ and $\boldsymbol{\psi}$) the ground-truth one $\boldsymbol{\beta}^*$, $\boldsymbol{\alpha}$ might not match (also modulo $\mathbf{P}_\pi$ and $\boldsymbol{\psi}$) the identity. This is analogous to regular RSs in NeSy CBMs (Section 2), in that the learned concepts do not reflect the ground-truth ones: while this is sufficient for high in-distribution performance (the training likelihood is in fact optimal), it may yield poor OOD behavior. Second, even if $\boldsymbol{\alpha}$ matches the identity, the inference layer $\boldsymbol{\beta}$ might not match the ground-truth one $\boldsymbol{\beta}^*$. In our experiments Section 5, we observe that this often stems from *simplicity bias* [70], i.e., the CBM's inference layer tends to acquire specialized functions $\boldsymbol{\beta}$ that are much simpler than $\boldsymbol{\beta}^*$, leading to erroneous predictions OOD. Finally, neither the inference layer $\boldsymbol{\beta}$ nor the map $\boldsymbol{\alpha}$ might match the ground-truth, opening the door to additional failure modes. Consider the following example:

**Example 3.5.** *Consider a* `MNIST-SumParity` *problem where the training set consists of all possible unique (even, even), (odd, odd), and (odd, even) pairs of MNIST digits, as in Fig. 1. A CBM trained on this data would achieve perfect accuracy by learning a JRS that extracts the parity of each input digit, yielding two binary concepts in $\{0, 1\}$, and computes the difference between these two concepts. This JRS involves **much simpler concepts and knowledge** compared to the ground-truth ones. It also mispredicts all OOD pairs where the even digits come before odd digits.*

While the presence of JRSs undermines both interpretability and OOD generalization, their absence alone does not guarantee OOD robustness. However, if JRSs are present, the likelihood of OOD failure increases substantially. For example, consider Example 3.5, where a CBM achieves perfect training accuracy by learning a shortcut that captures only the parity of each digit rather than the full ground-truth concepts. This shortcut fails on OOD pairs; for example, given an (even, odd) pair not present in the training set, the model wrongly outputs $-1$ as the final label.

## 3.2 Theoretical Analysis

We now count the *deterministic* JRSs admitted by a task, i.e., those that lie on the vertices of $\mathcal{A}$ and $\mathcal{B}$: $(\boldsymbol{\alpha}, \boldsymbol{\beta}) \in \mathsf{Vert}(\mathcal{A}) \times \mathsf{Vert}(\mathcal{B})$. We then show Theorem 3.9 that this number determines whether general (non-deterministic) JRSs exist.

**Theorem 3.6** (Informal). *Under Assumptions 3.1 and 3.2, the number of deterministic JRSs is:*

$$\sum_{(\boldsymbol{\alpha}, \boldsymbol{\beta})} \mathbb{1}\{\bigwedge_{\mathbf{g} \in \mathrm{supp}(\mathbf{G})} (\boldsymbol{\beta} \circ \boldsymbol{\alpha})(\mathbf{g}) = \boldsymbol{\beta}^*(\mathbf{g})\} - C[\mathcal{G}] \tag{6}$$

*where the sum runs over* $\mathsf{Vert}(\mathcal{A}) \times \mathsf{Vert}(\mathcal{B})$, *and* $C[\mathcal{G}]$ *counts the pairs with intended semantics.*

All proofs and the definition of $C[\mathcal{G}]$ can be found in Appendix C. Intuitively, the first count includes all deterministic pairs $(\boldsymbol{\alpha}, \boldsymbol{\beta})$ that achieve maximum likelihood on the training set (i.e., the predicted labels $(\boldsymbol{\beta} \circ \boldsymbol{\alpha})(\mathbf{g})$ matches the ground-truth one $\boldsymbol{\beta}^*(\mathbf{g})$ for all ground-truth concepts $\mathbf{g}$ appearing in the data), while the second term subtracts those $(\boldsymbol{\alpha}, \boldsymbol{\beta})$ that possess the intended semantics as per Definition 3.3. A positive count implies that a CBM trained via maximum likelihood *can* learn a deterministic JRS. Notice that the count of JRSs also depends on the choice of the concept space $\mathcal{G}$: a large number of fine-grained ground-truth concept may increase the number of JRSs, while coarser-grained concepts that are lower in number may reduce them.

As a sanity check, we show that if prior knowledge K is provided – fixing the inference layer to $\boldsymbol{\beta}^*$ – the number of deterministic JRSs in fact matches that of RSs, as expected:

**Corollary 3.7.** *Consider a fixed* $\boldsymbol{\beta}^* \in \mathsf{Vert}(\mathcal{B})$. *Under Assumptions 3.1 and 3.2, the number of deterministic JRSs* $(\boldsymbol{\alpha}, \boldsymbol{\beta}^*) \in \mathsf{Vert}(\mathcal{A}) \times \mathsf{Vert}(\mathcal{B})$ *is:*

$$\sum_{\boldsymbol{\alpha} \in \mathsf{Vert}(\mathcal{A})} \mathbb{1}\{\bigwedge_{\mathbf{g} \in \mathrm{supp}(\mathbf{G})} (\boldsymbol{\beta}^* \circ \boldsymbol{\alpha})(\mathbf{g}) = \boldsymbol{\beta}^*(\mathbf{g})\} - 1 \tag{7}$$

*This matches the count for deterministic RSs in [19].*

Corollary 3.7 implies that when $|\mathsf{Vert}(\mathcal{B})| > 1$, the number of deterministic JRSs in Eq. (6) can be much *larger* than the number of deterministic RSs (Eq. (7)). For example, in `MNIST-SumParity` with digits restricted to the range $[0, 4]$ there exist $64$ RSs but about $100$ *thousand* JRSs, and the gap increases as we extend the range $[0, N]$. An in-depth analysis appears in Appendix B.5.

Next, we show that whenever the number of deterministic JRSs in Eq. (6) is zero, there exist ***no JRSs at all***, including non-deterministic ones. This however only applies to CBMs that satisfy the following natural assumption:

**Assumption 3.8** (Extremality). *The inference layer* $\boldsymbol{\omega} \in \Omega$ *satisfies extremality if, for all* $\lambda \in (0, 1)$ *and all* $\mathbf{c} \neq \mathbf{c}' \in \mathcal{C}$ *such that* $\arg\max_{\mathbf{y} \in \mathcal{Y}} \boldsymbol{\omega}(\mathbb{1}\{\mathbf{C} = \mathbf{c}\})_{\mathbf{y}} \neq \arg\max_{\mathbf{y} \in \mathcal{Y}} \boldsymbol{\omega}(\mathbb{1}\{\mathbf{C} = \mathbf{c}'\})_{\mathbf{y}}$, *it holds:*

$$\max_{\mathbf{y} \in \mathcal{Y}} \boldsymbol{\omega}(\lambda \mathbb{1}\{\mathbf{C} = \mathbf{c}\} + (1 - \lambda)\mathbb{1}\{\mathbf{C} = \mathbf{c}'\})_{\mathbf{y}} < \max\left(\max_{\mathbf{y} \in \mathcal{Y}} \boldsymbol{\omega}(\mathbb{1}\{\mathbf{C} = \mathbf{c}\})_{\mathbf{y}}, \max_{\mathbf{y} \in \mathcal{Y}} \boldsymbol{\omega}(\mathbb{1}\{\mathbf{C} = \mathbf{c}'\})_{\mathbf{y}}\right)$$

Intuitively, a CBM satisfies this assumption if its inference layer $\boldsymbol{\omega}$ is "peaked": for any two concept vectors $\mathbf{c}$ and $\mathbf{c}'$ yielding distinct predictions, the label probability output by $\boldsymbol{\omega}$ for any mixture distribution thereof is no larger than the label probability that it associates to $\mathbf{c}$ and $\mathbf{c}'$. That is, distributing probability mass across concepts does not increase label likelihood. While this assumption does not hold for general CBMs, we show in Appendix E that it holds for popular architectures, including most of those that we experiment with. Under Assumption 3.8, we have:

**Theorem 3.9** (Identifiability). *Under Assumptions 3.1 and 3.2, **if** the number of deterministic JRSs (Eq. (6)) is zero **then** every CBM* $(\mathbf{f}, \boldsymbol{\omega}) \in \mathcal{F} \times \Omega$ *satisfying Assumption 3.8 that attains maximum log-likelihood (Eq. (4)) possesses the intended semantics (Definition 3.3). That is, the pair* $(\boldsymbol{\alpha}, \boldsymbol{\beta}) \in \mathcal{A} \times \mathcal{B}$ *entailed by each such CBM is equivalent to the ground-truth pair* $(\mathrm{id}, \boldsymbol{\beta}^*)$, *i.e.,* $(\boldsymbol{\alpha}, \boldsymbol{\beta}) \sim (\mathrm{id}, \boldsymbol{\beta}^*)$.

Clearly, a similar conclusion does not hold for models that do not satisfy Assumption 3.8: even when deterministic JRSs are absent, a CBM can still be affected by non-deterministic JRSs.

# 4 Practical Solutions

By Theorem 3.9, getting rid of *deterministic* JRSs from a task prevents CBMs trained via maximum log-likelihood to learn JRS altogether. The key question is how to make the JRSs count be zero. As previously explored in RSs, pairing the maximum likelihood objective with other well-known penalties in the literature can limit optimal $\alpha$'s for the joint training objective. Several mitigation strategies for RSs have shown to decrease the count of the $\alpha$'s, in some cases leading to zeroing deterministic RSs [19, 21]. We report the count reduction of different mitigations in Appendix D.

**Supervised strategies**. The most direct strategies for controlling the semantics of learned CBMs rely on supervision. ***Concept supervision*** is the go-to solution for improving concept quality in CBMs [3, 71, 4, 5] and NeSy-CBMs [22, 21]. Full concept supervision prevents learning $\alpha \neq \mathrm{id}$. However, this does not translate into guarantees on the inference layer $\beta$, at least in tasks affected by confounding factors such as spurious correlations among concepts [14]. E.g., if the $i$-th and $j$-th ground-truth concepts are strongly correlated, the inference layer $\beta$ can interchangeably use either, regardless of what $\beta^*$ does.

Another option is employing ***knowledge distillation*** [72], that is, supplying supervision of the form $(\mathbf{g}, \mathbf{y})$ to encourage the learned knowledge $\beta$ to be close to $\beta^*$ for the supervised $\mathbf{g}$'s. This supervision is available for free provided concept supervision is available, but can also be obtained separately, e.g., by interactively collecting user interventions [54, 55, 56]. This strategy cannot avoid *all* JRSs because even if $\beta = \beta^*$, the CBM may suffer from regular RSs (i.e., $\alpha$ does not match the identity function $\mathrm{id}$).

Another option is to fit a CBM on ***multiple tasks*** [73] sharing concepts. It is known that a NeSy-CBM attaining optimal likelihood on multiple NeSy tasks with different prior knowledges is also optimal for the *single* NeSy task obtained by conjoining those knowledges [19]: this more constrained knowledge better steers the semantics of the learned concepts, ruling out potential JRSs. Naturally, collecting a sufficiently large number of diverse tasks using the same concepts can be highly non-trivial, depending on the application.

**Unsupervised strategies**. Many popular strategies for improving concept quality in (NeSy) CBMs are unsupervised. A cheap but effective one when dealing with multiple inputs is to ***process inputs individually***, preventing mixing between their concepts. E.g., In `MNIST-SumParity` one can apply the same digit extractor to each digit separately, while for images with multiple objects one can first segment them (e.g., with YoloV11 [74]) and then process the resulting bounding boxes individually. This is extremely helpful for reducing, and possibly overcoming, RSs [19] and frequently used in practice [41, 75, 50, 51]. We apply it in all our experiments.

Both supervised [5] and unsupervised [1, 76] CBMs may employ a ***reconstruction*** penalty [77, 78] to encourage learning informative concepts. Reconstruction can help prevent collapsing distinct ground-truth concepts into single learned concepts, e.g., in `MNIST-SumParity` the odd digits cannot be collapsed together without impairing reconstruction.[4] Alternatively, one can employ ***entropy maximization*** [79] to spread concept activations evenly across the bottleneck. This can be viewed as a less targeted but more efficient alternative to reconstruction, which becomes impractical for higher dimensional inputs. Another option is ***contrastive learning*** [80], in that augmentations render learned concepts more robust to style variations [81], e.g., for MNIST-based tasks it helps to cluster distinct digits [82]. Unsupervised strategies all implicitly counteract *simplicity bias* whereby $\alpha$ ends up collapsing.

# 5 Case Studies

We tackle the following key research questions: **Q1**. Are CBMs affected by JRSs in practice? **Q2**. Do JRSs affect interpretability and OOD behavior? **Q3**. Can existing mitigation strategies prevent JRSs? Appendix A reports additional details about the tasks, architectures, and model selection.

**Models.** We evaluate several (also NeSy) CBMs. DeepProbLog (DPL) [41, 79] is the *only* method supplied with the ground-truth knowledge K, and uses probabilistic-logic reasoning to ensure predictions are consistent with it. CBNM is a Concept-Bottleneck Model [3] with no supervision on

---

[4]This is provably the case *context-style separation* i.e., assuming concepts are independent of stylistic features like calligraphy [19, Proposition 6].

Table 1: Results for `MNIST-Add`.

| METHOD | $F_1(Y)$ (↑) | $F_1(C)$ (↑) | $\mathrm{Cls}(C)$ (↓) | $F_1(\boldsymbol{\beta})$ (↑) |
|---|---|---|---|---|
| DPL | $0.98 \pm 0.01$ | $0.99 \pm 0.01$ | $0.01 \pm 0.01$ | — |
| CBNM | $0.98 \pm 0.01$ | $0.99 \pm 0.01$ | $0.01 \pm 0.01$ | $1.00 \pm 0.01$ |
| SENN | $0.97 \pm 0.01$ | $0.80 \pm 0.07$ | $0.01 \pm 0.01$ | $0.75 \pm 0.08$ |
| DSL | $0.96 \pm 0.02$ | $0.98 \pm 0.01$ | $0.01 \pm 0.01$ | $1.00 \pm 0.01$ |
| DPL* | $0.91 \pm 0.09$ | $0.92 \pm 0.08$ | $0.01 \pm 0.01$ | $0.90 \pm 0.08$ |
| bears* | $0.76 \pm 0.15$ | $0.87 \pm 0.13$ | $0.01 \pm 0.01$ | $0.67 \pm 0.14$ |

Table 2: Results for `MNIST-SumParity`.

| METHOD | $F_1(Y)$ (↑) | $F_1(C)$ (↑) | $\mathrm{Cls}(C)$ (↓) | $F_1(\boldsymbol{\beta})$ (↑) |
|---|---|---|---|---|
| DPL | $0.99 \pm 0.01$ | $0.43 \pm 0.08$ | $0.36 \pm 0.15$ | — |
| CBNM | $0.90 \pm 0.18$ | $0.09 \pm 0.03$ | $0.66 \pm 0.09$ | $0.54 \pm 0.04$ |
| SENN | $0.99 \pm 0.01$ | $0.49 \pm 0.05$ | $0.01 \pm 0.01$ | $0.53 \pm 0.06$ |
| DSL | $0.94 \pm 0.03$ | $0.07 \pm 0.03$ | $0.80 \pm 0.01$ | $0.52 \pm 0.01$ |
| DPL* | $0.99 \pm 0.01$ | $0.07 \pm 0.01$ | $0.80 \pm 0.01$ | $0.50 \pm 0.05$ |
| bears* | $0.99 \pm 0.01$ | $0.30 \pm 0.03$ | $0.22 \pm 0.04$ | $0.36 \pm 0.09$ |

Table 3: Results for `Clevr`.

| METHOD | $F_1(Y)$ (↑) | $F_1(C)$ (↑) | $\mathrm{Cls}(C)$ (↓) | $F_1(\boldsymbol{\beta})$ (↑) |
|---|---|---|---|---|
| DPL | $0.99 \pm 0.01$ | $0.25 \pm 0.05$ | $0.57 \pm 0.02$ | — |
| CBNM | $0.99 \pm 0.01$ | $0.19 \pm 0.06$ | $0.35 \pm 0.09$ | $0.01 \pm 0.02$ |
| DPL* | $0.99 \pm 0.01$ | $0.22 \pm 0.04$ | $0.57 \pm 0.03$ | $0.01 \pm 0.01$ |

Table 4: CBNM on biased `MNIST-SumParity`.

| C-SUP | K-SUP | ID | | OOD | |
|---|---|---|---|---|---|
| | | $F_1(Y)$ (↑) | $F_1(\boldsymbol{\beta})$ (↑) | $F_1(Y)$ (↑) | $F_1(\boldsymbol{\beta})$ (↑) |
| 0% | 0% | $0.99 \pm 0.01$ | $0.55 \pm 0.05$ | $0.01 \pm 0.01$ | $0.47 \pm 0.07$ |
| 0% | 100% | $0.99 \pm 0.01$ | $0.56 \pm 0.06$ | $0.01 \pm 0.01$ | $0.58 \pm 0.08$ |
| 100% | 0% | $0.97 \pm 0.01$ | $0.88 \pm 0.04$ | $0.40 \pm 0.14$ | $0.31 \pm 0.12$ |
| 100% | 100% | $0.97 \pm 0.01$ | $0.95 \pm 0.01$ | $0.97 \pm 0.02$ | $0.69 \pm 0.27$ |

Table 5: Results for `BDD-OIA`.

| METHOD | $F_1(Y)$ (↑) | $F_1(C)$ (↑) | $\mathrm{Cls}(C)$ (↓) | $F_1(\boldsymbol{\beta})$ (↑) |
|---|---|---|---|---|
| DPL | $0.69 \pm 0.03$ | $0.44 \pm 0.01$ | $0.88 \pm 0.01$ | — |
| CBNM | $0.62 \pm 0.03$ | $0.43 \pm 0.02$ | $0.06 \pm 0.02$ | $0.42 \pm 0.03$ |

the concepts. Deep Symbolic Learning (`DSL`) [50] is a SOTA NeSy-CBM that learns concepts and symbolic knowledge jointly; it implements the latter as a truth table and reasoning using fuzzy logic. We also consider two variants: `DPL*` replaces fuzzy with probabilistic logic, while `bears*` wraps DPL* within BEARS [63], an ensemble approach for handling regular RSs. In short, `bears*` consists of a learned inference layer and multiple concept extractors. The former is learned only once, the latter are learned sequentially. We also evaluate Self-explainable Neural Networks (`SENN`) [1], unsupervised CBMs that include a reconstruction penalty. Since they do not satisfy our assumptions, they are useful to empirically assess the generality of our remarks. In our experiments, we train all CBM variants in a joint manner [3] using only label supervision and do not rely on foundation models for gathering annotations [32, 33, 34]. For additional details, see Appendix A.

**Data sets.** To assess both learned concepts and inference layer, we use three representative NeSy tasks with explicit concept annotations and prior knowledge. `MNIST-Add` [41] involves predicting the sum of two MNIST digits, and serves as a sanity check as it provably contains no RSs. `MNIST-SumParity` is similar except we have to predict the *parity* of the sum, as in Fig. 1, and admits many JRSs. We also include a variant of `Clevr` [83] in which images belong to 3 classes as determined by a logic formula and only contain 2 to 4 objects, for scalability. Finally, we carry out basic checks also on `BDD-OIA` [84], a real-world autonomous driving task where images are annotated with actions (`move_forward`, `stop`, `turn_left`, `turn_right`) and 21 binary concepts.

**Metrics.** For each CBM, we evaluate predicted labels and concepts with the $F_1$ score (resp. $F_1(Y)$ and $F_1(C)$) on the test split. Computing $F_1(C)$ requires to first align the predicted concepts to ground-truth annotations. We do so by using the Hungarian algorithm [64] to find a permutation $\pi$ that maximizes the Pearson correlation across concepts. In this way, we directly evaluate *disentanglement* of the concepts as per Eq. (5) (left). Concept collapse $\mathrm{Cls}(C)$ quantifies to what extent the concept extractor blends different *ground-truth* concepts together: high collapse indicates that it predicts fewer concepts than expected. We measure the quality of the learned inference layer $\boldsymbol{\beta}$ as follows: for each input $\mathbf{x}$, (1) we reorder the corresponding concept annotations $\mathbf{g}$ using $\pi$, (2) feed the result to $\boldsymbol{\beta}$, and (3) compute the $F_1$-score ($F_1(\boldsymbol{\beta})$) of its predictions. Step (1) permutes and applies element-wise invertible transformations on the ground-truth concepts using the inverse of the map returned by Hungarian matching. This evaluates whether the learned inference layer can correctly handle (an invertible transformation of) the ground-truth concept annotations, measuring the *generalization* as per Eq. (5) (right).This procedure is precisely detailed in Appendix A.6.

**Mitigation strategies**. We evaluate all strategies discussed in Section 4 as well as a combination of all unsupervised mitigations, denoted `H+R+C` (entropy, reconstruction, contrastive). Implementations are taken verbatim from [22]. For contrastive learning, we apply an InfoNCE loss to the predicted concept distribution with the augmentations suggested by Chen et al. [80]. For knowledge distillation, the inference layer is fed ground-truth concepts and trained to optimize the log-likelihood of the corresponding labels.

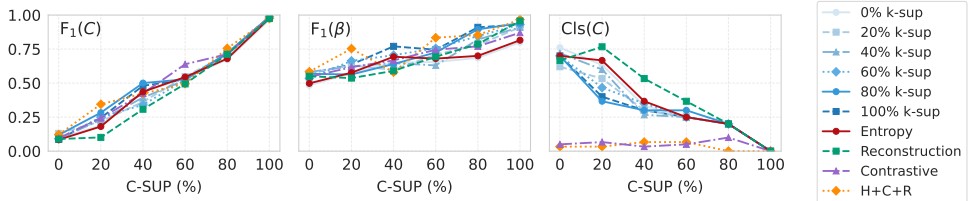

Figure 3: **Traditional mitigations have limited effect** on `CBNM` for `MNIST-SumParity`. The only outlier is contrastive learning (orange and purple), which consistently ameliorates concept collapse.

**Q1: All tested CBMs suffer from JRSs and simplicity bias.** We analyze three learning tasks of increasing complexity. `MNIST-Add` provably contains no JRSs and thus satisfies the preconditions of Theorem 3.9. Compatibly, all CBMs attain high label performance ($F_1(Y) \geq 90\%$) and high-quality semantics, as shown by the values of $F_1(C)$ ($\geq 90\%$) and $F_1(\beta)$ ($\geq 75\%$) seen in Table 1. The only exception is `bears`*, which by design trades off performance for calibration. However, in `MNIST-SumParity` and `Clevr`,[5] which are more complex, ***all models lack intended semantics***. This is clear from Tables 2 and 3: despite attaining high $F_1(Y)$ ($\geq 90\%$), the learned concepts and inference layer are low quality: $F_1(C)$ is always low – it never crosses the $50\%$ mark – and $F_1(\beta)$ is at best around chance level. Our results on `BDD-OIA` in Table 5 show the very same trend.

The behavior of concept collapse strongly hints at *simplicity bias*. While no collapse takes place in `MNIST-Add` ($\text{Cls}(C) \leq 0.01$, i.e., digits are not compacted together), in `MNIST-SumParity` and `Clevr` collapse is omnipresent ($\text{Cls}(C) \geq 0.22$), suggesting that CBMs are prone to simplicity bias, as expected. `SENN` is an outlier, likely due to the higher capacity of its inference layer (rather than reconstruction, which is entirely ineffective in **Q3**). Even having access to the ground-truth knowledge is ineffective, as shown by the high degree of collapse displayed by `DPL`.

**Q2: JRSs compromise OOD behavior and supervision only partially helps**. We evaluate the impact of JRSs and supervision on OOD behavior by training a `CBNM` on the biased version of `MNIST-SumParity` in Fig. 1. The in-distribution (ID) data comprise all (odd, odd), (even, even), and (odd, even) pairs of MNIST digits, while the OOD split contains all (even, odd) pairs. Table 4 reports performance under varying levels of concept and knowledge supervision. While all models produce high-quality predictions in-distribution ($F_1(Y) \geq 0.97$), only the `CBNM` receiving complete supervision attains acceptable OOD predictions: for the others, $F_1(Y) \leq 0.40$. Even in this case, though, the inference layer still does not have intended semantics, as shown by $F_1(\beta) \leq 70\%$.

**Q3: Popular CBM and RS mitigations fall short**. Finally, we test mitigation strategies on `CBNM` trained on `MNIST-SumParity` and `Clevr`, ablating the amount of supervision. The results for `MNIST-SumParity` in Fig. 3 show while concept supervision ($x$-axis) helps all metrics, adding reconstruction, entropy maximization, and contrastive learning to the mix brings no benefits for concept and knowledge ($F_1(C)$, $F_1(\beta)$) quality, not even when combined (orange curve). Knowledge supervision is also ineffective (blue lines), likely because `MNIST-SumParity` and `Clevr` suffer from RSs even when the model is supplied prior knowledge. Contrastive learning improves concept collapse ($\text{Cls}(C)$): the orange and purple curves in Fig. 3 (right) show it can prevent CBMs from mixing concepts together. The results for `Clevr` in Appendix B show similar trends.

## 6 Related Work

**Shortcuts in machine learning**. Shortcuts occur when machine learning models solve a prediction task by using features that correlate with but do not cause the desired output, leading to poor OOD behavior [12, 10, 85, 86]. Existing work focuses on black-box models rather than CBMs. Existing studies on the semantics of CBM concepts [87, 15, 88, 89] focus on individual failures but lack formal notion of intended semantics. One exception is [17] which, however, ignores the role of the inference layer. We build on known results on reasoning shortcuts [90, 19, 20, 53] which are restricted to NeSy-CBMs in which the prior knowledge is given and fixed. Our analysis upgrades these results to general CBMs. Furthermore, our simulations indicate that strategies that are effective

---

[5]For `Clevr`, we focus on representative CBMs with high fit on the training data.

for CBMs and RSs no longer work for JRSs. At the same time, while NeSy approaches that learn prior knowledge and concepts jointly are increasingly popular [48, 91, 49, 50, 51, 52], existing work neglects the issue of shortcuts in this setting. Our work partially fills this gap. Shortcuts and JRSs are related but distinct. While shortcuts can induce JRSs, the converse is not true: a model can be affected by a JRS even if it does not rely on confounders, as shown in Fig. 2. This distinction also implies that mitigation strategies designed for vanilla shortcuts do not necessarily transfer to JRSs.

**Relation to identifiability**. Works in Independent Component Analysis and Causal Representation Learning focus on the identifiability of representations up to an equivalence relation, typically up to permutations and rescaling [92, 93, 69] or more general transformations [94, 95]. The equivalence relation introduced along with intended semantics (Definition 3.3) establishes a specific connection between the maps $\alpha$ and $\beta$. This aligns with existing works that aim to identify representations and linear inference layers using supervised labels [68, 96, 97, 98, 99]. Moreover, while prior works primarily focus on identifying real-valued representations, we examine the identifiability of *discrete* concepts *and* of the inference layer. [100] explores the identifiability of (potentially discrete) concepts linearly encoded in model representations but focuses on multi-environment settings. Our results differ in that we show how concepts can be identified in CBMs by circumventing reasoning shortcuts. A concurrent work by Goyal et al. [101] identifies latent concepts through label supervision and a sparsity regularization, showing that a variant of CBNMs can provably recover concepts with the intended semantics and thus avoiding JRSs altogether, see [101, Theorem 2].

# 7 Conclusion

We study the issue of learning CBMs and NeSy-CBMs with high-quality concepts and inference layers. We formalize intended semantics and joint reasoning shortcuts (JRSs), showing how, under suitable assumptions, zeroing the number of *deterministic* JRSs provably prevents *all* JRSs, yielding identifiability. Numerical simulations indicate that JRSs are very frequent and impactful, and that the only (partially) beneficial mitigation stratgy is contrastive learning. Our work paves the way to the design of effective solutions and therefore more robust and interpretable CBMs. In future work, we plan to extend our theory to general neural nets and large language models, and we will take a closer look at more complex mitigations such as smartly constraining the inference layer [49, 52], debiasing [13], and human interaction [10]. Another asset is considering concept-level supervision gathered from foundation models like CLIP, however recent results [35] call for caution in using them to supervise the bottleneck. Wrong concept-level supervision may transfer biases of foundation models and induce JRSs, thus resulting ineffective as a mitigation strategy.

**Broader impact**. With this work, we aim to highlight the issue of joint reasoning shortcuts in concept-based models and their subtle but impactful consequences on the trustworthiness and interpretability of these models, as well as on their reliability in out-of-distribution scenarios. It also highlights the limited effect of unsupervised mitigation strategies, thus pointing out a significant gap in our toolbox for addressing joint reasoning shortcuts effectively and cheaply. Our work also provides a solid theoretical foundation upon which further studies can build.

**Limitations**. Our theoretical analysis of JRSs builds on two common assumptions for the data generation process, which limit its scope. Relaxing them, while not straightforward, will allow treating models and NeSy tasks with intrinsic uncertainty on label and concepts. Our experiments span both synthetically-generated and a real-world vision datasets; further analysis should highlight the impact of JRSs in other real-world settings and beyond the vision domain.

### Acknowledgements

The authors are grateful to Antonio Vergari, Emile van Krieken, Marco Pacini, and Luigi Gresele for useful discussions. Funded by the European Union. Views and opinions expressed are however those of the author(s) only and do not necessarily reflect those of the European Union or the European Health and Digital Executive Agency (HaDEA). Neither the European Union nor the granting authority can be held responsible for them. Grant Agreement no. 101120763 - TANGO. The work of AP was partially supported by the MUR PNRR project FAIR - Future AI Research (PE00000013) funded by the NextGenerationEU. PM was supported by the MSCA project Probabilistic Formal Verification for Provably Trustworthy AI - PFV-4-PTAI under GA no. 101110960.

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

# A    Implementation Details

Here, we provide additional details about metrics, datasets, and models, for reproducibility. All the experiments were implemented using Python 3.9 and Pytorch 1.13 and run on one A100 GPU. The implementations of DPL, and the code for RSs mitigation strategies were taken verbatim from [19], while the code for DSL was taken from [50]. DPL$^*$ and `bears`$^*$ were implement on top of DSL codebase. The complete codebase is publicly available at `https://github.com/samuelebortolotti/joint-reasoning-shortcuts`.

## A.1    Notation

We summarize the notation used in this paper.

Table 6: Glossary of common symbols.

| Symbol | Meaning |
|---|---|
| $x, y, z$ | Scalar constants |
| $X, Y, Z$ | Scalar variables |
| $\mathbf{x}, \mathbf{X}$ | Vector constant and vector variable |
| | |
| $\mathcal{X}$ | Space of input |
| $\mathcal{Y}, \mathcal{G}, \mathcal{C}$ | Sets of labels, ground-truth, and learned concepts |
| $\Delta_{\mathcal{Y}}, \Delta_{\mathcal{G}}, \Delta_{\mathcal{C}}$ | Simplices of probability distributions on sets elements |
| | |
| $\mathbf{f} : \mathcal{X} \to \Delta_{\mathcal{C}}$ | Concept extractor |
| $\boldsymbol{\omega} : \Delta_{\mathcal{C}} \to \Delta_{\mathcal{Y}}$ | Inference layer |
| $\mathcal{F}, \Omega$ | Space of concept extractors and inference layers |
| | |
| $\boldsymbol{\alpha} : \mathcal{G} \to \Delta_{\mathcal{C}}$ | Map from ground-truth concepts to distribution of model concepts |
| $\boldsymbol{\beta} : \mathcal{C} \to \Delta_{\mathcal{Y}}$ | Map of the inference layer on concept vectors |
| $\mathcal{A}, \mathcal{B}$ | Spaces of $\boldsymbol{\alpha}$'s and $\boldsymbol{\beta}$'s |
| | |
| $p^*$ | distribution over inputs, label, and ground-truth concepts |
| K | Prior knowledge |
| $\models$ | Logical entailment |

## A.2    Datasets

All data sets were generated using the `rsbench` library [22].

### A.2.1    `MNIST-Add`

`MNIST-Add` [41] consists of pairs of `MNIST` digits [45], ranging from 0 to 9, and the goal is to predict their sum. The prior knowledge used for generating the labels is simply the rule of addition: $K = (Y = C_1 + C_2)$. This task does not admit RSs nor JRSs when digits are processed separately. The training and test examples take the following form:

$$\{((\boxed{0}, \boxed{1}), 1), \ ((\boxed{5}, \boxed{8}), 13), \ ((\boxed{6}, \boxed{6}), 12), \ ((\boxed{4}, \boxed{1}), 5)\} \tag{8}$$

All splits cover all possible pairs of concepts, from $(0,0)$ to $(9,9)$, i.e., there is no sampling bias. Overall, `MNIST-Add` has $42,000$ training examples, $12,000$ validation examples, and $6,000$ test examples.

## A.3    `MNIST-SumParity`

Similarly, in `MNIST-SumParity` the goal is to predict the *parity* of the sum, i.e., the prior knowledge is the rule: $K = (Y = (C_1 + C_2) \bmod 2)$. This task is more challenging for our purposes, as it is affected by both RSs and JRSs, cf. Fig. 1. Below we report four examples:

$$\{((\boxed{0}, \boxed{1}), 1), \ ((\boxed{5}, \boxed{8}), 1), \ ((\boxed{6}, \boxed{6}), 0), \ ((\boxed{4}, \boxed{1}), 1)\} \tag{9}$$

We consider two variants: like `MNIST-Add`, in `MNIST-SumParity` proper the training and test splits contain all possible combinations of digits, while in **_biased_** `MNIST-SumParity` the training set contains (even, even), (odd, odd) and (odd, even) pairs, while the test set contains (even, odd) pairs only. This fools CBMs into learning an incorrect inference layer implementing subtraction instead of addition-and-parity, as illustrated in Fig. 1. Both variants have the same number of training, validation, and test examples as `MNIST-Add`.

### A.4 `Clevr`

`Clevr` [83] consists of of 3D scenes of simple objects rendered with Blender, see Fig. 4. Images are $3 \times 128 \times 128$ and contain a variable number of objects. We consider only images with 2 to 4 objects each, primarily due to scalability issues with DPL. The ground-truth labels were computed by applying the rules (prior knowledge) proposed by [14], reported in Table 8. Objects are entirely determined by four concepts, namely shape, color, material, and size. We list all possible values in Table 7. Overall, `Clevr` has 6000 training examples, 1200 validation examples, and 1800 test examples.

| Property | Value |
|---|---|
| **Shapes** | Cube, Sphere, Cylinder |
| **Colors** | Gray
Red, Blue, Green, Brown,
Purple, Cyan, Yellow |
| **Materials** | Rubber, Metal |
| **Sizes** | Large, Small |

| Class | Condition |
|---|---|
| **Class 1** | $\exists x_1 \, (\text{Cube}(x_1) \wedge \text{Large}(x_1))$
$\wedge \exists x_2 \, (\text{Cylinder}(x_2) \wedge \text{Large}(x_2))$ |
| **Class 2** | $\exists x_1 \, (\text{Cube}(x_1) \wedge \text{Small}(x_1) \wedge \text{Metal}(x_1))$
$\wedge \exists x_2 \, (\text{Sphere}(x_2) \wedge \text{Small}(x_2))$ |
| **Class 3** | $\exists x_1 \, (\text{Sphere}(x_1) \wedge \text{Large}(x_1) \wedge \text{Blue}(x_1))$
$\wedge \exists x_2 \, (\text{Sphere}(x_2) \wedge \text{Small}(x_2) \wedge \text{Yellow}(x_2))$ |

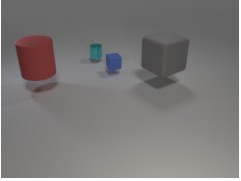

Table 7: `Clevr` properties.          Table 8: `Clevr` classes.          Figure 4: Example of `Clevr` data.

**Preprocessing**. All CBMs process objects separately, as follows. Following [102], we fine-tune a pretrained `YoloV11` model [74] (for 10 epochs, random seed 13) on a subset of the training set's bounding boxes, and use it to predict bounding boxes for all images. For each training and test image, we concatenate the regions in the bounding boxes, obtaining a list 2 to 4 images which plays the role of input for our CBMs.

Each object has $8 \times 3 \times 2 \times 2 = 96$ possible configurations of concept values, and we handle between 2 and 4 objects, hence the inference layer has to model $96^2 + 96^3 + 96^4$ distinct possibilities. Due to the astronomical number of possibilities (which would entail a huge truth table/inference layer for DSL and related CBMs), we split the inference layer in two: we first predict individual concepts, and then we aggregate them (into, e.g., `large-cube` and `small-yellow-sphere`) and use these for prediction. Despite this simplification, `Clevr` still induces JRSs in learned CBMs.

### A.5 `BDD-OIA`

`BDD-OIA` is an autonomous driving dataset, composed of images extracted from driving scene videos [84]. Each image is annotated with four binary labels: `move_forward`, `stop`, `turn_left`, and `turn_right`. Each image comes with 21 binary concepts that constitute the explanations for the corresponding actions. The training set contains almost 16k fully labeled frames, while the validation and test sets include almost 2k and 4.5k annotated samples, respectively. We adopt the same prior knowledge used in [22], which we briefly summarize here for the sake of completeness.

For the `move_forward` and `stop` actions, the rules are as follows:

$$\begin{cases} \texttt{red\_light} \Rightarrow \neg\texttt{green\_light} \\ \texttt{obstacle} = \texttt{car} \vee \texttt{person} \vee \texttt{rider} \vee \texttt{other\_obstacle} \\ \texttt{road\_clear} \Longleftrightarrow \neg\texttt{obstacle} \\ \texttt{green\_light} \vee \texttt{follow} \vee \texttt{clear} \Rightarrow \texttt{move\_forward} \\ \texttt{red\_light} \vee \texttt{stop\_sign} \vee \texttt{obstacle} \Rightarrow \texttt{stop} \\ \texttt{stop} \Rightarrow \neg\texttt{move\_forward} \end{cases} \quad (10)$$

The rules for the `turn_left` and the `turn_right` action, instead, are:

$$\begin{cases} \texttt{can\_turn} = \texttt{left\_lane} \lor \texttt{left\_green\_lane} \lor \texttt{left\_follow} \\ \texttt{cannot\_turn} = \texttt{no\_left\_lane} \lor \texttt{left\_obstacle} \lor \texttt{left\_solid\_line} \\ \texttt{can\_turn} \land \neg\texttt{cannot\_turn} \Rightarrow \texttt{turn\_left} \end{cases} \quad (11)$$

An overview of the classes of `BDD-OIA` can be found in [84].

**Preprocessing**. `BDD-OIA` images are preprocessed following the approach described in [27]. We first used a Faster R-CNN [103] model pre-trained on MS-COCO and fine-tuned on BDD-100k, then we extract 2048-dimensional features using a pre-trained convolutional layer from [27].

## A.6 Metrics

For all models, we report the metrics averaged over ***five random seeds***.

**Label quality**. We measure the macro average $F_1$-score and Accuracy, to account for imbalance.

**Concept quality**. We compute concept quality via Accuracy and $F_1$-score. However, the order of ground-truth and predicted concepts might might differ, e.g., in `Clevr` $G_1$ might represent whether "color = red", while $C_1$ whether "color = blue". Therefore, in order to compute these metrics we have to first align them using the Hungarian matching algorithm using concept-wise Pearson correlation on the test data [64]. The algorithm computes the correlation matrix $R$ between **g** and **c**, i.e., $R_{ij} = \text{corr}(G_i, C_j)$ where $\text{corr}$ is the correlation coefficient, and then infers a permutation $\pi$ between concept values (e.g., mapping ground-truth color "red" to predicted value "blue") that maximizes the above objective. The metrics are computed by comparing the reordered ground-truth annotations with the predicted concepts.

**Concept Collapse.** To compute the ***Concept Collapse*** metric $\mathsf{Cls}(C)$ [22], we follow the procedure described in Bortolotti et al. [22], which we briefly describe here. Specifically, we extract a confusion matrix in the multilabel setting by encoding each binary concept vector (e.g., $(0, 1, 1) \mapsto 3$) into a categorical label. Let $\mathcal{C}^* \subseteq \{0, 1\}^k$ be the set of ground-truth concept vectors in the test set, and $\mathcal{C}$ be the set of predicted concept vectors. After converting both sets to categorical values, denoted $\mathcal{F}(\mathcal{C}^*)$ and $\mathcal{F}(\mathcal{C})$ respectively, we construct the confusion matrix $C \in [0, 1]^{m \times m}$, where $m = |\mathcal{F}(\mathcal{C}) \cup \mathcal{F}(\mathcal{C}^*)|$. From this matrix, we compute the number of predicted categories $p$ as:

$$p = \sum_{j=1}^{m} \mathbb{1}\{\exists i . C_{ij} > 0\} = |\mathcal{F}(\mathcal{C})| \quad (12)$$

Then, the $\mathsf{Cls}(C)$ metric is given by:

$$\mathsf{Cls}(C) = 1 - \frac{p}{m} \quad (13)$$

Where $m = 2^k$.

**Inference layer quality**. The same permutation is also used to assess knowledge quality. Specifically, we apply it to reorder the ground-truth concepts and then feed these to the learned inference layer, evaluating the accuracy and $F_1$ score of the resulting predictions.

For `MNIST-Add` and `MNIST-SumParity`, we conducted an exhaustive evaluation since the number of possible combinations to supervise is 100. The results in Table 1 and Table 2 evaluate the knowledge exhaustively, whereas in Table 4, we separately assess the knowledge of in-distribution and out-of-distribution combinations.

In `Clevr`, as discussed in Appendix A.4, there are too many combinations of concept values to possibly evaluate them all. Therefore, both supervision and evaluation are performed by randomly sampling 100 combinations of ground-truth concepts. We follow the same procedure for `BDD-OIA`, except that we sample more combinations (2000) due to the larger space of possible concept configurations.

## A.7 Architectures

For `MNIST-SumParity` and `MNIST-Add`, We employed the architectures from [22], while for DSL, DPL$^*$, and bears$^*$, we followed the architecture described in [50]. For `Clevr` instead, we adopted

the decoder presented in Table 9 and the encoder presented in Table 10. All models process different objects (e.g., digits) separately using the same backbone. For `BDD-OIA`, we adopt the architectures defined in [22].

Table 9: Encoder architecture for `Clevr`.

| INPUT | LAYER TYPE | PARAMETER | OUTPUT |
|---|---|---|---|
| $(128, 128, 3)$ | Convolution | depth=32, kernel=3, stride=1, pad=1 | ReLU |
| $(128, 128, 32)$ | MaxPool2d | kernel=2, stride=2 | |
| $(64, 64, 32)$ | Convolution | depth=64, kernel=3, stride=1, pad=1 | ReLU |
| $(64, 64, 64)$ | MaxPool2d | kernel=2, stride=2 | |
| $(64, 32, 32)$ | Convolution | depth=128, kernel=3, stride=1, pad=1 | ReLU |
| $(128, 32, 32)$ | AdaptiveAvgPool2d | out=$(1, 1)$ | |
| $(128, 1, 1)$ | Flatten | | |
| $(128)$ | Linear | dim=40, bias = True | $\mathbf{c}$ |
| $(128)$ | Linear | dim=640, bias = True | $\mu$ |
| $(128)$ | Linear | dim=640, bias = True | $\sigma$ |

Table 10: Decoder architecture for `Clevr`

| INPUT | LAYER TYPE | PARAMETER | ACTIVATION |
|---|---|---|---|
| $(255)$ | Linear | dim=32768, bias = True | |
| $(8, 8, 512)$ | ConvTranspose2d | depth=256, kernel=4, stride=2, pad=1 | |
| $(16, 16, 256)$ | BatchNorm2d | | ReLU |
| $(16, 16, 256)$ | ConvTranspose2d | depth=128, kernel=4, stride=2, pad=1 | |
| $(32, 32, 128)$ | BatchNorm2d | | ReLU |
| $(32, 32, 128)$ | ConvTranspose2d | depth=64, kernel=4, stride=2, pad=1 | |
| $(64, 64, 64)$ | BatchNorm2d | | ReLU |
| $(64, 64, 64)$ | ConvTranspose2d | depth=3, kernel=4, stride=2, pad=1 | Tanh |

**Details of `CBNM`.** For `Clevr`, we used a standard linear layer. Since this is not expressive enough for `MNIST-Add` and `MNIST-SumParity`, we replaced it with an interpretable second-degree polynomial layer, i.e., it associates weights to *combinations of two predicted digits* rather than individual digits. We also include a small entropy term, which empirically facilitates data fit when little or no concept supervision is available.

**Details of `SENN`.** `SENN` [1] are unsupervised concept-based models consisting of a neural network that produces a distribution over concepts, a neural network that assigns a score to those concepts, and a decoder. The final prediction is made by computing the dot product between the predicted concepts and their predicted scores. `SENN` by default includes a reconstruction penalty, a robustness loss, and a sparsity term.

In our implementation, we focus only on the reconstruction penalty. The robustness loss, which aims to make the relevance scores Lipschitz continuous, requires computing Jacobians, making the training of those models infeasible in terms of computational time and resources. Additionally, we did not implement the sparsity term, as it conflicts with the quality of concepts we aim to study.

**Details of `bears`*.** `bears`* combines DSL [50] and BEARS [63]. BEARS is a NeSy architectures based on deep ensembles [104], designed to improve the calibration of NeSy models in the presence of RSs and provide uncertainty estimation for concepts that may be affected by such RSs and it assumes to be given prior knowledge.

Since in our setup the inference layer is learned, we built an ensemble with one learnable inference layer as in `DSL` and multiple concept extractors. The first element of the ensemble learns both the concepts and the inference layer. Then, we freeze it and train the other concept extractors using the frozen inference layer for prediction. In this sense, `bears`* provides awareness of the reasoning shortcuts present in the knowledge learned by the first model.

## A.8 Mitigation Strategies

**Concept supervision**. When evaluating mitigation strategies, concept supervision for `MNIST-SumParity` was incrementally supplied for two more concepts at each step, following this order: 3, 4, 1, 0, 2, 8, 9, 5, 6, 7. For `Clevr`, supervision was specifically provided for sizes, shapes, materials, and colors, in this order.

**Knowledge distillation**. This was implemented using a cross-entropy on the inference layer. For `DSL` and `DPL`, ground-truth concepts were fed into the learned truth table, expecting the correct label.

Similarly, for `CBNMs`, which use a linear layer, the same approach was applied. In evaluating the mitigation strategies, we supervised all possible combinations for `MNIST-Add`, whereas for `Clevr`, we supervised a maximum of 500 randomly sampled combinations.

**Entropy maximization**. The entropy regularization term follows the implementation in [19]. The loss is defined as $1 - \frac{1}{k}\sum_{i=1}^{k} H_{m_i}[p_\theta(c_i)]$ where $p_\theta(C)$ represents the marginal distribution over the concepts, and $H_{m_i}$ is the normalized Shannon entropy over $m_i$ possible values of the distribution.

**Reconstruction**. The reconstruction penalty is the same as in [19]. In this case, the concept extractor outputs both concepts **c** and style variables **z**, i.e., it implements a distribution $p_\theta(c, z \mid x) = p_\theta(c \mid x) \cdot p_\theta(z \mid x)$. The reconstruction penalty is defined as: $-\mathbb{E}_{(c,z)\sim p_\theta(c,z|x)}\left[\log p_\psi(x \mid c, z)\right]$ where $p_\psi(x \mid c, z)$ represents the decoder network.

**Contrastive loss**. We implemented contrastive learning using the InfoNCE loss, comparing the predicted *concept distribution* of the current batch with that of its respective augmentations. Following the recommendations of Chen et al. [80], we applied the following augmentations to each batch: random cropping (20 pixels for `MNIST` and 125 pixels for `Clevr`), followed by resizing, color jittering, and Gaussian blur.

## A.9 Model Selection

**Optimization**. For `DPL*`, `bears*`, `CBNM` and `SENN` we used the Adam optimizer [105], while for `DSL` and `DPL` we achieved better results using the Madgrad optimizer [106].

**Hyperparameter search**. We performed an extensive grid search on the validation set over the following hyperparameters: (*i*) Learning rate ($\gamma$) in {1e-4, 1e-3, 1e-2}; (*ii*) Weight decay ($\omega$) in {0, 1e-4, 1e-3, 1e-2, 1e-1}; (*iii*) Reconstruction, contrastive loss, entropy loss, concept supervision loss and knowledge supervision loss weights ($w_r$, $w_c$, $w_h$, $w_{csup}$ and $w_k$, respectively) in {0.1, 1, 2, 5, 8, 10}; (*iv*) Batch size ($\nu$) in {32, 64, 128, 256, 512}. `DSL` and `DPL*` have additional hyperparameters for the truth table: $\varepsilon_{sym}$ and $\varepsilon_{rul}$ for `DSL`, and $\lambda_r$, the truth table learning rate, for `DPL*`.

All experiments were run for approximately 50 epochs for `MNIST` variants, 30 for `BDD-OIA` and 100 epochs for `Clevr` using early stopping based on validation set $F_1(Y)$ performance. The exponential decay rate $\beta$ was set to 0.99 for all experiments, as we empirically found it to provide the best performance across tasks. In all experiments, we selected $\gamma = 1e - 3$.

To answer the first research question in Section 5 on `MNIST-Add`, for `DPL` we used: $\nu = 32$ and $\omega = 0.0001$; for `DSL`: $\nu = 128$, $\omega = 0.001$, $\varepsilon_{sym} = 0.2807344052335263$, and $\varepsilon_{rul} = 0.107711951632426$; for `CBNM`: $\nu = 256$ and $\omega = 0.0001$; for `DPL*`: $\nu = 32$, $\omega = 0.01$, and $\lambda_r = 0.0001$; for `SENN`: $\nu = 64$, $\omega = 0.001$, and $w_r = 0.1$; while for `bears*` we set the diversification loss term to 0 and the entropy term to 0.1. On `MNIST-SumParity` and its biased version, for `DPL` we used: $\nu = 512$ and $\omega = 0.0001$; for `DSL`: $\nu = 128$, $\omega = 0.0001$, $\varepsilon_{sym} = 0.2807344052335263$, and $\varepsilon_{rul} = 0.107711951632426$; for `CBNM`: $\nu = 256$ and $\omega = 0.0001$; for `DPL*`: $\nu = 32$, $\omega = 0.01$, and $\lambda_r = 0.01$; for `SENN`: $\nu = 64$, $\omega = 0.001$, and $w_r = 0.1$; while for `bears*` we set the diversification loss term to 0.1 and the entropy term to 0.2. On `Clevr`, for `DPL`, `CBNM`, and `DPL*`, we used $\nu = 32$ and $\omega = 0.001$; for `DPL*`, $\lambda_r = 0.001$. On `BDD-OIA`, for `DPL` we used $\nu = 128$ and $\omega = 10^{-4}$, while for `CBNM` we used $\nu = 64$ and $\omega = 0$.

For the second research question, we performed a grid search for each mitigation strategy individually. Additionally, since our analysis focuses on optimal models, we trained multiple models and retained only those achieving optimal $F_1(Y)$ performance on the validation set. For `MNIST-SumParity`, we set $\nu = 128$, $\omega = 0.001$, and $w_{csup} = 1$ across all experiments. For specific mitigation strategies, we used $w_h = 1$ for entropy regularization, $w_r = 0.01$ for reconstruction, $w_c = 0.1$ for contrastive learning, and $w_k = 1$ for knowledge supervision. For `Clevr`, we applied the same settings: $\nu = 128$, $\omega = 0.001$, and $w_{csup} = 1$ for all experiments. The specific mitigation strategy weights were set as follows: $w_h = 1$ for entropy regularization, $w_r = 1$ for reconstruction, $w_c = 0.1$ for contrastive learning, and $w_k = 1$ for knowledge supervision. When concept supervision is applied, we follow a sequential training approach as in [3]. During the initial epochs, the model learns only the concepts, and later, it jointly optimizes both the concepts and the label, as they are challenging to learn simultaneously.

# B   Additional results

## B.1   Additional Results for Q1

 to  report additional results for **Q1** including two additional models and metrics. The additional models are: "CBNM noent", which is identical to CBNM except it does not include any entropy term; and CBNM $_{20}$ which is a regular CBNM for which $20\%$ of the concept combinations (e.g., pairs of digits in MNIST-based tasks) receive full supervision. The metrics include label accuracy $\text{Acc}_Y$, concept accuracy $\text{Acc}_C$, inference layer accuracy $\text{Acc}(\boldsymbol{\beta})$, and negative log-likelihood, all evaluated on the test set.

Table 11: **Complete results for** `MNIST-Add`.

| METHOD | $\text{Acc}_Y$ | $F_1(Y)$ | $\text{Acc}_C$ | $F_1(C)$ | $\text{Cls}(C)\,(\downarrow)$ | $F_1(\boldsymbol{\beta})$ | $\text{Acc}(\boldsymbol{\beta})$ | NLL |
|---|---|---|---|---|---|---|---|---|
| DPL | $0.98 \pm 0.01$ | $0.98 \pm 0.01$ | $0.99 \pm 0.01$ | $0.99 \pm 0.01$ | $0.01 \pm 0.01$ | $-$ | $-$ | $0.17 \pm 0.03$ |
| CBNM | $0.98 \pm 0.01$ | $0.98 \pm 0.01$ | $0.99 \pm 0.01$ | $0.99 \pm 0.01$ | $0.01 \pm 0.01$ | $1.00 \pm 0.01$ | $1.00 \pm 0.01$ | $0.11 \pm 0.01$ |
| CBNM noent | $0.98 \pm 0.01$ | $0.98 \pm 0.01$ | $0.80 \pm 0.10$ | $0.74 \pm 0.13$ | $0.08 \pm 0.04$ | $0.37 \pm 0.09$ | $0.42 \pm 0.06$ | $0.26 \pm 0.01$ |
| CBNM $_{20}$ | $0.98 \pm 0.01$ | $0.98 \pm 0.01$ | $0.99 \pm 0.01$ | $0.99 \pm 0.01$ | $0.01 \pm 0.01$ | $0.67 \pm 0.02$ | $0.59 \pm 0.02$ | $0.70 \pm 0.02$ |
| CBNM $_{20}$ noent | $0.92 \pm 0.02$ | $0.94 \pm 0.01$ | $0.60 \pm 0.09$ | $0.48 \pm 0.10$ | $0.22 \pm 0.12$ | $0.02 \pm 0.01$ | $0.12 \pm 0.02$ | $1.09 \pm 0.01$ |
| DSL | $0.96 \pm 0.02$ | $0.96 \pm 0.02$ | $0.98 \pm 0.01$ | $0.98 \pm 0.01$ | $0.01 \pm 0.01$ | $1.00 \pm 0.01$ | $1.00 \pm 0.01$ | $0.38 \pm 0.14$ |
| DPL$^*$ | $0.90 \pm 0.09$ | $0.91 \pm 0.09$ | $0.93 \pm 0.07$ | $0.92 \pm 0.08$ | $0.01 \pm 0.01$ | $0.90 \pm 0.08$ | $0.90 \pm 0.09$ | $0.02 \pm 0.01$ |
| bears$^*$ | $0.81 \pm 0.12$ | $0.76 \pm 0.15$ | $0.87 \pm 0.12$ | $0.87 \pm 0.13$ | $0.01 \pm 0.01$ | $0.67 \pm 0.14$ | $0.65 \pm 0.10$ | $0.34 \pm 0.22$ |
| SENN | $0.97 \pm 0.01$ | $0.97 \pm 0.01$ | $0.84 \pm 0.05$ | $0.80 \pm 0.07$ | $0.01 \pm 0.01$ | $0.75 \pm 0.08$ | $0.75 \pm 0.08$ | $0.01 \pm 0.01$ |

Table 12: **Complete results for** `MNIST-SumParity`.

| METHOD | $\text{Acc}_Y$ | $F_1(Y)$ | $\text{Acc}_C$ | $F_1(C)$ | $\text{Cls}(C)\,(\downarrow)$ | $F_1(\boldsymbol{\beta})$ | $\text{Acc}(\boldsymbol{\beta})$ | NLL |
|---|---|---|---|---|---|---|---|---|
| DPL | $0.99 \pm 0.01$ | $0.99 \pm 0.01$ | $0.47 \pm 0.04$ | $0.43 \pm 0.08$ | $0.36 \pm 0.15$ | $-$ | $-$ | $0.33 \pm 0.01$ |
| CBNM | $0.90 \pm 0.18$ | $0.90 \pm 0.18$ | $0.21 \pm 0.06$ | $0.09 \pm 0.03$ | $0.66 \pm 0.09$ | $0.54 \pm 0.04$ | $0.21 \pm 0.06$ | $0.52 \pm 1.04$ |
| CBNM noent | $0.98 \pm 0.01$ | $0.98 \pm 0.01$ | $0.22 \pm 0.01$ | $0.08 \pm 0.02$ | $0.72 \pm 0.13$ | $0.53 \pm 0.09$ | $0.53 \pm 0.09$ | $0.07 \pm 0.02$ |
| DSL | $0.94 \pm 0.03$ | $0.94 \pm 0.03$ | $0.20 \pm 0.01$ | $0.07 \pm 0.01$ | $0.80 \pm 0.01$ | $0.52 \pm 0.01$ | $0.51 \pm 0.02$ | $0.29 \pm 0.16$ |
| DPL$^*$ | $0.99 \pm 0.01$ | $0.99 \pm 0.01$ | $0.22 \pm 0.01$ | $0.07 \pm 0.01$ | $0.80 \pm 0.01$ | $0.50 \pm 0.05$ | $0.50 \pm 0.06$ | $0.06 \pm 0.01$ |
| bears$^*$ | $0.99 \pm 0.01$ | $0.99 \pm 0.01$ | $0.28 \pm 0.05$ | $0.30 \pm 0.03$ | $0.22 \pm 0.04$ | $0.36 \pm 0.09$ | $0.37 \pm 0.09$ | $0.04 \pm 0.01$ |
| SENN | $0.99 \pm 0.01$ | $0.99 \pm 0.01$ | $0.53 \pm 0.05$ | $0.49 \pm 0.05$ | $0.01 \pm 0.01$ | $0.53 \pm 0.06$ | $0.53 \pm 0.06$ | $0.01 \pm 0.01$ |

Table 13: **Complete Results for** `MNIST-SumParity` **biased**.

| METHOD | $\text{Acc}_Y$ | $F_1(Y)$ | $\text{Acc}_C$ | $F_1(C)$ | $\text{Cls}(C)\,(\downarrow)$ | $F_1(\boldsymbol{\beta})$ | $\text{Acc}(\boldsymbol{\beta})$ | NLL |
|---|---|---|---|---|---|---|---|---|
| DPL | $0.99 \pm 0.01$ | $0.99 \pm 0.01$ | $0.65 \pm 0.05$ | $0.59 \pm 0.06$ | $0.08 \pm 0.07$ | $-$ | $-$ | $0.06 \pm 0.01$ |
| CBNM noent | $0.99 \pm 0.01$ | $0.99 \pm 0.01$ | $0.22 \pm 0.01$ | $0.08 \pm 0.02$ | $0.76 \pm 0.06$ | $0.52 \pm 0.04$ | $0.52 \pm 0.04$ | $0.05 \pm 0.01$ |
| CBNM $_{20}$ | $0.99 \pm 0.01$ | $0.99 \pm 0.01$ | $0.21 \pm 0.01$ | $0.07 \pm 0.01$ | $0.80 \pm 0.01$ | $0.51 \pm 0.01$ | $0.52 \pm 0.01$ | $0.07 \pm 0.04$ |
| CBNM $_{20}$ noent | $0.99 \pm 0.01$ | $0.99 \pm 0.01$ | $0.22 \pm 0.01$ | $0.07 \pm 0.01$ | $0.80 \pm 0.01$ | $0.49 \pm 0.04$ | $0.48 \pm 0.04$ | $0.04 \pm 0.01$ |
| DSL | $0.94 \pm 0.03$ | $0.94 \pm 0.03$ | $0.20 \pm 0.01$ | $0.07 \pm 0.01$ | $0.80 \pm 0.01$ | $0.02 \pm 0.01$ | $0.09 \pm 0.02$ | $0.29 \pm 0.16$ |
| bears$^*$ | $0.99 \pm 0.01$ | $0.99 \pm 0.01$ | $0.20 \pm 0.01$ | $0.07 \pm 0.01$ | $0.78 \pm 0.04$ | $0.48 \pm 0.04$ | $0.49 \pm 0.04$ | $0.03 \pm 0.01$ |

## B.2   Additional Results for Q1: The `BDD-OIA` Task

**Experimental setup**. `BDD-OIA` is a challenging real-world learning task with 21 binary concepts and, unlike in `Clevr`, these cannot be decomposed into unary predicates. Moreover, each concept cannot be separately assigned to individual actions as done for DPL in [22], as this would introduce a negative bias in the learned solutions, effectively forcing the model to use a fixed concept budget per action without interference. *Due to scalability constraints, we compared only DPL and CBNM, as implementing DSL or DPL$^*$ would require allocating at least four tensors of size $2^{21}$, which exceeds our available computational resources.*

**Results**. The results for DPL and CBNM are reported in . Both competitors perform sub-optimally on `BDD-OIA`, as expected given the challenging nature of this task: DPL scores around $70\%$ $F_1(Y)$ while CBNM only $62\%$. Perhaps surprising, CBNM does not collapse concepts together and tends to use most of the concept bottleneck across the test set ($\text{Cls}(C) \approx 0.06$). Despite the lack of collapse, both architectures are far from attaining the intended semantics, with $F_1(C)$ scores of $44\%$ for DPL and $43\%$ for CBNM. Additionally, CBNM deviates from the ground-truth knowledge, achieving a low $F_1(\boldsymbol{\beta})$ of around $42\%$. The low collapse for CBNM may be due to the trained model

being far from optimal. We hypothesize that adding a sparsity penalty to the linear layer of CBNM would substantially increase concept collapse, but we leave a detailed examination to future work.

Table 14: **Results for** BDD-OIA.

| METHOD | $F_1(Y)$ (↑) | $F_1(C)$ (↑) | $\text{Cls}(C)$ (↓) | $F_1(\boldsymbol{\beta})$ (↑) |
|---|---|---|---|---|
| DPL | $0.69 \pm 0.03$ | $0.44 \pm 0.01$ | $0.88 \pm 0.01$ | — |
| CBNM | $0.62 \pm 0.03$ | $0.43 \pm 0.02$ | $0.06 \pm 0.02$ | $0.42 \pm 0.03$ |

## B.3 Additional Results for Q3

Here we show the effect of traditional mitigation strategies on MNIST-SumParity and Clevr as concept and knowledge supervision increase. As discussed in Section 5, traditional strategies have limited impact, with the exception of concept supervision.

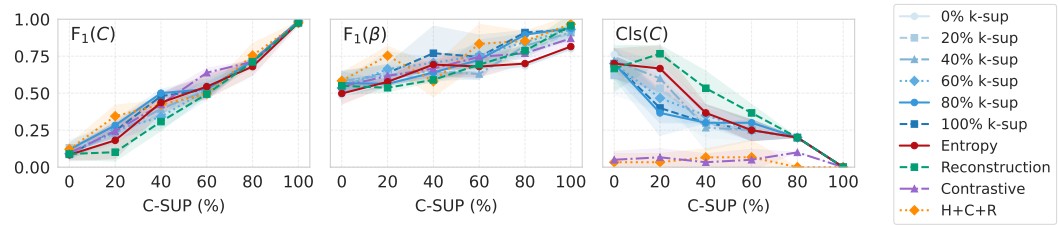

Figure 5: Fig. 3 with standard deviation over 5 seeds

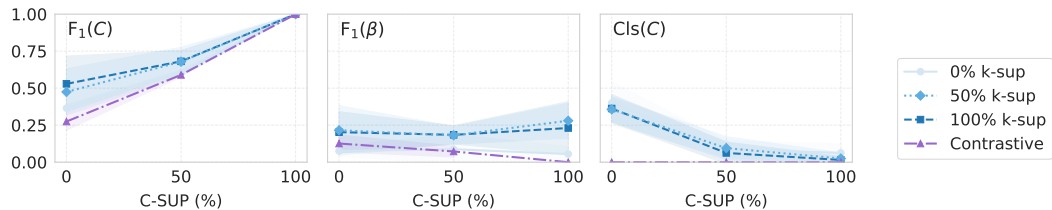

Figure 6: **Effect of standard mitigation strategies** evaluated for CBNM on Clevr, with standard deviation over 5 seeds. We evaluated only the contrastive strategy as it performed best. Since supervising all knowledge is infeasible, we observe that for sampled configurations (leading to out-of-distribution settings), the learned knowledge does not generalize.

## B.4 Concept Confusion Matrices and Learned Inference Layers

Here, we present the concept and knowledge confusion matrices for different datasets and models to show examples of joint reasoning shortcut solutions that they learn.

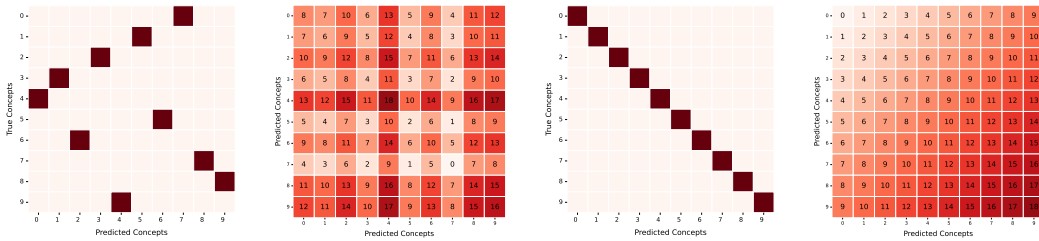

Figure 7: **Concept confusion matrices for** CBNM **on** MNIST-Add and the corresponding learned inference layer. Left two matrices: concepts and knowledge before alignment. Right two matrices: same but after alignment. The learned inference layer is visualized as a colored matrix where the numbers in the cells indicate the model's predictions.

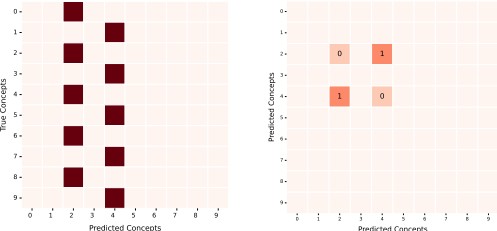

Figure 8: Concepts and inference layer learned by CBNM on MNIST-SumParity before (two left) and after (two right) the post-hoc alignment step. For concepts, we report the confusion matrix. For the inference layer, we visualize the learned operation as a truth table: numbers within cells indicate the label predicted for each combination of predicted digits.

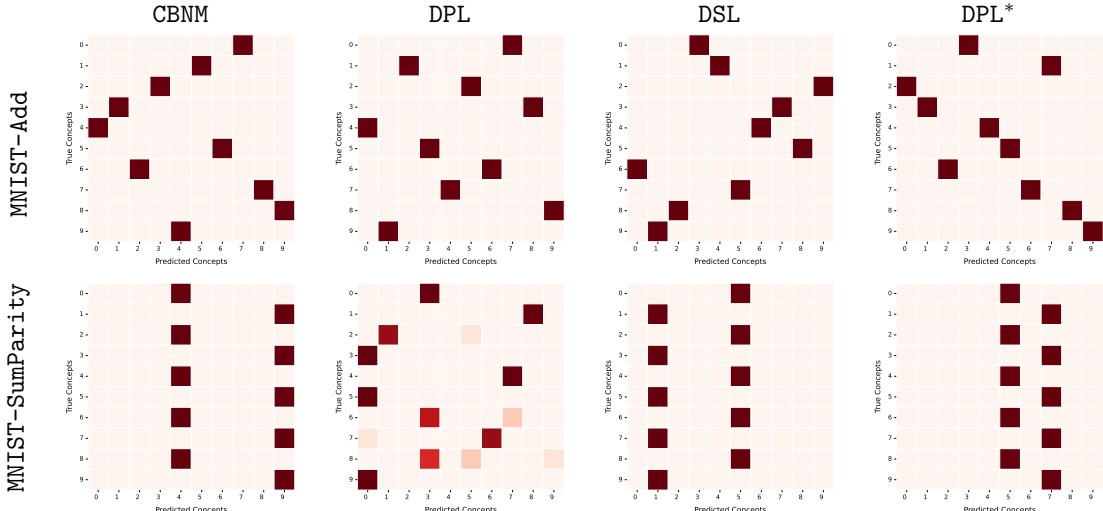

Figure 9: **Concept confusion matrices for** MNIST-SumParity. All models tend to favor $\alpha$'s that collapse concepts together, hinting at *simplicity bias*.

## B.5 Number of Reasoning Shortcuts and Joint Reasoning Shortcuts

Here, we report the approximate number of deterministic reasoning shortcuts and joint reasoning shortcuts for MNIST-Add and MNIST-SumParity. We obtained these numbers by extending the count-rss software included in rsbench [22] to the enumeration of JRSs, compatibly with Eq. (7) and Eq. (6).

We considered downsized versions of both MNIST-Add and MNIST-SumParity, with input digits restricted to the range $\{0, \dots, N\}$, and resorted to the approximate model counter approxMC [107] for obtaining approximate counts in the hardest cases. All counts assume object independence is in place (i.e., that the two input digits are processed separately), for simplicity. The situation is similar when this is not the case, except that the all counts grow proportionally.

The number of RSs and JRSs for increasing $N$ are reported in Table 15. Specifically, the #RSs column indicates the number of regular RSs obtained by fixing the prior knowledge (that is, assuming $\beta = \beta^*$) and matches the count in [19] for the same tasks. The #JRSs columns refer to the CBM case, where the inference layer is learned from data (i.e., $\beta$ might not match $\beta^*$). We report two distinct counts for JRSs: one assumes that $\beta$'s that match on all concepts actively output by $\alpha$ should be considered identical despite possible differences on "inactive" concepts that $\alpha$ predicts as constant (#JRSs non-redundant) or not (# JRSs).

As can be seen from the results, JRSs quickly outnumber regular RSs as $N$ grows, as anticipated in Section 3.1. This suggests that, as expected, learning CBMs with intended semantics by optimizing label likelihood is more challenging when prior knowledge is not given *a priori*.

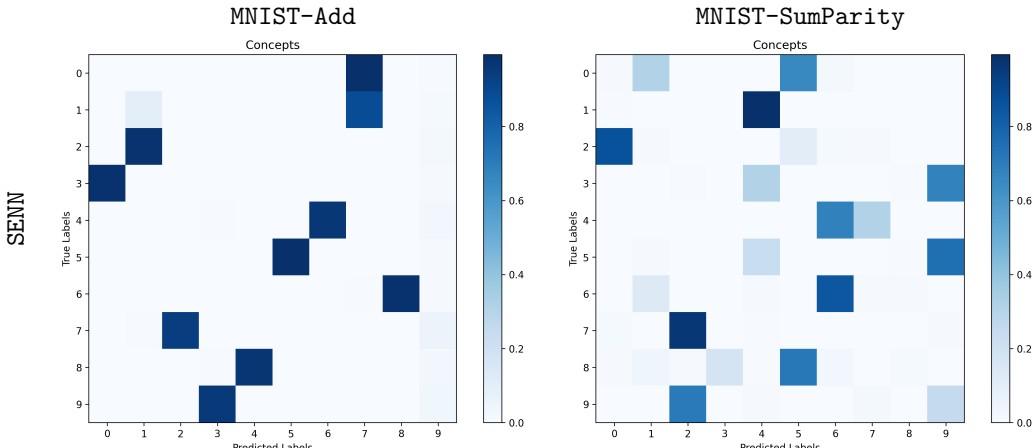

Figure 10: **Concept confusion matrices for** `SENN` **on** `MNIST-Add` **and** `MNIST-SumParity`. SENN provide *local* explanations for individual predictions, as their inference layer depends on the input and does *not* yield a global, interpretable rule set. No simplicity bias is evident in these two confusion matrices; we hypothesize that this stems from the models flexibility in modeling local rules and from the reconstruction penalty applied to the concepts during training.

Table 15: Number of RSs vs. JRSs in `MNIST-SumParity`.

| $N$ | #RSs | #JRSs | #JRSs (non-redundant) |
|---|---|---|---|
| 3 | $(1.10 \pm 0.00) \times 10^1$ | $(3.84 \pm 0.60) \times 10^2$ | $(1.20 \pm 0.00) \times 10^1$ |
| 4 | $(6.30 \pm 0.00) \times 10^1$ | $(1.11 \pm 0.05) \times 10^5$ | $(1.27 \pm 0.06) \times 10^2$ |
| 5 | $(3.70 \pm 0.15) \times 10^2$ | $(1.16 \pm 0.05) \times 10^8$ | $(1.40 \pm 0.70) \times 10^3$ |
| 6 | $(2.98 \pm 0.10) \times 10^3$ | $(4.45 \pm 0.19) \times 10^{11}$ | $(1.74 \pm 0.13) \times 10^4$ |
| 7 | $(2.56 \pm 0.26) \times 10^4$ | $(8.08 \pm 0.33) \times 10^{15}$ | $(2.61 \pm 0.14) \times 10^5$ |
| 8 | $(2.62 \pm 0.15) \times 10^5$ | $(5.39 \pm 0.25) \times 10^{20}$ | $(4.21 \pm 0.10) \times 10^6$ |

## C  Theoretical Material

In this section, we start by summarizing the prerequisite material. We begin by recalling a central Lemma proven in [19] that allows us to write the condition for likelihood-optimal models in terms of $\boldsymbol{\alpha}$. For ease of comparison, let $p(\mathbf{C} \mid \mathbf{X}) := \mathbf{f}(\mathbf{X})$ and $p(\mathbf{Y} \mid \mathbf{X}) := (\boldsymbol{\omega} \circ \mathbf{f})(\mathbf{X})$ denote the conditional distributions defined by the CBM $(\mathbf{f}, \boldsymbol{\omega}) \in \mathcal{F} \times \Omega$, and:

$$p(\mathbf{Y} \mid \mathbf{G}) := \mathbb{E}_{\mathbf{x} \sim p^*(\mathbf{X} \mid \mathbf{G})}[p(\mathbf{Y} \mid \mathbf{x})] \tag{14}$$

Also, we consider the following joint distributions:

$$p^*(\mathbf{X}, \mathbf{Y}) = p^*(\mathbf{Y} \mid \mathbf{X})p^*(\mathbf{X}), \quad p(\mathbf{X}, \mathbf{Y}) = p(\mathbf{Y} \mid \mathbf{X})p^*(\mathbf{X}) \tag{15}$$

**Lemma C.1** ([19]). *It holds that: (i) The true risk of $p$ can be upper bounded as follows:*

$$\mathbb{E}_{(\mathbf{x},\mathbf{y}) \sim p^*(\mathbf{X},\mathbf{Y})}[\log p(\mathbf{y} \mid \mathbf{x})] = -\mathsf{KL}[p^*(\mathbf{X};\mathbf{Y}) \,\|\, p(\mathbf{X},\mathbf{Y})] - H[p^*(\mathbf{X},\mathbf{Y})]] \tag{16}$$

$$\leq \mathbb{E}_{\mathbf{g} \sim p(\mathbf{G})}\big( -\mathsf{KL}[p^*(\mathbf{Y} \mid \mathbf{g}; \mathsf{K}) \,\|\, p(\mathbf{Y} \mid \mathbf{g})] - H[p^*(\mathbf{Y} \mid \mathbf{g}; \mathsf{K})]\big) \tag{17}$$

*where $\mathsf{KL}$ is the Kullback-Leibler divergence and $H$ is the Shannon entropy. Moreover, under Assumptions 3.1 and 3.2, $p(\mathbf{Y} \mid \mathbf{X}; \mathsf{K})$ is an optimum of the LHS of Eq. (17) if and only if $p(\mathbf{Y} \mid \mathbf{G})$ is an optimum of the RHS. (ii) There exists a bijection between the deterministic concept distributions $p(\mathbf{C} \mid \mathbf{X})$ that, for each $\mathbf{g} \in \mathrm{supp}(\mathbf{G})$, are constant over the support of $p^*(\mathbf{X} \mid \mathbf{g})$ apart for zero-measure subspaces $\mathcal{X}^0 \subset \mathrm{supp}(p^*(\mathbf{X} \mid \mathbf{g}))$ and the deterministic distributions of the form $p(\mathbf{C} \mid \mathbf{G})$.*

Point (i) connects the optima of the likelihood $p(\mathbf{Y} \mid \mathbf{X})$ and the optima of $p(\mathbf{Y} \mid \mathbf{G})$. This implies that, under Assumptions 3.1 and 3.2, knowing the optima of $p(\mathbf{Y} \mid \mathbf{G})$ informs us about optimal CBMs $p(\mathbf{Y} \mid \mathbf{X})$. Notice that a single $\boldsymbol{\alpha}(\mathbf{G}) = p(\mathbf{C} \mid \mathbf{G})$ might correspond to multiple

choices of $\mathbf{f} \in \mathcal{F}$. However, by (ii), if we restrict ourselves to deterministic distributions $p(\mathbf{C} \mid \mathbf{G})$, the correspondence with distributions $p(\mathbf{C} \mid \mathbf{X})$ that are almost constant in the support becomes one-to-one.

Next, we describe which distributions $p(\mathbf{Y} \mid \mathbf{G})$ are both deterministic – and thus comply with (ii) – and optimal.

**Lemma C.2** (Deterministic optima of the likelihood). *Under Assumptions 3.1 and 3.2, for all CBMs $(\mathbf{f}, \boldsymbol{\omega}) \in \mathcal{F} \times \Omega$ where, for all $\mathbf{g} \in \mathcal{G}$, $\mathbf{f} : \mathbf{x} \mapsto \mathbf{c}$ is constant over the support of $p^*(\mathbf{X} \mid \mathbf{g})$ apart for a zero-measure subspace $\mathcal{X}^0 \subset \mathsf{supp}(p^*(\mathbf{X} \mid \mathbf{g}))$, it holds that the optima of the likelihood for $p(\mathbf{Y} \mid \mathbf{G})$ are obtained when:*

$$\forall \mathbf{g} \in \mathsf{supp}(p^*(\mathbf{G})), \; (\boldsymbol{\beta} \circ \boldsymbol{\alpha})(\mathbf{g}) = \boldsymbol{\beta}^*(\mathbf{g}) \tag{18}$$

*Here, $\boldsymbol{\alpha}(\mathbf{g}) := \mathbb{E}_{\mathbf{x} \sim p^*(\mathbf{X}|\mathbf{g})}[\mathbf{f}(\mathbf{x})] \in \mathsf{Vert}(\mathcal{A})$ and $\boldsymbol{\beta}(\mathbf{c}) := \boldsymbol{\omega}(\mathbb{1}\{\mathbf{C} = \mathbf{c}\}) \in \mathsf{Vert}(\mathcal{B})$.*

*Proof.* Notice that by Lemma C.1, it holds that $\boldsymbol{\alpha} \in \mathsf{Vert}(\mathcal{A})$ is a deterministic conditional distribution. Since $\mathbf{f} : \mathbf{x} \to \mathbf{c}$ is constant over all $\mathbf{x} \sim p^*(\mathbf{X} \mid \mathbf{g})$ for a fixed $\mathbf{g} \in \mathcal{G}$, it holds that $\mathbf{f}(\mathbf{x}) = \boldsymbol{\alpha}(\mathbf{g})$ apart for some $\mathbf{x}$ in a measure-zero subspace $\mathcal{X}^0 \subset \mathsf{supp}(p^*(\mathbf{X} \mid \mathbf{g}))$. Therefore we get:

$$\mathbb{E}_{\mathbf{x} \sim p^*(\mathbf{X}|\mathbf{g})}[\boldsymbol{\omega}(\mathbf{f}(\mathbf{x}))] = \boldsymbol{\omega}(\boldsymbol{\alpha}(\mathbf{g})) = (\boldsymbol{\beta} \circ \boldsymbol{\alpha})(\mathbf{g}) \tag{19}$$

In the last line, we substitute $\boldsymbol{\omega}$ with $\boldsymbol{\beta}$ since $\boldsymbol{\alpha}$ is deterministic. For $p(\mathbf{Y} \mid \mathbf{G})$, the necessary and sufficient condition for maximum log-likelihood is:

$$\forall \mathbf{g} \in \mathsf{supp}(p^*(\mathbf{G})), \; p(\mathbf{Y} \mid \mathbf{g}) = \boldsymbol{\beta}^*(\mathbf{g}) \tag{20}$$

and substituting Eq. (19) we get:

$$\forall \mathbf{g} \in \mathsf{supp}(p^*(\mathbf{G})), \; (\boldsymbol{\beta} \circ \boldsymbol{\alpha})(\mathbf{g}) = \boldsymbol{\beta}^*(\mathbf{g}) \tag{21}$$

This proves the claim. $\square$

## C.1 Equivalence relation given by Definition 3.3

We start by proving that Definition 3.3 defines an equivalence relation.

**Proposition C.3.** *Definition 3.3 defines an equivalence relation $\sim$ between pairs $(\boldsymbol{\alpha}, \boldsymbol{\beta})$, $(\boldsymbol{\alpha}', \boldsymbol{\beta}') \in \mathcal{A} \times \mathcal{B}$, as follows: $(\boldsymbol{\alpha}, \boldsymbol{\beta}) \sim (\boldsymbol{\alpha}', \boldsymbol{\beta}')$ iff there exist a permutation $\pi : [k] \to [k]$ and $k$ element-wise invertible functions $\psi_1, \dots, \psi_k$ such that:*

$$\forall \mathbf{g} \in \mathcal{G}, \; \boldsymbol{\alpha}(\mathbf{g}) = (\mathbf{P}_\pi \circ \boldsymbol{\psi} \circ \boldsymbol{\alpha}')(\mathbf{g}) \tag{22}$$

$$\forall \mathbf{c} \in \mathcal{C}, \; \boldsymbol{\beta}(\mathbf{c}) = (\boldsymbol{\beta}' \circ \boldsymbol{\psi}^{-1} \circ \mathbf{P}_\pi^{-1})(\mathbf{c}) \tag{23}$$

*where $\mathbf{P}_\pi : \mathcal{C} \to \mathcal{C}$ is the permutation matrix induced by $\pi$ and $\boldsymbol{\psi}(\mathbf{c}) := (\psi_1(c_1), \dots, \psi_k(c_k))$.*

*Proof.* It is useful to analyze how $\mathbf{P}_\pi$ and $\boldsymbol{\psi}$ are related. Let for any $\mathbf{c} \in \mathcal{C}$:

$$(\mathbf{P}_\pi \circ \boldsymbol{\psi})(\mathbf{c}) = \mathbf{P}_\pi(\psi_1(c_1), \dots, \psi_k(c_k)) \tag{24}$$

$$= (\psi_{\pi(1)}(c_{\pi(1)}), \dots, \psi_{\pi(k)}(c_{\pi(k)})) \tag{25}$$

Using the shorthand $\boldsymbol{\psi}_\pi(\mathbf{c}) := (\psi_{\pi(1)}(c_1), \dots, \psi_{\pi(k)}(c_k))$, we have that:

$$(\mathbf{P}_\pi \circ \boldsymbol{\psi})(\mathbf{c}) = (\boldsymbol{\psi}_\pi \circ \mathbf{P}_\pi)(\mathbf{c}) \tag{26}$$

From this expression, notice that for all $\mathbf{c} \in \mathcal{C}$ it holds that:

$$(\mathbf{P}_\pi \circ \boldsymbol{\psi})(\mathbf{c}) = (\mathbf{P}_\pi \circ \boldsymbol{\psi} \circ \mathbf{P}_\pi^{-1} \circ \mathbf{P}_\pi)(\mathbf{c}) \tag{27}$$

$$(\boldsymbol{\psi}_\pi \circ \mathbf{P}_\pi)(\mathbf{c}) = \big((\mathbf{P}_\pi \circ \boldsymbol{\psi} \circ \mathbf{P}_\pi^{-1}) \circ \mathbf{P}_\pi\big)(\mathbf{c}) \tag{28}$$

so that we can equivalently write $\boldsymbol{\psi}_\pi := \mathbf{P}_\pi \circ \boldsymbol{\psi} \circ \mathbf{P}_\pi^{-1}$.

**Reflexivity**. This follows by choosing $\mathbf{P}_\pi = \mathrm{id}$ and similarly $\psi_i = \mathrm{id}$ for all $i \in [k]$.

**Symmetry**. We have to prove that $(\boldsymbol{\alpha}, \boldsymbol{\beta}) \sim (\boldsymbol{\alpha}', \boldsymbol{\beta}') \implies (\boldsymbol{\alpha}', \boldsymbol{\beta}') \sim (\boldsymbol{\alpha}, \boldsymbol{\beta})$. Since $(\boldsymbol{\alpha}, \boldsymbol{\beta}) \sim (\boldsymbol{\alpha}', \boldsymbol{\beta}')$, we can write $\boldsymbol{\alpha}$ in terms of $\boldsymbol{\alpha}'$ as follows:

$$\boldsymbol{\alpha}(\mathbf{g}) = (\mathbf{P}_\pi \circ \boldsymbol{\psi} \circ \boldsymbol{\alpha}')(\mathbf{g}) \tag{29}$$

$$= (\boldsymbol{\psi}_\pi \circ \mathbf{P}_\pi \circ \boldsymbol{\alpha}')(\mathbf{g}) \tag{30}$$

where in the last step we used Eq. (26). By first inverting $\psi_\pi$ and then $\mathbf{P}_\pi$, we obtain that:

$$\alpha'(\mathbf{g}) = (\mathbf{P}_\pi^{-1} \circ \psi_\pi^{-1} \circ \alpha)(\mathbf{g}) \tag{31}$$

The inverses exist by construction/definition. Showing the symmetry of $\alpha$. With similar steps, we can show that a similar relation also holds for $\beta'$, that is for all $\mathbf{c} \in \mathcal{C}$:

$$\beta'(\mathbf{c}) = (\beta \circ \psi_\pi \circ \mathbf{P}_\pi)(\mathbf{c}) \tag{32}$$

**Transitivity**. We have to prove that if $(\alpha, \beta) \sim (\alpha', \beta')$ and $(\alpha', \beta') \sim (\alpha^\dagger, \beta^\dagger)$ then also $(\alpha, \beta) \sim (\alpha^\dagger, \beta^\dagger)$. We start from the expression of $\alpha$ and $\alpha'$, where we have that $\forall \mathbf{g} \in \mathcal{G}$:

$$\alpha(\mathbf{g}) = (\mathbf{P}_\pi \circ \psi \circ \alpha')(\mathbf{g}) \tag{33}$$

$$\alpha'(\mathbf{g}) = (\mathbf{P}_{\pi'} \circ \psi' \circ \alpha^\dagger)(\mathbf{g}) \tag{34}$$

We proceed by substituting the expression of $\alpha'$ in $\alpha$ to obtain:

$$\alpha(\mathbf{g}) = (\mathbf{P}_\pi \circ \psi \circ \mathbf{P}_{\pi'} \circ \psi' \circ \alpha^\dagger)(\mathbf{g}) \tag{35}$$

$$= (\mathbf{P}_\pi \circ \mathbf{P}_{\pi'} \circ \mathbf{P}_{\pi'}^{-1} \circ \psi \circ \mathbf{P}_{\pi'} \circ \psi' \circ \alpha^\dagger)(\mathbf{g})$$
$$\text{(Composing } \mathbf{P}_\pi \text{ with the identity } \mathbf{P}_{\pi'} \circ \mathbf{P}_{\pi'}^{-1})$$

$$= (\mathbf{P}_\pi \circ \mathbf{P}_{\pi'} \circ \psi_{\pi'^{-1}} \circ \psi' \circ \alpha^\dagger)(\mathbf{g}) \qquad \text{(Using that } \psi_{\pi'^{-1}} := \mathbf{P}_{\pi'}^{-1} \circ \psi \circ \mathbf{P}_{\pi'}.)$$

$$= (\mathbf{P}_{\pi^\dagger} \circ \psi^\dagger \circ \alpha^\dagger)(\mathbf{g}) \tag{36}$$

where we defined $\mathbf{P}_{\pi^\dagger} := \mathbf{P}_\pi \circ \mathbf{P}_{\pi'}$ and $\psi^\dagger := \psi_{\pi'^{-1}} \circ \psi'$. With similar steps, we obtain that $\beta$ can be related to $\beta^\dagger$ as:

$$\beta(\mathbf{c}) = (\beta^\dagger \circ {\psi^\dagger}^{-1} \circ \mathbf{P}_{\pi^\dagger}^{-1})(\mathbf{c}) \tag{37}$$

for all $\mathbf{c} \in \mathcal{C}$. This shows the equivalence relation of Definition 3.3. $\qquad \square$

We now prove how solutions with intended semantics (Definition 3.3) relate to the optima of the log-likelihood:

**Lemma C.4** (Intended semantics entails optimal models). *If a pair $(\alpha, \beta) \in \mathcal{A} \times \mathcal{B}$ possesses the intended semantics (Definition 3.3), it holds that:*

$$\forall \mathbf{g} \in \mathcal{G}, \ (\beta \circ \alpha)(\mathbf{g}) = \beta^*(\mathbf{g}) \tag{38}$$

*Proof.* By Definition 3.3, we can write:

$$\alpha(\mathbf{g}) = (\mathbf{P}_\pi \circ \psi \circ \mathrm{id})(\mathbf{g}) \tag{39}$$

$$\beta(\mathbf{c}) = (\beta^* \circ \psi^{-1} \circ \mathbf{P}_\pi^{-1})(\mathbf{c}) \tag{40}$$

Combining Eqs. (39) and (40), we get, for all $\mathbf{g} \in \mathcal{G}$, that:

$$(\beta \circ \alpha)(\mathbf{g}) = (\beta^* \circ \mathbf{P}_\pi^{-1} \circ \psi^{-1} \circ \psi \circ \mathbf{P}_\pi)(\mathbf{g}) \tag{41}$$

$$= (\beta^* \circ \mathbf{P}_\pi^{-1} \circ \mathbf{P}_\pi)(\mathbf{g}) \tag{42}$$

$$= \beta^*(\mathbf{g}) \tag{43}$$

yielding the claim. $\qquad \square$

## C.2 Proof of Theorem 3.6 and Corollary 3.7

**Theorem C.5.** *Let $s\mathcal{G} := \bigcup_{i=1}^k \{|\mathcal{G}_i|\}$ be the set of cardinalities of each concept $G_i \subseteq \mathbf{G}$ and $ms\mathcal{G}$ denote the multi-set $ms\mathcal{G} := \{(|\mathcal{G}_i|, m(|\mathcal{G}_i|)), \ i \in [k]\}$, where $m(\bullet)$ denotes the multiplicity of each element of $s\mathcal{G}$. Under Assumptions 3.1 and 3.2, the number of deterministic JRSs $(\alpha, \beta) \in \mathsf{Vert}(\mathcal{A}) \times \mathsf{Vert}(\mathcal{B})$ amounts to:*

$$\sum_{(\alpha, \beta) \in \mathsf{Vert}(\mathcal{A}) \times \mathsf{Vert}(\mathcal{B})} \mathbb{1}\{\bigwedge_{\mathbf{g} \in \mathrm{supp}(\mathbf{G})} (\beta \circ \alpha)(\mathbf{g}) = \beta^*(\mathbf{g})\} - C[\mathcal{G}] \tag{44}$$

*where $C[\mathcal{G}]$ is the total number of pairs with the intended semantics, given by $C[\mathcal{G}] := \prod_{\xi \in s\mathcal{G}} m(\xi)! \times \prod_{i=1}^k |\mathcal{G}_i|!$.*

*Proof.* Since we are considering pairs $(\boldsymbol{\alpha}, \boldsymbol{\beta}) \in \mathsf{Vert}(\mathcal{A}) \times \mathsf{Vert}(\mathcal{B})$, $\boldsymbol{\alpha}$ defines a (deterministic) function $\boldsymbol{\alpha} : \mathcal{G} \to \mathcal{C}$ and that, similarly, $\boldsymbol{\beta}$ defines a (deterministic) function $\boldsymbol{\beta} : \mathcal{C} \to \mathcal{Y}$. In this case, $\boldsymbol{\beta} \circ \boldsymbol{\alpha} : \mathcal{G} \to \mathcal{Y}$ is a map from ground-truth concepts to labels. We start by Definition 3.3 and consider the pairs that attain maximum likelihood by Lemma C.2:

$$\forall \mathbf{g} \in \mathsf{supp}(p^*(\mathbf{G})), \ (\boldsymbol{\beta} \circ \boldsymbol{\alpha})(\mathbf{g}) = \boldsymbol{\beta}^*(\mathbf{g}) \tag{45}$$

Since $\boldsymbol{\alpha} \in \mathsf{Vert}(\mathcal{A})$ is deterministic, we can replace $\boldsymbol{\omega}$ with $\boldsymbol{\beta}$. Since both $\mathsf{Vert}(\mathcal{A})$ and $\mathsf{Vert}(\mathcal{B})$ are countable, we can count the number of pairs that attain maximum likelihood as follows:

$$\sum_{(\boldsymbol{\alpha}, \boldsymbol{\beta}) \in \mathsf{Vert}(\mathcal{A}) \times \mathsf{Vert}(\mathcal{B})} \mathbb{1}\{ \bigwedge_{\mathbf{g} \in \mathsf{supp}(p^*(\mathbf{G}))} (\boldsymbol{\beta} \circ \boldsymbol{\alpha})(\mathbf{g}) = \boldsymbol{\beta}^*(\mathbf{g})\} \tag{46}$$

where, for each pair $(\boldsymbol{\alpha}, \boldsymbol{\beta}) \in \mathsf{Vert}(\mathcal{A}) \times \mathsf{Vert}(\mathcal{B})$ the sum increases by one if the condition of Eq. (45) is satisfied.

Among these optimal pairs, some possess the intended semantics and therefore do not count as JRSs. By Lemma C.4, these correspond to all possible permutations of the concepts combined with all possible element-wise invertible transformations of the concept values.

We begin by evaluating the number of possible element-wise invertible transformations. The $i$-th concept can attain $|\mathcal{G}_i|$ values. The overall number of possible invertible transformations is then given by the number of possible permutations, resulting in a total of $|\mathcal{G}_i|!$ maps. Hence, the number of element-wise invertible transformations is:

$$\prod_{i=1}^{k} |\mathcal{G}_i|! \tag{47}$$

Next, we consider what permutation of the concepts are possible. To this end, consider the set and the multi-set comprising all different $\mathcal{G}_i$ cardinalities given by:

$$s\mathcal{G} := \bigcup_{i=1}^{k} \{|\mathcal{G}_i|\}, \quad ms\mathcal{G} := \{(|\mathcal{G}_i|, m(|\mathcal{G}_i|) : \ i \in [k]\} \tag{48}$$

where $m(\bullet)$ counts how many repetitions are present. When different $\mathcal{G}_i$ and $\mathcal{G}_j$ have the same cardinality, we can permute $G_i$ and $G_j$ without compromising optimality, as there always exists a $\boldsymbol{\beta}$ that inverts the permutation and thus provides the same output. Thankfully, this shows in the multiplicity $m(\cdot)$ and we can account for this. Therefore, for each element $\xi \in s\mathcal{G}$ we have that the total number of permutations of concepts amount to:

$$\mathsf{perm}(\mathcal{G}) := \prod_{\xi \in s\mathcal{G}} m(\xi)! \tag{49}$$

Putting everything together, we obtain that the number of optimal pairs possessing the intended semantics is $\mathsf{perm}(\mathcal{G}) \times \prod_{i=1}^{k} |\mathcal{G}_i|!$, meaning that the total number of deterministic JRSs is:

$$\sum_{(\boldsymbol{\alpha}, \boldsymbol{\beta}) \in \mathsf{Vert}(\mathcal{A}) \times \mathsf{Vert}(\mathcal{B})} \mathbb{1}\{ \bigwedge_{\mathbf{g} \in \mathsf{supp}(p^*(\mathbf{G}))} (\boldsymbol{\beta} \circ \boldsymbol{\alpha})(\mathbf{g}) = \boldsymbol{\beta}^*(\mathbf{g})\} - \mathsf{perm}(\mathcal{G}) \times \prod_{i=1}^{k} |\mathcal{G}_i|! \tag{50}$$

yielding the claim. $\qquad\square$

**Corollary 3.7.** *Consider a fixed $\boldsymbol{\beta}^* \in \mathsf{Vert}(\mathcal{B})$. Under Assumptions 3.1 and 3.2, the number of deterministic JRSs $(\boldsymbol{\alpha}, \boldsymbol{\beta}^*) \in \mathsf{Vert}(\mathcal{A}) \times \mathsf{Vert}(\mathcal{B})$ is:*

$$\sum_{\boldsymbol{\alpha} \in \mathsf{Vert}(\mathcal{A})} \mathbb{1}\{\bigwedge_{\mathbf{g} \in \mathsf{supp}(\mathbf{G})} (\boldsymbol{\beta}^* \circ \boldsymbol{\alpha})(\mathbf{g}) = \boldsymbol{\beta}^*(\mathbf{g})\} - 1 \tag{7}$$

*This matches the count for deterministic RSs in [19].*

*Proof.* The proof follows immediately by replacing $\mathsf{Vert}(\mathcal{B})$ with $\{\boldsymbol{\beta}^*\}$, allowing only for the ground-truth inference function. In this context, by following similar steps of Theorem 3.6, we arrive at a similar count to Eq. (46), that is:

$$\sum_{(\boldsymbol{\alpha}, \boldsymbol{\beta}) \in \mathsf{Vert}(\mathcal{A}) \times \{\boldsymbol{\beta}^*\}} \mathbb{1}\{ \bigwedge_{\mathbf{g} \in \mathsf{supp}(p^*(\mathbf{G}))} (\boldsymbol{\beta} \circ \boldsymbol{\alpha})(\mathbf{g}) = \boldsymbol{\beta}^*(\mathbf{g})\} \tag{51}$$

In the context of regular RSs, the only pair possessing the intended semantics is $(\mathsf{id}, \boldsymbol{\beta}^*)$, which has to subtracted from Eq. (51). This proves the claim. $\qquad\square$

## C.3 Proof of Theorem 3.9

**Theorem 3.9** (Identifiability). *Under Assumptions 3.1 and 3.2, **if** the number of deterministic JRSs (Eq. (6)) is zero **then** every CBM $(\mathbf{f}, \boldsymbol{\omega}) \in \mathcal{F} \times \Omega$ satisfying Assumption 3.8 that attains maximum log-likelihood (Eq. (4)) possesses the intended semantics (Definition 3.3). That is, the pair $(\boldsymbol{\alpha}, \boldsymbol{\beta}) \in \mathcal{A} \times \mathcal{B}$ entailed by each such CBM is equivalent to the ground-truth pair $(\mathrm{id}, \boldsymbol{\beta}^*)$, i.e., $(\boldsymbol{\alpha}, \boldsymbol{\beta}) \sim (\mathrm{id}, \boldsymbol{\beta}^*)$.*

*Proof.* Consider a pair $(\boldsymbol{\alpha}, \boldsymbol{\beta}) \in \mathcal{A} \times \mathcal{B}$, where both $\boldsymbol{\alpha}$ and $\boldsymbol{\beta}$ can be non-deterministic conditional probability distributions.

**Step** (i). We begin by first recalling Lemma C.1. Under Assumptions 3.1 and 3.2, a CBM $(\mathbf{f}, \boldsymbol{\omega}) \in \mathcal{F} \times \Omega$ is optimal, i.e., it attains maximum likelihood, if it holds that:

$$\forall \mathbf{g} \in \mathsf{supp}(p^*(\mathbf{G})), \ \mathbb{E}_{\mathbf{x} \sim p^*(\mathbf{X}|\mathbf{g})}[(\boldsymbol{\omega} \circ \mathbf{f})(\mathbf{x})] = \boldsymbol{\beta}^*(\mathbf{g}) \tag{52}$$

Notice that, according to Assumptions 3.1 and 3.2, it also holds that the labels must be predicted with probability one to attain maximum likelihood, that is:

$$\forall \mathbf{g} \in \mathcal{G}, \ \max \boldsymbol{\beta}^*(\mathbf{g}) = 1 \tag{53}$$

Now consider a pair $(\boldsymbol{\alpha}, \boldsymbol{\beta}) \in \mathcal{A} \times \mathcal{B}$. By Lemma C.1 (i), it holds that to obtain maximum likelihood, the learned knowledge $\boldsymbol{\beta} : \mathcal{C} \to \Delta_{\mathcal{Y}}$ must be deterministic:

$$\max \boldsymbol{\beta}(\mathbf{c}) = 1 \tag{54}$$

which implies that, necessarily, for $\boldsymbol{\beta}$ to be optimal the space $\mathcal{B}$ restricts to the set $\mathsf{Vert}(\mathcal{B})$, so any learned knowledge $\boldsymbol{\beta}$ must belong to $\mathsf{Vert}(\mathcal{B})$. This shows that optimal pairs $(\boldsymbol{\alpha}, \boldsymbol{\beta})$ belong to $\mathcal{A} \times \mathsf{Vert}(\mathcal{B})$.

**Step** (ii). Next, we make use of Assumption 3.8 and consider only inference functions $\boldsymbol{\omega}$ that satisfy it. We start by considering a pair $(\mathbf{a}', \mathbf{b}') \in \mathsf{Vert}(\mathcal{A}) \times \mathsf{Vert}(\mathcal{B})$ that attain maximum likelihood, that is, according to Theorem 3.6:

$$\forall \mathbf{g} \in \mathsf{supp}(p^*(\mathbf{G})), \ (\mathbf{b}' \circ \mathbf{a}')(\mathbf{g}) = \boldsymbol{\beta}^*(\mathbf{g}) \tag{55}$$

Now, given $\mathbf{b}' \in \mathsf{Vert}(\mathcal{B})$, we have to prove that no non-deterministic $\boldsymbol{\alpha} \in \mathcal{A} \setminus \mathsf{Vert}(\mathcal{A})$ can attain maximum likelihood for any of the $\boldsymbol{\omega}' \in \Omega$ that correspond to $\mathbf{b}'$, that is, such that $\boldsymbol{\omega}'(\mathbb{1}\{\mathbf{C} = \mathbf{c}\}) = \mathbf{b}'(\mathbf{c})$ for all $\mathbf{c} \in \mathcal{C}$. Since $\mathcal{A}$ is a simplex, we can always construct a non-deterministic conditional probability distribution $\boldsymbol{\alpha} \in \mathcal{A}$ from a convex combination of the vertices $\mathbf{a}_i \in \mathsf{Vert}(\mathcal{A})$, i.e., for all $\mathbf{g} \in \mathcal{G}$ it holds:

$$\boldsymbol{\alpha}(\mathbf{g}) = \sum_{\boldsymbol{\alpha}_i \in \mathsf{Vert}(\mathcal{A})} \lambda_i \mathbf{a}_i(\mathbf{g}) \tag{56}$$

Consider another $\mathbf{a}'' \neq \mathbf{a}' \in \mathsf{Vert}(\mathcal{A})$ and consider an arbitrary convex combination that defines a non-deterministic $\boldsymbol{\alpha} \in \mathcal{A}$:

$$\forall \mathbf{g} \in \mathcal{G}, \ \boldsymbol{\alpha}(\mathbf{g}) := \lambda \mathbf{a}'(\mathbf{g}) + (1 - \lambda) \mathbf{a}''(\mathbf{g}) \tag{57}$$

where $\lambda \in (0, 1)$. When the deterministic JRSs count (Eq. (6)) equals to zero, we know that by Theorem 3.6 only $(\mathbf{a}', \mathbf{b}')$ attains maximum likelihood, whereas $(\mathbf{a}'', \mathbf{b}')$ is not optimal. Therefore, there exists at least one $\hat{\mathbf{g}} \in \mathsf{supp}(p^*(\mathbf{G}))$ such that

$$(\boldsymbol{\beta}' \circ \mathbf{a}')(\hat{\mathbf{g}}) \neq (\boldsymbol{\beta}' \circ \mathbf{a}'')(\hat{\mathbf{g}}) \tag{58}$$

We now have to look at the form of $\mathbf{f}$ that can induce such a non-deterministic $\boldsymbol{\alpha} = \lambda \mathbf{a}' + (1 - \lambda) \mathbf{a}''$. Recalling, the definition of $\boldsymbol{\alpha}$ (Eq. (2)) we have that:

$$\boldsymbol{\alpha}(\mathbf{g}) = \mathbb{E}_{\mathbf{x} \sim p^*(\mathbf{X}|\mathbf{g})}[\mathbf{f}(\mathbf{x})] \tag{59}$$
$$= \lambda \mathbf{a}'(\mathbf{g}) + (1 - \lambda) \mathbf{a}''(\mathbf{g}) \tag{60}$$

For this $\boldsymbol{\alpha}$, there are two possible kinds of $\mathbf{f} \in \mathcal{F}$ that can express it.

(1) In one case, we can have $\mathbf{f} : \mathbb{R}^n \to \mathsf{Vert}(\Delta_{\mathcal{C}})$ – mapping inputs to "hard" distributions over concepts. Since $\boldsymbol{\alpha}$ is not a "hard" distribution (provided $\lambda \in (0, 1)$), $\mathbf{f}(\mathbf{x})$ must be equal to $\mathbf{a}'(\mathbf{g})$ for a fraction $\lambda$ of the examples $\mathbf{x} \in \mathsf{supp}(p^*(\mathbf{X} \mid \mathbf{g}))$ and to $\mathbf{a}''(\mathbf{g})$ for a fraction of $1 - \lambda$. In that case, it holds that there exist a subspace of non-vanishing measure $\mathcal{X}'' \subset \mathsf{supp}(p^*(\mathbf{X} \mid \hat{\mathbf{g}}))$ such that

$\mathbf{f}(\mathbf{x}) = \mathbf{a}''(\hat{\mathbf{g}})$. Therefore, for all $\mathbf{x} \in \mathcal{X}''$ we have that $(\mathbf{b}' \circ \mathbf{a}'')(\mathbf{x}) = (\boldsymbol{\omega} \circ \mathbf{a}'')(\hat{\mathbf{g}}) \neq \boldsymbol{\beta}^*(\hat{\mathbf{g}})$, and such an $\mathbf{f}$ is suboptimal.

(2) The remaining option is that $\mathbf{f} : \mathbb{R}^n \to \Delta_\mathcal{C} \setminus \mathsf{Vert}(\Delta_\mathcal{C})$. In light of this, we rewrite $\mathbf{f}$ as follows:

$$\forall \mathbf{x} \in \mathsf{supp}(p^*(\mathbf{X} \mid \hat{\mathbf{g}})), \ \mathbf{f}(\mathbf{x}) = \sum_{\mathbf{c} \in \mathcal{C}} p(\mathbf{c} \mid \mathbf{x}) \mathbb{1}\{\mathbf{C} = \mathbf{c}\} \tag{61}$$

where $p(\mathbf{c} \mid \mathbf{x}) := \mathbf{f}(\mathbf{x})_\mathbf{c}$. Let $\hat{\mathbf{c}}' := \mathbf{a}'(\hat{\mathbf{g}})$ and $\hat{\mathbf{c}}'' := \mathbf{a}''(\hat{\mathbf{g}})$. But $\boldsymbol{\alpha}$ is a convex combination of $\mathbf{a}'$ and $\mathbf{a}''$, hence the function $\mathbf{f}$ must attribute non-zero probability mass only to the concept vectors $\hat{\mathbf{c}}'$ and $\hat{\mathbf{c}}''$. Hence, we rewrite it as:

$$\forall \mathbf{x} \in \mathcal{X} \subseteq \mathsf{supp}(p^*(\mathbf{X} \mid \hat{\mathbf{g}})), \ \mathbf{f}(\mathbf{x}) = p(\hat{\mathbf{c}}' \mid \mathbf{x}) \mathbb{1}\{\mathbf{C} = \hat{\mathbf{c}}'\} + p(\hat{\mathbf{c}}'' \mid \mathbf{x}) \mathbb{1}\{\mathbf{C} = \hat{\mathbf{c}}''\} \tag{62}$$

where $\mathcal{X}$ has measure one. Notice that in this case it holds $\mathrm{argmax}_{\mathbf{y} \in \mathcal{Y}} \mathbf{b}'(\hat{\mathbf{c}}')_\mathbf{y} \neq \mathrm{argmax}_{\mathbf{y} \in \mathcal{Y}} \mathbf{b}'(\hat{\mathbf{c}}'')_\mathbf{y}$. By Assumption 3.8 we have that, for any choice of $\lambda \in (0, 1)$ it holds that for all $\mathbf{x} \in \mathcal{X}$:

$$\max \boldsymbol{\omega}(p(\mathbf{c}' \mid \mathbf{x}) \mathbb{1}\{\mathbf{C} = \mathbf{c}'\} + p(\mathbf{c}'' \mid \mathbf{x}) \mathbb{1}\{\mathbf{C} = \mathbf{c}''\}) < \max(\underset{\mathbf{y} \in \mathcal{Y}}{\mathrm{argmax}}\, \mathbf{b}'(\hat{\mathbf{c}}')_\mathbf{y}, \underset{\mathbf{y} \in \mathcal{Y}}{\mathrm{argmax}}\, \mathbf{b}'(\hat{\mathbf{c}}'')_\mathbf{y})$$
$$\text{(Using the condition from Assumption 3.8)}$$
$$= 1 \tag{63}$$

where the last step in Eq. (63) follows from the fact that $\mathbf{b}' \in \mathsf{Vert}(\mathcal{B})$, so giving point-mass conditional probability distributions.

(1) and (2) together show that any $\boldsymbol{\alpha} \in \mathcal{A} \setminus \mathsf{Vert}(\mathcal{A})$ constructed as a convex combination of $\mathbf{a}'$ and an $\mathbf{a}'' \neq \mathbf{a}'$ cannot attain maximum likelihood. Hence, the optimal pairs restrict to $(\boldsymbol{\alpha}, \boldsymbol{\beta}) \in \mathsf{Vert}(\mathcal{A}) \times \mathsf{Vert}(\mathcal{B})$ and since the count of JRSs (Eq. (6)) is zero, all $(\boldsymbol{\alpha}, \boldsymbol{\beta}) \in \mathcal{A} \times \mathcal{B}$ that are optimal also possess the intended semantics. This concludes the proof. $\qquad \square$

## D  Impact of mitigation strategies on the deterministic JRSs count

We focus on how the count can be updated for some mitigation strategies that we accounted for in Section 4. We consider those strategies that gives a constraint for the Eq. (6), namely: ***multi-task learning***, ***concept supervision***, ***knowledge distillation***, and ***reconstruction***.

### D.1  Multi-task learning

In the following, we use $\tau$ to indicate an additional learning task with corresponding. Suppose that for a subset $\mathcal{G}^\tau \subseteq \mathsf{supp}(p^*(\mathbf{G}))$, input examples are complemented with additional labels $\mathbf{y}^\tau$, obtained by applying an inference layer $\boldsymbol{\beta}_{\mathsf{K}^\tau}(\mathbf{g})$. Here, $\mathsf{K}^\tau$ is the conjunction of the prior knowledges for the original and additional task, and it yields a set $\mathcal{Y}^\tau$ of additional labels when applied to the examples in $\mathcal{G}^\tau$. To understand how multi-task learning impacts the count of JRSs, we have to consider those $\boldsymbol{\beta} : \mathcal{C} \to \mathcal{Y} \times \mathcal{Y}^\tau$, where a component $\boldsymbol{\beta}_\mathcal{Y}$ maps to the original task labels $\mathbf{Y}$ and the other component $\boldsymbol{\beta}_{\mathcal{Y}^\tau}$ maps to the augmented labels $\mathbf{Y}^\tau$. Let $\mathcal{B}^* := \mathcal{B} \times \mathcal{B}^\tau$ the space where such functions are defined. As obtained in [19], the constraint for a pair $(\boldsymbol{\alpha}, \boldsymbol{\beta}) \in \mathsf{Vert}(\mathcal{A}) \times \mathsf{Vert}(\mathcal{B}^*)$ can be written as:

$$\mathbb{1}\{\bigwedge_{\mathbf{g} \in \mathcal{G}^\tau} (\boldsymbol{\beta}_{\mathcal{Y}^\tau} \circ \boldsymbol{\alpha})(\mathbf{g}) = \boldsymbol{\beta}_{\mathsf{K}^\tau}(\mathbf{g})\} \tag{64}$$

This term can be readily combined with the one appearing in Theorem 3.6. Doing so, reduces the number of possible maps $\boldsymbol{\alpha} \in \mathsf{Vert}(\mathcal{A})$ that successfully allow a function $\boldsymbol{\beta} \in \mathcal{B}^*$ to predict both $(\mathbf{y}, \mathbf{y}^\tau)$ consistently. Notice that, in the limit where the augmented knowledge comprehends enough additional labels and the support covers the whole , the only admitted solutions consist of:

$$(\boldsymbol{\beta}_{\mathcal{Y}^\tau} \circ \mathrm{id})(\mathbf{g}) = (\underbrace{\boldsymbol{\beta}_{\mathcal{Y}^\tau} \circ \boldsymbol{\phi}^{-1}}_{=:\boldsymbol{\beta}_{\mathcal{Y}^\tau}} \circ \underbrace{\boldsymbol{\phi}}_{=:\boldsymbol{\alpha}})(\mathbf{g}) \tag{65}$$

for all $\mathbf{g} \in \mathcal{G}$, where $\boldsymbol{\phi} : \mathcal{G} \to \mathcal{C}$ is an invertible function. This shows that $\boldsymbol{\alpha}$ can be only a bijection from $\mathcal{G}$ to $\mathcal{C}$. Provided all the concept vectors $\mathbf{g} \in \mathcal{G}$, the number of such possible $\boldsymbol{\alpha}$'s amounts to the number of possible permutation of the $|\mathcal{G}|$ elements, that is $|\mathcal{G}|!$.

## D.2 Concept supervision

The intuition is the same as in [19]. Assume that we supply concept supervision for a subset of concepts $\mathbf{G}_\mathcal{I} \subseteq \mathbf{G}$, with $\mathcal{I} \subseteq [k]$ and pair the regular log-likelihood objective over the labels with a log-likelihood over the concepts, i.e., $-\sum_{i \in \mathcal{I}} \log p_\theta(C_i = g_i \mid \mathbf{x})$. Let $\mathcal{G}^C \subseteq \operatorname{supp}(p^*(\mathbf{G}))$ be the subset of values that receive supervision. The only deterministic $\boldsymbol{\alpha}$'s that attain maximum likelihood on the concepts are those that match said supervision, i.e., that satisfy the constraint:

$$\mathbb{1}\{\textstyle\bigwedge_{\mathbf{g} \in \mathcal{G}^C} \bigwedge_{i \in \mathcal{I}} \alpha_i(\mathbf{g}) = g_i\} \tag{66}$$

This can be immediately used to obtain an updated (and smaller) JRS count. Note that if $\mathcal{I} = [k]$ and $\mathcal{G}^C = \mathcal{G}$, then the only suitable map is $\boldsymbol{\alpha} \equiv \operatorname{id}$. Naturally, this requires dense annotations for all possible inputs, which may be prohibitive.

## D.3 Knowledge distillation

We consider the case where samples $(\mathbf{g}, \boldsymbol{\beta}^*(\mathbf{g}))$ are used to distill the ground-truth knowledge. Since by Assumption 3.2 the ground-truth inference layer is deterministic, we consider the following objective for distillation. Letting $\mathcal{G}^K \subseteq \operatorname{supp}(p^*(\mathbf{G}))$ be the subset of supervised values, we augment the original log-likelihood objective with the following log-likelihood term:

$$\textstyle\sum_{\mathbf{g} \in \mathcal{G}^K} \log \boldsymbol{\omega}(\mathbb{1}\{\mathbf{C} = \mathbf{g}\})_{\boldsymbol{\beta}^*(\mathbf{g})} \tag{67}$$

which penalizes the target CBM, for each $\mathbf{g} \in \mathcal{G}^K$, based on the $\boldsymbol{\beta}^*(\mathbf{g})$ component of $\boldsymbol{\omega}$, i.e., the predicted probability of the ground-truth label. This can be rewritten as a constraint on $\boldsymbol{\beta}$ as follows:

$$\mathbb{1}\{\textstyle\bigwedge_{\mathbf{g} \in \mathcal{G}^K} \boldsymbol{\beta}(\mathbf{g}) = \boldsymbol{\beta}^*(\mathbf{g})\} \tag{68}$$

When $\mathcal{G}^K = \mathcal{G}$ it necessarily holds that $\boldsymbol{\beta} \equiv \boldsymbol{\beta}^*$.

## D.4 Reconstruction penalty

When focusing on reconstructing the input from the bottleneck, the probability distributions $p(\mathbf{C} \mid \mathbf{X})$ may not carry enough information to completely determine the input $\mathbf{X}$. In fact, if $\mathbf{X}$ depends also on stylistic variables $\mathbf{S} \in \mathbb{R}^q$, it becomes impossible to determine the input solely from $\mathbf{G}$. This means that training a decoder only passing $\mathbf{c} \sim p(\mathbf{C} \mid \mathbf{X})$ would not convey enough information and would reduce the benefits of the reconstruction term. Marconato et al. [19] have studied this setting and proposed to also include additional variables $\mathbf{Z} \in \mathbb{R}^q$ in the bottleneck to overcome this problem. The encoder/decoder architecture works as follows: first the input $\mathbf{x}$ is encoded into $p(\mathbf{C}, \mathbf{Z} \mid \mathbf{x})$ by the model and after sampling $(\mathbf{c}, \mathbf{z}) \sim p(\mathbf{C}, \mathbf{Z} \mid \mathbf{x})$ the decoder $\mathbf{d}$ reconstruct the an input image $\hat{\mathbf{x}} = \mathbf{d}(\mathbf{c}, \mathbf{z})$. Following, under the assumption of *content-style separation*, i.e., both the encoder and the decoder process independently the concept and the stylistic variables, it holds that the constraint fr maps $\boldsymbol{\alpha} \in \mathsf{Vert}(\mathcal{A})$ given by the reconstruction penalty result in:

$$\mathbb{1}\{\textstyle\bigwedge_{\mathbf{g}, \mathbf{g}' \in \operatorname{supp}(\mathbf{G}) : \mathbf{g} \neq \mathbf{g}'} \boldsymbol{\alpha}(\mathbf{g}) \neq \boldsymbol{\alpha}(\mathbf{g}')\} \tag{69}$$

The proof for this can be found in [19, Proposition 6]. Notice that, with full support over $\mathcal{G}$, the only such maps $\boldsymbol{\alpha}$ must not confuse one concept vector for another, i.e., at most there are at most $|\mathcal{G}|!$ different valid $\boldsymbol{\alpha}$'s that are essentially a bijection from $\mathcal{G}$ to $\mathcal{C}$.

# E Models that satisfy Assumption 3.8

## E.1 Theoretical analysis

**Probabilistic Logic Models**. For models like DeepProbLog [41], the Semantic Loss [39], Semantic-Probabilistic Layers [44], $\boldsymbol{\omega}$ is defined as:

$$\boldsymbol{\omega}(p(\mathbf{C})) := \sum_{\mathbf{c} \in \mathcal{C}} \boldsymbol{\beta}(\mathbf{c}) p(\mathbf{C} = \mathbf{c}) \tag{70}$$

Now, take $\mathbf{c}, \mathbf{c}' \in \mathcal{C}$ such that: $\boldsymbol{\beta}(\mathbf{c}) \neq \boldsymbol{\beta}(\mathbf{c}')$. Then, for any $\lambda \in (0, 1)$ it holds that:

$$\boldsymbol{\omega}(\lambda \mathbb{1}\{\mathbf{C} = \mathbf{c}\} + (1-\lambda)\mathbb{1}\{\mathbf{C} = \mathbf{c}'\}) = \lambda\boldsymbol{\beta}(\mathbf{c}) + (1-\lambda)\boldsymbol{\beta}(\mathbf{c}') \tag{71}$$

$$\implies \max_{\mathbf{y} \in \mathcal{Y}} \boldsymbol{\omega}(\lambda \mathbb{1}\{\mathbf{C} = \mathbf{c}\} + (1-\lambda)\mathbb{1}\{\mathbf{C} = \mathbf{c}'\})_{\mathbf{y}} = \max_{\mathbf{y} \in \mathcal{Y}} \left(\lambda\boldsymbol{\beta}(\mathbf{c})_{\mathbf{y}} + (1-\lambda)\boldsymbol{\beta}(\mathbf{c}')_{\mathbf{y}}\right) \tag{72}$$

Notice that for all $\mathbf{y} \in \mathcal{Y}$, by the convexity of the $\max$ operator it holds that:

$$\max_{\mathbf{y} \in \mathcal{Y}} \left( \lambda \boldsymbol{\beta}(\mathbf{c})_{\mathbf{y}} + (1 - \lambda) \boldsymbol{\beta}(\mathbf{c}')_{\mathbf{y}} \right) \leq \tag{73}$$

$$\lambda \max \left( \max_{\mathbf{y} \in \mathcal{Y}} \boldsymbol{\beta}(\mathbf{c})_{\mathbf{y}}, \max_{\mathbf{y} \in \mathcal{Y}} \boldsymbol{\beta}(\mathbf{c}')_{\mathbf{y}} \right) + (1 - \lambda) \max \left( \max_{\mathbf{y} \in \mathcal{Y}} \boldsymbol{\beta}(\mathbf{c})_{\mathbf{y}}, \max_{\mathbf{y} \in \mathcal{Y}} \boldsymbol{\beta}(\mathbf{c}')_{\mathbf{y}} \right) \tag{74}$$

$$= \max \left( \max_{\mathbf{y} \in \mathcal{Y}} (\boldsymbol{\beta}(\mathbf{c})_{\mathbf{y}}), \max_{\mathbf{y} \in \mathcal{Y}} \boldsymbol{\beta}(\mathbf{c}')_{\mathbf{y}} \right) \tag{75}$$

Equality holds if and only if $\operatorname{argmax}_{\mathbf{y} \in \mathcal{Y}} \boldsymbol{\beta}(\mathbf{c})_{\mathbf{y}} = \operatorname{argmax}_{\mathbf{y} \in \mathcal{Y}} \boldsymbol{\beta}(\mathbf{c}')_{\mathbf{y}}$ and $\max_{\mathbf{y} \in \mathcal{Y}} \boldsymbol{\beta}(\mathbf{c})_{\mathbf{y}} = \max_{\mathbf{y} \in \mathcal{Y}} \boldsymbol{\beta}(\mathbf{c}')_{\mathbf{y}}$. This proves that probabilistic logic methods satisfy Assumption 3.8.

**Deep Symbolic Learning** [50]. At inference time, DSL predicts a concept vector $\mathbf{c} \in \mathcal{C}$ from a conditional probability concept distribution, that is, it implements a function $\mathbf{f}_{DSL} : \mathbb{R}^n \to \mathcal{C}$ where:

$$\mathbf{f}_{DSL}(\mathbf{x}) := \operatorname*{argmax}_{\mathbf{c} \in \mathcal{C}} \tilde{p}(\mathbf{c} \mid \mathbf{x}) \tag{76}$$

and $\tilde{p}(\mathbf{C} \mid \mathbf{X})$ is a learned conditional distribution on concepts. (Note that $\mathbf{f}_{DSL}(\mathbf{X}) \neq \tilde{p}(\mathbf{C} \mid \mathbf{X})$.) Hence, $\boldsymbol{\alpha} \in \mathcal{A}$ can be defined as:

$$\boldsymbol{\alpha}(\mathbf{g}) := \mathbb{E}_{\mathbf{x} \sim p^*(\mathbf{X} | \mathbf{g})} [\mathbb{1}\{\mathbf{C} = \mathbf{f}_{DSL}(\mathbf{x})\}] \tag{77}$$

Notice that the learned $\boldsymbol{\beta} : \mathcal{G} \to \Delta_{\mathcal{Y}}$ consists in a look up table from concepts vectors $\mathbf{c} \in \mathcal{C}$, thereby giving the conditional distribution:

$$p_{DSL}(\mathbf{Y} \mid \mathbf{g}) := \mathbb{E}_{\mathbf{x} \sim p^*(\mathbf{X} | \mathbf{g})} [\boldsymbol{\beta}(\mathbf{f}_{DSL}(\mathbf{x}))] \tag{78}$$

By considering the measure defined by $p^*(\mathbf{x} \mid \mathbf{g}) d\mathbf{x}$ and the transformation $\mathbf{c} = \mathbf{f}_{DSL}(\mathbf{x})$, we can pass to the new measure $p(\mathbf{c} \mid \mathbf{g}) = \boldsymbol{\alpha}(\mathbf{g})$ obtaining:

$$p_{DSL}(\mathbf{Y} \mid \mathbf{g}) = \sum_{\mathbf{c} \in \mathcal{C}} \boldsymbol{\beta}(\mathbf{c}) p(\mathbf{c} \mid \mathbf{g}) \tag{79}$$

$$= \sum_{\mathbf{c} \in \mathcal{C}} \boldsymbol{\beta}(\mathbf{c}) \boldsymbol{\alpha}(\mathbf{g})_{\mathbf{c}} \tag{80}$$

$$= (\boldsymbol{\omega}_{DSL} \circ \boldsymbol{\alpha})(\mathbf{g}) \tag{81}$$

where we used:

$$\boldsymbol{\omega}_{DSL}(p(\mathbf{C})) := \sum_{\mathbf{c} \in \mathcal{C}} \boldsymbol{\beta}(\mathbf{c}) p(\mathbf{C} = \mathbf{c}) \tag{82}$$

This form matches that of probabilistic logic methods in Eq. (70), hence Assumption 3.8 similarly applies.

**Concept Bottleneck Models**. We consider CBNMs implementing a linear layer as $\boldsymbol{\omega} : \Delta_{\mathcal{C}} \to \Delta_{\mathcal{Y}}$, as customary [3]. The linear layer consists of a matrix $\mathbf{W} \in \mathbb{R}^{a \times b}$ with $a := |\mathcal{G}|$ rows and $b := |\mathcal{Y}|$ columns. In this setting, a probability distribution $p(\mathbf{C}) \in \Delta_{\mathcal{C}}$ corresponds to the input vector passed to the inference layer $\boldsymbol{\omega}$, implemented by the linear layer $\mathbf{W}$ and a softmax operator. Notice that we can equivalently consider the tensor associated with the linear layer, $\mathbf{w}_{\mathbf{c}}^{\mathbf{y}}$, where lower-indices $\mathbf{c} = (c_1, \ldots, c_k)$ are the possible values of concepts and higher-indices $\mathbf{y} = (y_1, \ldots, y_\ell)$ are the possible values of the labels. The scalar index $c$ (resp. $y$) runs over concept vectors $\mathbf{c} \in \mathcal{C}$ (resp. $\mathbf{y}$), e.g., for two dimensional binary concepts, $c = 1$ corresponds to $\mathbf{c} = (0, 1)^\top \in \{0, 1\}^2$ ($\mathbf{W}_{1,:} = \mathbf{w}_{01}^{\mathbf{y}}$) and $c = 2$ to $\mathbf{c} = (1, 0)^\top$ ($\mathbf{W}_{2,:} = \mathbf{w}_{10}^{\mathbf{y}}$). In the following, we make use of the tensor notation for simplicity of exposition.

Given a concept probability vector $\mathbf{p} \in \Delta_{\mathcal{C}}$, with components $\mathbf{p}_{\mathbf{c}} = p(\mathbf{C} = \mathbf{c})$, the conditional distribution over labels is given by the softmax operator:

$$p(\mathbf{y} \mid \mathbf{c}) = \boldsymbol{\omega}(\mathbf{p})_{\mathbf{y}} = \frac{\exp \left( \sum_{\mathbf{c} \in \mathcal{C}} \mathbf{W}_{\mathbf{c},\mathbf{y}} \mathbf{p}_{\mathbf{c}} \right)}{\sum_{\mathbf{y}' \in \mathcal{Y}} \exp \left( \sum_{\mathbf{c} \in \mathcal{C}} \mathbf{W}_{\mathbf{c},\mathbf{y}'} \mathbf{p}_{\mathbf{c}} \right)} = \frac{\exp \left( \sum_{\mathbf{c} \in \mathcal{C}} \mathbf{w}_{\mathbf{c}}^{\mathbf{y}} \mathbf{p}_{\mathbf{c}} \right)}{\sum_{\mathbf{y}' \in \mathcal{Y}} \exp \left( \sum_{\mathbf{c} \in \mathcal{C}} \mathbf{w}_{\mathbf{c}}^{\mathbf{y}'} \mathbf{p}_{\mathbf{c}} \right)} \tag{83}$$

A CBNM may not satisfy Assumption 3.8 for arbitrary choices of weights $\mathbf{W}$. Notice also that, since $\boldsymbol{\omega}$ includes a softmax operator, the $\boldsymbol{\beta} \in \mathsf{Vert}(\mathcal{B})$ expressed by CBNMs *cannot* be deterministic .[6] This is a

---

[6] In fact, attributing near 1 probability to a label $Y$ is only possible if the weights are extremely high in magnitude.

limitation for all CBNM. Specifically, in the context of Assumption 3.2, it means that they cannot reach an optimum of maximum likelihood and therefore learn deterministic maps $\beta^* \in \mathsf{Vert}(\mathcal{B})$ (Eq. (8)).

To proceed, we focus on a special case: a near-optimal CBNM that can express an "almost deterministic" conditional distribution $\beta \in \mathcal{B}$, where a subset of weights $\mathbf{w}_\mathbf{c}^\mathbf{y}$ is very high or very low. For this to happen, we need $\beta$ to be peaked on concept vectors $\mathbf{c} \in \mathcal{C}$. This can happen only when the magnitude of one $\mathbf{w}_\mathbf{c}^\mathbf{y}$ is much higher than other $\mathbf{w}_\mathbf{c}^{\mathbf{y}'}$. Hence, we formulate the following condition for CBNM:

**Definition E.1.** *Consider a CBNM $(\mathbf{f}, \boldsymbol{\omega}) \in \mathcal{F} \times \Omega$ with weights $\mathbf{W} = \{\mathbf{w}_\mathbf{c}^\mathbf{y}\}$ and let $M > |\mathcal{Y} - 1|$. We say that it is $\log(M)$-deterministic if, for all $\mathbf{c} \in \mathcal{C}$, there exists a $\mathbf{y} \in \mathcal{Y}$ such that:*

$$\forall \mathbf{y}' \neq \mathbf{y}, \ \mathbf{w}_\mathbf{c}^\mathbf{y} - \mathbf{w}_\mathbf{c}^{\mathbf{y}'} \geq \log M \tag{84}$$

$$\forall \mathbf{y}' \neq \mathbf{y}, \forall \mathbf{y}'' \neq \mathbf{y}, \ |\mathbf{w}_\mathbf{c}^{\mathbf{y}'} - \mathbf{w}_\mathbf{c}^{\mathbf{y}''}| \leq \log M \tag{85}$$

This is helpful, as any $log(M)$-deterministic CBNM can flexibly approximate deterministic $\beta$'s, as we show next:

**Proposition E.2.** *Consider a $\log(M)$-deterministic CBNM $(\mathbf{f}, \boldsymbol{\omega}) \in \mathcal{F} \times \Omega$. We have:*

$$\forall \mathbf{c} \in \mathcal{C}, \ \max_{\mathbf{y} \in \mathcal{Y}} \boldsymbol{\omega}(\mathbf{c})_\mathbf{y} \geq \frac{1}{1 + (|\mathcal{Y}| - 1)/M} \tag{86}$$

*Also:*

$$\forall \mathbf{c} \in \mathcal{C}, \ \lim_{M \to +\infty} \max_{\mathbf{y} \in \mathcal{Y}} \boldsymbol{\omega}(\mathbf{c})_\mathbf{y} = 1 \tag{87}$$

*Proof.* For $\mathbf{c} \in \mathcal{C}$, let $\mathbf{y} = \mathrm{argmax}_{\mathbf{y}'} \boldsymbol{\omega}(\mathbb{1}\{\mathbf{C} = \mathbf{c}\})_{\mathbf{y}'}$. We consider the expression:

$$\boldsymbol{\omega}(\mathbb{1}\{\mathbf{C} = \mathbf{c}\}) = \frac{\exp \mathbf{w}_\mathbf{c}^\mathbf{y}}{\sum_{\mathbf{y}' \in \mathcal{C}} \exp \mathbf{w}_\mathbf{c}^{\mathbf{y}'}} \tag{88}$$

$$= \frac{1}{1 + \sum_{\mathbf{y}' \neq \mathbf{y}} \exp(-\mathbf{w}_\mathbf{c}^\mathbf{y} + \mathbf{w}_\mathbf{c}^{\mathbf{y}'})} \tag{89}$$

We now make use of the fact that the CBNM is $\log(M)$-deterministic and use Eq. (84) to obtain:

$$\frac{1}{1 + \sum_{\mathbf{y}' \neq \mathbf{y}} \exp(-(\mathbf{w}_\mathbf{c}^\mathbf{y} - \mathbf{w}_\mathbf{c}^{\mathbf{y}'}))} \geq \frac{1}{1 + \sum_{\mathbf{y}' \neq \mathbf{y}} \exp(-\log M)} \tag{90}$$

$$= \frac{1}{1 + \sum_{\mathbf{y}' \neq \mathbf{y}} \frac{1}{M}} \tag{91}$$

$$= \frac{1}{1 + (|\mathcal{Y}| - 1)/M} \tag{92}$$

Putting everything together yields:

$$\max_{\mathbf{y} \in \mathcal{Y}} \boldsymbol{\omega}(\mathbf{c})_\mathbf{y} \geq \frac{1}{1 + (|\mathcal{Y}| - 1)/M} \tag{93}$$

Now, consider the limit for large $M \in \mathbb{R}$:

$$\lim_{M \to +\infty} \max_{\mathbf{y} \in \mathcal{Y}} \boldsymbol{\omega}(\mathbf{c})_\mathbf{y} \geq \lim_{M \to +\infty} \frac{1}{1 + (|\mathcal{Y}| - 1)/M} = 1 \tag{94}$$

This concludes the proof. $\qquad\square$

The last point of Proposition E.2 shows a viable way to get peaked label distributions from a $\log(M)$-deterministic CBNM. Specifically, the limit guarantees that these CBNMs can approach an optimal likelihood. Now, we prove that a $\log(M)$-deterministic CBNM respects Assumption 3.8.

**Proposition E.3.** *A CBNM $(\mathbf{f}, \boldsymbol{\omega}) \in \mathcal{F} \times \Omega$ that is $\log(M)$-deterministic (Definition E.1) satisfies Assumption 3.8, i.e., for all $\lambda \in (0, 1)$ and for all $\mathbf{c} \neq \mathbf{c}'$ such that $\mathrm{argmax}_{\mathbf{y} \in \mathcal{Y}} \boldsymbol{\omega}(\mathbb{1}\{\mathbf{C} = \mathbf{c}\})_\mathbf{y} \neq \mathrm{argmax}_{\mathbf{y} \in \mathcal{Y}} \boldsymbol{\omega}(\mathbb{1}\{\mathbf{C} = \mathbf{c}'\})_\mathbf{y}$, it holds :*

$$\max_{\mathbf{y} \in \mathcal{Y}} \boldsymbol{\omega}(\lambda \mathbb{1}\{\mathbf{C} = \mathbf{c}_1\} + (1 - \lambda)\mathbb{1}\{\mathbf{C} = \mathbf{c}_2\})_\mathbf{y} < \max \left( \max_{\mathbf{y} \in \mathcal{Y}} \boldsymbol{\omega}(\mathbb{1}\{\mathbf{C} = \mathbf{c}_1\})_\mathbf{y}, \max_{\mathbf{y} \in \mathcal{Y}} \boldsymbol{\omega}(\mathbb{1}\{\mathbf{C} = \mathbf{c}_2\})_\mathbf{y} \right) \tag{95}$$

*Proof.* Let $\mathbf{y}_1 := \operatorname{argmax}_{\mathbf{y}\in\mathcal{Y}} \boldsymbol{\omega}(\mathbb{1}\{\mathbf{C} = \mathbf{c}_1\})_{\mathbf{y}}$ and $\mathbf{y}_2 := \operatorname{argmax}_{\mathbf{y}\in\mathcal{Y}} \boldsymbol{\omega}(\mathbb{1}\{\mathbf{C} = \mathbf{c}_2\})_{\mathbf{y}}$ and $\lambda_1 := \lambda$ and $\lambda_2 := 1 - \lambda$, with $\lambda \in (0,1)$ and $\mathbf{y}_1 \neq \mathbf{y}_2$. The proof consists of two steps.

First, we check that the values taken by $\boldsymbol{\omega}(\lambda\mathbb{1}\{\mathbf{C} = \mathbf{c}_1\} + (1 - \lambda)\mathbb{1}\{\mathbf{C} = \mathbf{c}_2\})_{\mathbf{y}_i}$, for $i \in \{1, 2\}$, are lower than those taken at the extremes. The explicit expression for $\boldsymbol{\omega}(\cdot)_{\mathbf{y}_1}$ is given by:

$$\boldsymbol{\omega}(\lambda_1\mathbb{1}\{\mathbf{C} = \mathbf{c}_1\} + \lambda_2\mathbb{1}\{\mathbf{C} = \mathbf{c}_2\})_{\mathbf{y}_1} = \frac{e^{\lambda_1\mathbf{w}_{\mathbf{c}_1}^{\mathbf{y}_1} + \lambda_2\mathbf{w}_{\mathbf{c}_2}^{\mathbf{y}_1}}}{\sum_{\mathbf{y}\in\mathcal{Y}} e^{\lambda_1\mathbf{w}_{\mathbf{c}_1}^{\mathbf{y}} + \lambda_2\mathbf{w}_{\mathbf{c}_2}^{\mathbf{y}}}} \tag{96}$$

We differentiating over $\lambda$, leveraging the fact that $\partial\lambda_1/\partial\lambda = 1$ and $\partial\lambda_2/\partial\lambda = -1$. Let $Z(\lambda) := \sum_{\mathbf{y}\in\mathcal{Y}} \exp(\lambda_1\mathbf{w}_{\mathbf{c}_1}^{\mathbf{y}} + \lambda_2\mathbf{w}_{\mathbf{c}_2}^{\mathbf{y}})$. We first evaluate the derivative of this expression:

$$\frac{\partial Z(\lambda)}{\partial\lambda} = \sum_{\mathbf{y}\in\mathcal{Y}} \frac{\partial}{\partial\lambda} e^{\lambda_1\mathbf{w}_{\mathbf{c}_1}^{\mathbf{y}} + \lambda_2\mathbf{w}_{\mathbf{c}_2}^{\mathbf{y}}} \tag{97}$$

$$= \sum_{\mathbf{y}\in\mathcal{Y}} e^{\lambda_1\mathbf{w}_{\mathbf{c}_1}^{\mathbf{y}} + \lambda_2\mathbf{w}_{\mathbf{c}_2}^{\mathbf{y}}} (\partial_\lambda\lambda_1\mathbf{w}_{\mathbf{c}_1}^{\mathbf{y}} + \partial_\lambda\lambda_2\mathbf{w}_{\mathbf{c}_2}^{\mathbf{y}}) \tag{98}$$

$$= \sum_{\mathbf{y}\in\mathcal{Y}} e^{\lambda_1\mathbf{w}_{\mathbf{c}_1}^{\mathbf{y}} + \lambda_2\mathbf{w}_{\mathbf{c}_2}^{\mathbf{y}}} (\mathbf{w}_{\mathbf{c}_1}^{\mathbf{y}} - \mathbf{w}_{\mathbf{c}_2}^{\mathbf{y}}) \tag{99}$$

$$= Z(\lambda) \sum_{\mathbf{y}\in\mathcal{Y}} \frac{e^{\lambda_1\mathbf{w}_{\mathbf{c}_1}^{\mathbf{y}} + \lambda_2\mathbf{w}_{\mathbf{c}_2}^{\mathbf{y}}}}{Z(\lambda)} (\mathbf{w}_{\mathbf{c}_1}^{\mathbf{y}} - \mathbf{w}_{\mathbf{c}_2}^{\mathbf{y}}) \tag{100}$$

$$= Z(\lambda) \sum_{\mathbf{y}\in\mathcal{Y}} (\mathbf{w}_{\mathbf{c}_1}^{\mathbf{y}} - \mathbf{w}_{\mathbf{c}_2}^{\mathbf{y}})\boldsymbol{\omega}(\lambda_1\mathbb{1}\{\mathbf{C} = \mathbf{c}_1\} + \lambda_2\mathbb{1}\{\mathbf{C} = \mathbf{c}_2\})_{\mathbf{y}} \tag{101}$$

This is legitimate as $Z(\lambda) > 0$ by definition. Using this result we get:

$$\frac{\partial}{\partial\lambda} \frac{e^{\lambda_1\mathbf{w}_{\mathbf{c}_1}^{\mathbf{y}_1} + \lambda_2\mathbf{w}_{\mathbf{c}_2}^{\mathbf{y}_2}}}{Z(\lambda)} = \frac{\partial_\lambda e^{\lambda_1\mathbf{w}_{\mathbf{c}_1}^{\mathbf{y}_1} + \lambda_2\mathbf{w}_{\mathbf{c}_2}^{\mathbf{y}_1}} Z(\lambda) - e^{\lambda_1\mathbf{w}_{\mathbf{c}_1}^{\mathbf{y}_1} + \lambda_2\mathbf{w}_{\mathbf{c}_2}^{\mathbf{y}_1}} \partial_\lambda Z(\lambda)}{Z(\lambda)^2} \tag{102}$$

$$= \frac{e^{\lambda_1\mathbf{w}_{\mathbf{c}_1}^{\mathbf{y}_1} + \lambda_2\mathbf{w}_{\mathbf{c}_2}^{\mathbf{y}_1}} (\mathbf{w}_{\mathbf{c}_1}^{\mathbf{y}_1} - \mathbf{w}_{\mathbf{c}_2}^{\mathbf{y}_1}) Z(\lambda)}{Z(\lambda)^2}$$
$$- \frac{e^{\lambda_1\mathbf{w}_{\mathbf{c}_1}^{\mathbf{y}_1} + \lambda_2\mathbf{w}_{\mathbf{c}_2}^{\mathbf{y}_1}} Z(\lambda) \sum_{\mathbf{y}\in\mathcal{Y}}(\mathbf{w}_{\mathbf{c}_1}^{\mathbf{y}} - \mathbf{w}_{\mathbf{c}_2}^{\mathbf{y}})\boldsymbol{\omega}(\lambda_1\mathbb{1}\{\mathbf{C} = \mathbf{c}_1\} + \lambda_2\mathbb{1}\{\mathbf{C} = \mathbf{c}_2\})_{\mathbf{y}}}{Z(\lambda)^2} \tag{103}$$

$$= \frac{e^{\lambda_1\mathbf{w}_{\mathbf{c}_1}^{\mathbf{y}_1} + \lambda_2\mathbf{w}_{\mathbf{c}_2}^{\mathbf{y}_1}} (\mathbf{w}_{\mathbf{c}_1}^{\mathbf{y}_1} - \mathbf{w}_{\mathbf{c}_2}^{\mathbf{y}_1})}{Z(\lambda)}$$
$$- \frac{e^{\lambda_1\mathbf{w}_{\mathbf{c}_1}^{\mathbf{y}_1} + \lambda_2\mathbf{w}_{\mathbf{c}_2}^{\mathbf{y}_1}} \sum_{\mathbf{y}\in\mathcal{Y}}(\mathbf{w}_{\mathbf{c}_1}^{\mathbf{y}} - \mathbf{w}_{\mathbf{c}_2}^{\mathbf{y}})\boldsymbol{\omega}(\lambda_1\mathbb{1}\{\mathbf{C} = \mathbf{c}_1\} + \lambda_2\mathbb{1}\{\mathbf{C} = \mathbf{c}_2\})_{\mathbf{y}}}{Z(\lambda)} \tag{104}$$

$$= \frac{e^{\lambda_1\mathbf{w}_{\mathbf{c}_1}^{\mathbf{y}_1} + \lambda_2\mathbf{w}_{\mathbf{c}_2}^{\mathbf{y}_1}} \left((\mathbf{w}_{\mathbf{c}_1}^{\mathbf{y}_1} - \mathbf{w}_{\mathbf{c}_2}^{\mathbf{y}_1}) - \sum_{\mathbf{y}\in\mathcal{Y}}(\mathbf{w}_{\mathbf{c}_1}^{\mathbf{y}} - \mathbf{w}_{\mathbf{c}_2}^{\mathbf{y}})\boldsymbol{\omega}(\lambda_1\mathbb{1}\{\mathbf{C} = \mathbf{c}_1\} + \lambda_2\mathbb{1}\{\mathbf{C} = \mathbf{c}_2\})_{\mathbf{y}}\right)}{Z(\lambda)}$$
$$\tag{105}$$

Next, we analyze the sign of the derivative. Notice that we can focus only on the following term since the others are always positive:

$$(\mathbf{w}_{\mathbf{c}_1}^{\mathbf{y}_1} - \mathbf{w}_{\mathbf{c}_2}^{\mathbf{y}_1}) - \sum_{\mathbf{y}\in\mathcal{Y}}(\mathbf{w}_{\mathbf{c}_1}^{\mathbf{y}} - \mathbf{w}_{\mathbf{c}_2}^{\mathbf{y}})\boldsymbol{\omega}(\lambda_1\mathbb{1}\{\mathbf{C} = \mathbf{c}_1\} + \lambda_2\mathbb{1}\{\mathbf{C} = \mathbf{c}_2\})_{\mathbf{y}} \tag{106}$$

From this expression, we can make use of the fact that in general, for a scalar function $f(\mathbf{x})$, we have $\mathbb{E}_{\mathbf{X}}[f(\mathbf{X})] \leq \max_{\mathbf{x}} f(\mathbf{x})$ and consider the following:

$$(\mathbf{w}_{\mathbf{c}_1}^{\mathbf{y}_1} - \mathbf{w}_{\mathbf{c}_2}^{\mathbf{y}_1}) - \sum_{\mathbf{y}\in\mathcal{Y}}(\mathbf{w}_{\mathbf{c}_1}^{\mathbf{y}} - \mathbf{w}_{\mathbf{c}_2}^{\mathbf{y}})\boldsymbol{\omega}(\lambda_1\mathbb{1}\{\mathbf{C} = \mathbf{c}_1\} + \lambda_2\mathbb{1}\{\mathbf{C} = \mathbf{c}_2\})_{\mathbf{y}}$$

$$\geq (\mathbf{w}_{\mathbf{c}_1}^{\mathbf{y}_1} - \mathbf{w}_{\mathbf{c}_2}^{\mathbf{y}_1}) - \max_{\mathbf{y}\in\mathcal{Y}}(\mathbf{w}_{\mathbf{c}_1}^{\mathbf{y}} - \mathbf{w}_{\mathbf{c}_2}^{\mathbf{y}}) \tag{107}$$

$$= (\mathbf{w}_{\mathbf{c}_1}^{\mathbf{y}_1} - \mathbf{w}_{\mathbf{c}_2}^{\mathbf{y}_1}) - (\mathbf{w}_{\mathbf{c}_1}^{\mathbf{y}_1} - \mathbf{w}_{\mathbf{c}_2}^{\mathbf{y}_1}) \tag{108}$$

$$= 0 \tag{109}$$

where in the second line we used that the maximum is given by $\mathbf{y}_1$. Therefore, the derivative is always increasing in the $[0, 1]$ interval, meaning that:

$$\forall \lambda \in [0, 1], \ \boldsymbol{\omega}(\lambda_1 \mathbb{1}\{\mathbf{C} = \mathbf{c}_1\} + \lambda_2 \mathbb{1}\{\mathbf{C} = \mathbf{c}_2\})_{\mathbf{y}_1} \leq \boldsymbol{\omega}(\mathbb{1}\{\mathbf{C} = \mathbf{c}_1\})_{\mathbf{y}_1} \tag{110}$$

where the equality holds if an only if $\lambda = 1$. Similarly, the derivative for $\boldsymbol{\omega}(\cdot)_{\mathbf{y}_2}$ gives:

$$\frac{\partial \boldsymbol{\omega}(\lambda_1 \mathbb{1}\{\mathbf{C} = \mathbf{c}_1\} + \lambda_2 \mathbb{1}\{\mathbf{C} = \mathbf{c}_2\})_{\mathbf{y}_2}}{\partial \lambda}$$
$$= \frac{e^{\lambda_1 \mathbf{w}_{\mathbf{c}_1}^{\mathbf{y}_2} + \lambda_2 \mathbf{w}_{\mathbf{c}_2}^{\mathbf{y}_2}}\left((\mathbf{w}_{\mathbf{c}_1}^{\mathbf{y}_2} - \mathbf{w}_{\mathbf{c}_2}^{\mathbf{y}_2}) - \sum_{\mathbf{y} \in \mathcal{Y}}(\mathbf{w}_{\mathbf{c}_1}^{\mathbf{y}} - \mathbf{w}_{\mathbf{c}_2}^{\mathbf{y}})\boldsymbol{\omega}(\lambda_1 \mathbb{1}\{\mathbf{C} = \mathbf{c}_1\} + \lambda_2 \mathbb{1}\{\mathbf{C} = \mathbf{c}_2\})_{\mathbf{y}}\right)}{Z(\lambda)}$$
$$\tag{111}$$

By using the fact that $\mathbb{E}_{\mathbf{X}}[f(\mathbf{X})] \geq \min_{\mathbf{x}} f(\mathbf{x})$ for a scalar function $f(\mathbf{x})$, we obtain:

$$(\mathbf{w}_{\mathbf{c}_1}^{\mathbf{y}_2} - \mathbf{w}_{\mathbf{c}_2}^{\mathbf{y}_2}) - \sum_{\mathbf{y} \in \mathcal{Y}}(\mathbf{w}_{\mathbf{c}_1}^{\mathbf{y}} - \mathbf{w}_{\mathbf{c}_2}^{\mathbf{y}})\boldsymbol{\omega}(\lambda_1 \mathbb{1}\{\mathbf{C} = \mathbf{c}_1\} + \lambda_2 \mathbb{1}\{\mathbf{C} = \mathbf{c}_2\})_{\mathbf{y}}$$

$$\leq (\mathbf{w}_{\mathbf{c}_1}^{\mathbf{y}_2} - \mathbf{w}_{\mathbf{c}_2}^{\mathbf{y}_2}) - \min_{\mathbf{y} \in \mathcal{Y}}(\mathbf{w}_{\mathbf{c}_1}^{\mathbf{y}} - \mathbf{w}_{\mathbf{c}_2}^{\mathbf{y}}) \tag{112}$$
$$= (\mathbf{w}_{\mathbf{c}_1}^{\mathbf{y}_2} - \mathbf{w}_{\mathbf{c}_2}^{\mathbf{y}_2}) - (\mathbf{w}_{\mathbf{c}_1}^{\mathbf{y}_2} - \mathbf{w}_{\mathbf{c}_2}^{\mathbf{y}_2}) \tag{113}$$
$$= 0 \tag{114}$$

where in the second line we used that the minimum is given by $\mathbf{y}_2$. Hence, the derivative is always decreasing for $\lambda \in [0, 1]$ and it holds:

$$\forall \lambda \in [0, 1], \ \boldsymbol{\omega}(\lambda_1 \mathbb{1}\{\mathbf{C} = \mathbf{c}_1\} + \lambda_2 \mathbb{1}\{\mathbf{C} = \mathbf{c}_2\})_{\mathbf{y}_2} \leq \boldsymbol{\omega}(\mathbb{1}\{\mathbf{C} = \mathbf{c}_2\})_{\mathbf{y}_2} \tag{115}$$

where the equality holds if and only if $\lambda = 0$.

(2) Now, we check the same holds when choosing another element $\mathbf{y}' \neq \mathbf{y}_1, \mathbf{y}_2$. To this end, we consider the following expressions:

$$\frac{p(\mathbf{y} \mid \lambda_1 \mathbb{1}\{\mathbf{C} = \mathbf{c}_1\} + \lambda_2 \mathbb{1}\{\mathbf{C} = \mathbf{c}_2\})}{p(\mathbf{y}_1 \mid \mathbf{c}_1)}, \quad \frac{p(\mathbf{y} \mid \lambda_1 \mathbb{1}\{\mathbf{C} = \mathbf{c}_1\} + \lambda_2 \mathbb{1}\{\mathbf{C} = \mathbf{c}_2\})}{p(\mathbf{y}_2 \mid \mathbf{c}_2)} \tag{116}$$

where $p(\mathbf{y} \mid \mathbf{p}) := \boldsymbol{\omega}(\mathbf{p})_{\mathbf{y}}$. Consider the first expression. We can rewrite it as follows:

$$\frac{p(\mathbf{y} \mid \lambda_1 \mathbb{1}\{\mathbf{C} = \mathbf{c}_1\} + \lambda_2 \mathbb{1}\{\mathbf{C} = \mathbf{c}_2\})}{p(\mathbf{y}_1 \mid \mathbf{c}_1)} \tag{117}$$
$$= \frac{p(\mathbf{y} \mid \lambda_1 \mathbb{1}\{\mathbf{C} = \mathbf{c}_1\} + \lambda_2 \mathbb{1}\{\mathbf{C} = \mathbf{c}_2\})}{p(\mathbf{y}_1 \mid \lambda_1 \mathbb{1}\{\mathbf{C} = \mathbf{c}_1\} + \lambda_2 \mathbb{1}\{\mathbf{C} = \mathbf{c}_2\})} \frac{p(\mathbf{y}_1 \mid \lambda_1 \mathbb{1}\{\mathbf{C} = \mathbf{c}_1\} + \lambda_2 \mathbb{1}\{\mathbf{C} = \mathbf{c}_2\})}{p(\mathbf{y}_1 \mid \mathbf{c}_1)} \tag{118}$$
$$\leq \frac{p(\mathbf{y} \mid \lambda_1 \mathbb{1}\{\mathbf{C} = \mathbf{c}_1\} + \lambda_2 \mathbb{1}\{\mathbf{C} = \mathbf{c}_2\})}{p(\mathbf{y}_1 \mid \lambda_1 \mathbb{1}\{\mathbf{C} = \mathbf{c}_1\} + \lambda_2 \mathbb{1}\{\mathbf{C} = \mathbf{c}_2\})} \tag{119}$$

where in the last line we made use of the fact that the second fraction in the right-hand side of the first line is always $\leq 1$. Similarly we have that:

$$\frac{p(\mathbf{y} \mid \lambda_1 \mathbb{1}\{\mathbf{C} = \mathbf{c}_1\} + \lambda_2 \mathbb{1}\{\mathbf{C} = \mathbf{c}_2\})}{p(\mathbf{y}_2 \mid \mathbf{c}_2)} \leq \frac{p(\mathbf{y} \mid \lambda_1 \mathbb{1}\{\mathbf{C} = \mathbf{c}_1\} + \lambda_2 \mathbb{1}\{\mathbf{C} = \mathbf{c}_2\})}{p(\mathbf{y}_2 \mid \lambda_1 \mathbb{1}\{\mathbf{C} = \mathbf{c}_1\} + \lambda_2 \mathbb{1}\{\mathbf{C} = \mathbf{c}_2\})} \tag{120}$$

We proceed to substituting explicitly the expression for $p(\mathbf{y} \mid \lambda_1 \mathbb{1}\{\mathbf{C} = \mathbf{c}_1\} + \lambda_2 \mathbb{1}\{\mathbf{C} = \mathbf{c}_1\}$ into the upper bound of Eq. (119):

$$\frac{p(\mathbf{y} \mid \lambda_1 \mathbb{1}\{\mathbf{C} = \mathbf{c}_1\} + \lambda_2 \mathbb{1}\{\mathbf{C} = \mathbf{c}_2\})}{p(\mathbf{y}_1 \mid \lambda_1 \mathbb{1}\{\mathbf{C} = \mathbf{c}_1\} + \lambda_2 \mathbb{1}\{\mathbf{C} = \mathbf{c}_2\})} = \frac{e^{\lambda_1 \mathbf{w}_{\mathbf{c}_1}^{\mathbf{y}} + \lambda_2 \mathbf{w}_{\mathbf{c}_2}^{\mathbf{y}}}}{e^{\lambda_1 \mathbf{w}_{\mathbf{c}_1}^{\mathbf{y}_1} + \lambda_2 \mathbf{w}_{\mathbf{c}_2}^{\mathbf{y}_1}}} \frac{Z(\lambda)}{Z(\lambda)} \tag{121}$$
$$= \exp\left(\lambda_1(\mathbf{w}_{\mathbf{c}_1}^{\mathbf{y}} - \mathbf{w}_{\mathbf{c}_1}^{\mathbf{y}_1}) + \lambda_2(\mathbf{w}_{\mathbf{c}_2}^{\mathbf{y}} - \mathbf{w}_{\mathbf{c}_2}^{\mathbf{y}_1})\right) \tag{122}$$
$$\leq \exp\left(-\lambda_1 \log(M) + \lambda_2(\mathbf{w}_{\mathbf{c}_2}^{\mathbf{y}} - \mathbf{w}_{\mathbf{c}_2}^{\mathbf{y}_1})\right)$$
(Substituting the bound from Eq. (84))
$$\leq \exp\left(-\lambda_1 \log(M) + \lambda_2 \log(M)\right)$$
(Substituting the bound from Eq. (85))
$$= M^{\lambda_2 - \lambda_1} \tag{123}$$

With similar steps and substitutions we get that:

$$\frac{p(\mathbf{y} \mid \lambda_1 \mathbb{1}\{\mathbf{C} = \mathbf{c}_1\} + \lambda_2 \mathbb{1}\{\mathbf{C} = \mathbf{c}_2\})}{p(\mathbf{y}_2 \mid \lambda_1 \mathbb{1}\{\mathbf{C} = \mathbf{c}_1\} + \lambda_2 \mathbb{1}\{\mathbf{C} = \mathbf{c}_2\})} \leq M^{\lambda_1 - \lambda_2} \tag{124}$$

Taking the product of Eq. (123) and Eq. (124) we obtain:

$$\frac{p(\mathbf{y} \mid \lambda_1 \mathbb{1}\{\mathbf{C} = \mathbf{c}_1\} + \lambda_2 \mathbb{1}\{\mathbf{C} = \mathbf{c}_2\})}{p(\mathbf{y}_1 \mid \lambda_1 \mathbb{1}\{\mathbf{C} = \mathbf{c}_1\} + \lambda_2 \mathbb{1}\{\mathbf{C} = \mathbf{c}_2\})} \cdot \frac{p(\mathbf{y} \mid \lambda_1 \mathbb{1}\{\mathbf{C} = \mathbf{c}_1\} + \lambda_2 \mathbb{1}\{\mathbf{C} = \mathbf{c}_2\})}{p(\mathbf{y}_2 \mid \lambda_1 \mathbb{1}\{\mathbf{C} = \mathbf{c}_1\} + \lambda_2 \mathbb{1}\{\mathbf{C} = \mathbf{c}_2\})}$$

$$\leq M^{\lambda_2 - \lambda_1} M^{\lambda_1 - \lambda_2} \tag{125}$$

$$= 1 \tag{126}$$

Notice also that:

$$\frac{p(\mathbf{y} \mid \lambda_1 \mathbb{1}\{\mathbf{C} = \mathbf{c}_1\} + \lambda_2 \mathbb{1}\{\mathbf{C} = \mathbf{c}_2\})}{p(\mathbf{y}_1 \mid \lambda_1 \mathbb{1}\{\mathbf{C} = \mathbf{c}_1\} + \lambda_2 \mathbb{1}\{\mathbf{C} = \mathbf{c}_2\})} \cdot \frac{p(\mathbf{y} \mid \lambda_1 \mathbb{1}\{\mathbf{C} = \mathbf{c}_1\} + \lambda_2 \mathbb{1}\{\mathbf{C} = \mathbf{c}_2\})}{p(\mathbf{y}_2 \mid \lambda_1 \mathbb{1}\{\mathbf{C} = \mathbf{c}_1\} + \lambda_2 \mathbb{1}\{\mathbf{C} = \mathbf{c}_2\})}$$

$$\geq \min \left( \frac{p(\mathbf{y} \mid \lambda_1 \mathbb{1}\{\mathbf{C} = \mathbf{c}_1\} + \lambda_2 \mathbb{1}\{\mathbf{C} = \mathbf{c}_2\})^2}{p(\mathbf{y}_1 \mid \lambda_1 \mathbb{1}\{\mathbf{C} = \mathbf{c}_1\} + \lambda_2 \mathbb{1}\{\mathbf{C} = \mathbf{c}_2\})^2}, \frac{p(\mathbf{y} \mid \lambda_1 \mathbb{1}\{\mathbf{C} = \mathbf{c}_1\} + \lambda_2 \mathbb{1}\{\mathbf{C} = \mathbf{c}_2\})^2}{p(\mathbf{y}_2 \mid \lambda_1 \mathbb{1}\{\mathbf{C} = \mathbf{c}_1\} + \lambda_2 \mathbb{1}\{\mathbf{C} = \mathbf{c}_2\})^2} \right) \tag{127}$$

which in turn means that:

$$\min \left( \frac{p(\mathbf{y} \mid \lambda_1 \mathbb{1}\{\mathbf{C} = \mathbf{c}_1\} + \lambda_2 \mathbb{1}\{\mathbf{C} = \mathbf{c}_2\})^2}{p(\mathbf{y}_1 \mid \lambda_1 \mathbb{1}\{\mathbf{C} = \mathbf{c}_1\} + \lambda_2 \mathbb{1}\{\mathbf{C} = \mathbf{c}_2\})^2}, \frac{p(\mathbf{y} \mid \lambda_1 \mathbb{1}\{\mathbf{C} = \mathbf{c}_1\} + \lambda_2 \mathbb{1}\{\mathbf{C} = \mathbf{c}_2\})^2}{p(\mathbf{y}_2 \mid \lambda_1 \mathbb{1}\{\mathbf{C} = \mathbf{c}_1\} + \lambda_2 \mathbb{1}\{\mathbf{C} = \mathbf{c}_2\})^2} \right) \leq 1 \tag{128}$$

$$\implies \min \left( \frac{p(\mathbf{y} \mid \lambda_1 \mathbb{1}\{\mathbf{C} = \mathbf{c}_1\} + \lambda_2 \mathbb{1}\{\mathbf{C} = \mathbf{c}_2\})}{p(\mathbf{y}_1 \mid \lambda_1 \mathbb{1}\{\mathbf{C} = \mathbf{c}_1\} + \lambda_2 \mathbb{1}\{\mathbf{C} = \mathbf{c}_2\})}, \frac{p(\mathbf{y} \mid \lambda_1 \mathbb{1}\{\mathbf{C} = \mathbf{c}_1\} + \lambda_2 \mathbb{1}\{\mathbf{C} = \mathbf{c}_2\})}{p(\mathbf{y}_2 \mid \lambda_1 \mathbb{1}\{\mathbf{C} = \mathbf{c}_1\} + \lambda_2 \mathbb{1}\{\mathbf{C} = \mathbf{c}_2\})} \right) \leq 1 \tag{129}$$

This last expression is in line with the condition of Assumption 3.8, showing that, for all $\mathbf{y} \in \mathcal{Y}$, either $\boldsymbol{\omega}(\mathbb{1}\{\mathbf{C} = \mathbf{c}_1\})_{\mathbf{y}_1}$ or $\boldsymbol{\omega}(\mathbb{1}\{\mathbf{C} = \mathbf{c}_2\})_{\mathbf{y}_2}$ are greater or equal to $\boldsymbol{\omega}(\lambda \mathbb{1}\{\mathbf{C} = \mathbf{c}_1\} + (1-\lambda)\mathbb{1}\{\mathbf{C} = \mathbf{c}_2\})_{\mathbf{y}}$. Combining step (1) and (2), we obtain that Assumption 3.8 holds and that:

$$\max_{\mathbf{y} \in \mathcal{Y}} \boldsymbol{\omega}(\lambda \mathbb{1}\{\mathbf{C} = \mathbf{c}_1\} + (1-\lambda)\mathbb{1}\{\mathbf{C} = \mathbf{c}_2\})_{\mathbf{y}} \leq \max \left( \max_{\mathbf{y} \in \mathcal{Y}} \boldsymbol{\omega}(\mathbb{1}\{\mathbf{C} = \mathbf{c}_1\})_{\mathbf{y}}, \max_{\mathbf{y} \in \mathcal{Y}} \boldsymbol{\omega}(\mathbb{1}\{\mathbf{C} = \mathbf{c}_2\})_{\mathbf{y}} \right) \tag{130}$$

where equality holds if and only if $\lambda \in \{0, 1\}$. $\qquad \square$

### E.2 Numerical evaluation of Assumption 3.8

Finally, we investigate experimentally whether the models used in our experiments satisfy Assumption 3.8. We have already shown in Appendix E.1 that the DSL inference layer reduces to Probabilistic Logic methods and, therefore, we only consider DPL and DPL* as representatives, and our empirical results support this claim. We evaluate CBNMs separately. In this evaluation, train models on MNIST-Add, where there are two concepts $\mathbf{c} = (c_1, c_2) \in [10]^2$ and 19 total labels $y \in [19]$. This yields a total of $100 \times 19$ weights for the inference layer $\boldsymbol{\omega}$, which are fixed (by the prior knowledge) for DPL and learned from data for DPL* and CBNM.

We begin from DPL, where the prior knowledge K defines the inference layer $\boldsymbol{\omega}^*$. In this case, we find that – as expected – Assumption 3.8 is satisfied by all possible pairs $(\mathbf{c}_1, \mathbf{c}_2) \in [10]^2 \times [10]^2$ that predict distinct labels (as requested by the assumption), see Fig. 11.

We now turn to DPL*. Due to the linearity of the inference layer, in order to study the learned inference layer $\boldsymbol{\omega}$ it suffices to consider random weights for $\boldsymbol{\omega}$, cf. Eq. (71). Also in this case, we find that for all possible pairs $(\mathbf{c}_1, \mathbf{c}_2) \in [10]^2 \times [10]^2$ that predict distinct labels, Assumption 3.8 is satisfied. See Fig. 12.

Finally, we evaluate whether a CBNM trained on MNIST-Add and that is close to achieving optimal likelihood satisfies, at least approximately, Assumption 3.8. To this end, we train the CBNM for 150 epochs, reaching a mean negative log-likelihood of $0.0884$ on the test set. We find that Assumption 3.8 is satisfied for the $95\%$ of possible pairs $(\mathbf{c}_1, \mathbf{c}_2) \in [10]^2 \times [10]^2$ giving different labels. For the remaining $5\%$ the assumption is *marginally* violated with a maximum discrepancy:

$$\max_{\mathbf{c}_1, \mathbf{c}_2, \lambda} \left( \max_{\mathbf{y}} \boldsymbol{\omega}(\lambda \mathbb{1}\{\mathbf{C} = \mathbf{c}_i\} + (1-\lambda)\mathbb{1}\{\mathbf{C} = \mathbf{c}_j\})_{\mathbf{y}} - \max \left( \max_{\mathbf{y}} \boldsymbol{\omega}(\mathbb{1}\{\mathbf{c}_1\})_{\mathbf{y}}, \max_{\mathbf{y}} \boldsymbol{\omega}(\mathbb{1}\{\mathbf{c}_2\})_{\mathbf{y}} \right) \right) \tag{131}$$

$$\leq 1.38 \cdot 10^{-3} \tag{132}$$

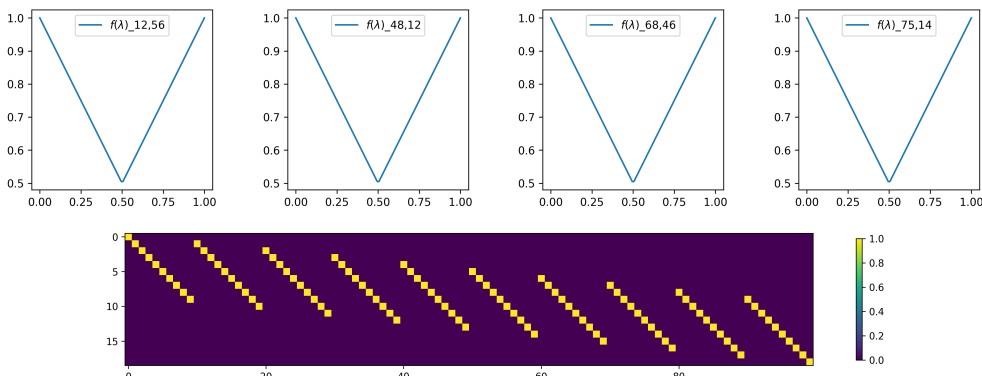

Figure 11: **The logic inference layer of DPL in** `MNIST-Add`. (Top) The behavior of $f(\lambda) := \max_{\mathbf{y}} \boldsymbol{\omega}(\lambda \mathbb{1}\{\mathbf{C} = \mathbf{c}_i\} + (1 - \lambda)\mathbb{1}\{\mathbf{C} = \mathbf{c}_j\})$, for four pairs $i, j$ sampled randomly from the 100 possible worlds. The x-axis represents $\lambda$ and the y-axis $f(\lambda)$. (Bottom) The linear layer weights for DPL.

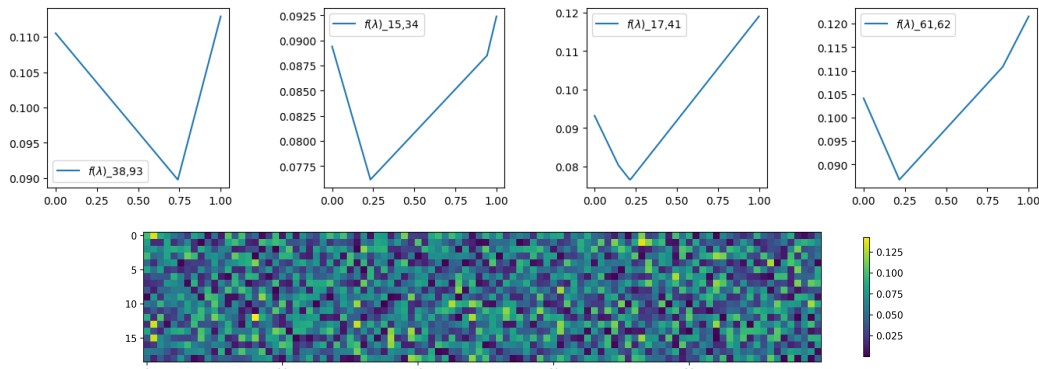

Figure 12: **A random inference layer for** `DPL*` **in** `MNIST-Add`. (Top) The behavior of $f(\lambda) := \max_{\mathbf{y}} \boldsymbol{\omega}(\lambda \mathbb{1}\{\mathbf{C} = \mathbf{c}_i\} + (1 - \lambda)\mathbb{1}\{\mathbf{C} = \mathbf{c}_j\})$, for four pairs $i, j$ sampled randomly from the 100 possible worlds. (Bottom) The linear layer weights for DPL*.

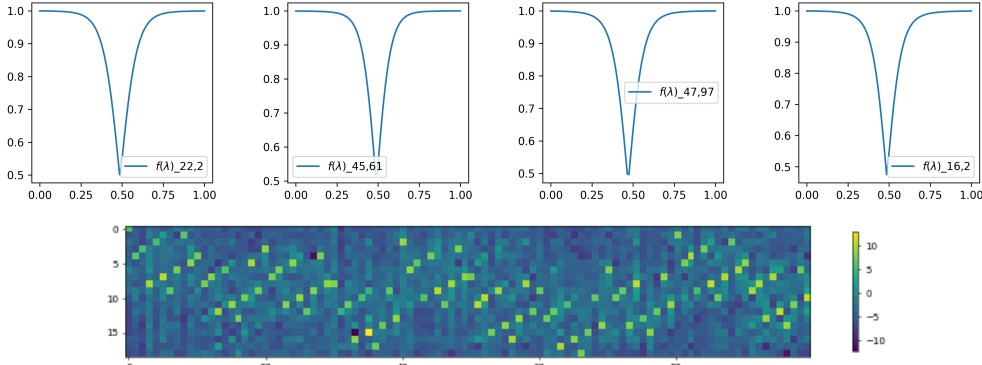

Figure 13: **Optimal** `CBNM` **in** `MNIST-Add`. We visualize the learned weights by a CBNM achieving maximum log-likelihood in `MNIST-Add`. (Top) The behavior of $f(\lambda) := \max_{\mathbf{y}} \boldsymbol{\omega}(\lambda \mathbb{1}\{\mathbf{C} = \mathbf{c}_i\} + (1 - \lambda)\mathbb{1}\{\mathbf{C} = \mathbf{c}_j\})$, for four pairs $i, j$ sampled randomly the 100 possible worlds. The sampled pairs align with Assumption 3.8. (Bottom) The linear layer weights for the trained CBNM.

The results are illustrated in Fig. 13.

