# OpenReview forum: "Shortcuts and Identifiability in Concept-based Models from a Neuro-Symbolic Lens"
_NeurIPS.cc/2025/Conference — NeurIPS 2025 poster_

### Official Review · Reviewer_7RjS · 2025-06-16

**Clarity:** 2
**Significance:** 2
**Originality:** 3
**Rating:** 5
**Confidence:** 3

**Summary:**

This paper investigates joint reasoning shortcuts (JRS) in the context of concept-based models. JRS describes shortcuts emerging from the two-level architecture of CBMs, where the concept extractor, the inference layer or both together can be subject to shortcut learning. The paper analyses these JRS from a theoretical perspective, formally describing situations under which these shortcuts can or cannot occur. Further, the paper illustrates that common mitigation strategies are limited in effect when mitigating JRS, highlighting the importance of the topic.

**Questions:**

Recommendations to improve the clarity of the beginning of the paper:
- For the introduction and section 2, the current setting is often unclear (for example what type of supervision is considered or what parts of the model are learned). I would suggest to make this more clear in the beginning, especially for the examples up to 2.2.
- Further, I would suggest to stick to one running example (and not mixing mnist with the pedestrians)

I am confused by the intended semantics:
- To my understanding, this allows $\alpha$ and $\beta$ to permute the concepts with respect to the ground-truth concepts as long as this inversion is corrected.
- I understand this in the context of a learned $\beta$ and no ground-truth concepts available. But it does not make sense to me in the context of a fixed inference layer, as this does not have the possibility to permute the concepts anymore. Is it then implied that in this case there is no permutation learned by $\alpha$?

Can the authors discuss the practicability of assumptions 3.1 and 3.2 in practice? In particular the assumption that the label can be predicted via without ambiguity from the ground-truth concepts seems unrealistic, as it assumes a perfect and complete ground-truth concept set. (Or, if stated differently: when the existence of such a concept set is assumed, it seems unlikely that assumption 3.1 holds and these concepts are always recoverable).

Further, can the authors provide a brief intuition how the problem of JRS is affected if the CBM is trained independently or if the concept extractor is pretrained in a general setup, as often done in recent CBMNs [1, 2]?

Questions about the experimental setup and evaluation:
- How is the CBNM trained if there is no concept supervision?
- Could the authors explain the evaluation of the learned inference layer? It seems confusing to use $\pi$ on the concepts if the inference layer is learned, assuming that the inference layer itself is not permutation invariant, as then it gets a different input permutation than usual during its training.


Minor:
- Caption of figure 2: introduce the abbreviation (JRS) before using it
- Typo in line 264 (reporhhts)

[1] Oikarinen, Tuomas, et al. "Label-free concept bottleneck models." _arXiv preprint arXiv:2304.06129_ (2023).

[2] Yang, Yue, et al. "Language in a bottle: Language model guided concept bottlenecks for interpretable image classification." _Proceedings of the IEEE/CVF Conference on Computer Vision and Pattern Recognition_. 2023.

**Ethical Concerns:**

["NO or VERY MINOR ethics concerns only"]

**Final Justification:**

During the rebuttal, my main concerns have been addressed.
In particular, the authors addressed the clarity issues of the paper (in an answer to a different reviewer), discussed the assumptions and their implications in practice in more detail, and answered several smaller questions.

However, while I appreciate the theoretical analysis of reasoning shortcuts in CBMs, I am not sure how much this impacts the practical development of these models.

Nevertheless, I overall recommend the acceptance of the paper as it advances the theoretical understanding of shortcuts in reasoning settings.

**Limitations:**

yes

**Quality:**

3

**Strengths And Weaknesses:**

Strenghts:
- The paper covers an important problem of concept-based models: joint reasoning shortcuts, which are even more difficult to handle than regular (reasoning) shortcuts.
- It introduces a theoretical framework that allows for analysis of these JRS and to describe situations where they occur or can be avoided.

Weaknesses:
- While the theoretical framework is important to analyse these shortcuts from a formal perspective, the given assumptions are unlikely to hold in practice.
- The clarity of motivation and setup can be improved.

---

> ### Author Rebuttal · Authors · 2025-07-31
>
> We thank the reviewer for finding the addressed problem important and for appreciating our theoretical contribution. Below, we reply to the points of their review.
>
> > **Clarity of presentation**
>
> Thank you for the corrections and suggestions, we will implement them. We will be more explicit in Sections 1 and 2 about the problem we aim to solve, what we learn, and how we use supervision. We have also made use of the **extra page** to include a table of notation to help navigate the formalization. Below, we address different points separately. See also the reply to reviewer D2Sw.
>
> > **Validity of assumptions**
>
> Assumptions 4.1 and 4.2 are mild and in fact they hold in all the tasks available in the largest benchmark of NeSy reasoning shortcuts [Bortolotti et al., 2024], including the ones we experiment with. They hold because, for each input image $\mathbf x$, it is possible to recover the ground-truth concepts in the image $\mathbf g$ and the corresponding labels $\mathbf y$.
>
> We agree that it would be useful to lift these assumptions assumptions, but doing so is technically challenging. It is possible to relax Assumption 4.2 (i.e., labels cannot be entirely determined by the concepts), following [Marconato et al. 2023]; however, this can lead to a scenario where Theorem 4.9 no longer holds. Precisely, the absence of deterministic RSs does not inform us about whether non-deterministic RSs would appear or not [Marconato et al. 2023, Proposition 3].
>
> We view extending the current theoretical framework to the case where the assumption can be violated as an exciting research direction.
>
> > **Question: Intended semantics with fixed inference layer. Is it then implied that no permutations are allowed?**
>
> Yes, you are correct. When learned in an unsupervised fashion, concepts are “anonymous”: their meaning can only be figured out in a post-hoc manner.
>
> If the inference layer is fixed, e.g., in the MNIST-Addition task, _concepts are no longer anonymous_, and permutations of the concepts (as in Eq. (5, left)) result in reasoning shortcuts, as in [Marconato et al. 2023]. These permutations can yield wrong results on other tasks. E.g., permuting the two values of the MNIST digits, while giving correct predictions in MNIST-Addition, would give incorrect results in “MNIST-Subtraction”.
>
> Excluding the permutations when the inference layer is fixed is accounted for in the deterministic RSs count of Corollary 3.7, where only one solution (precisely the identity) has the intended semantics. We will make this more explicit in the text.
>
> > **Question: How CBNMs are trained and the use of language-guided supervision**
>
> If concept supervision is not available at all, then it is unclear how to train CBNMs “independently” (that is, concept extractor first, inference layer later).
>
> We train CBNMs with joint training [1] using only label supervision: both the concept extractor and the inference layer are trained to minimize the cross-entropy loss on the labels.
>
> We do not consider weak supervision obtained from VLMs like CLIP, as done by recent architectures. While exploring JRSs in this setup is interesting future work, we stress that this setup does not resolve issues with concept quality. Although VLMs keep improving, recent works [2,3] show that the concepts they produce are not necessarily high quality even for widely used datasets (like CUB), meaning that it is unclear whether VLM-based concept supervision can be safely integrated to avoid joint reasoning shortcuts.
>
> We will discuss this in the revised manuscript.
>
> > **Question: Evaluation of the inference layer**
>
> The procedure we follow is described in Appendix A5.  Intuitively, we assess whether the learned inference layer can handle the ground-truth concept annotations properly. For this to work, we first permute the g-t annotations to reflect the position of the concepts learned by the model, using the (inverse of the) permutation returned by Hungarian matching. We will clarify the intuition in the text.
>
> ----
>
> **References**
>
> [1] Koh et al. Concept Bottleneck Models. ICML 2020.
>
> [2] Srivastava, Yan, Weng.  VLG-CBM: Training Concept Bottleneck Models with Vision-Language Guidance. NeurIPS 2024.
>
> [3] Debole, Barbiero, Giannini, Passerini, Teso, Marconato. If Concept Bottlenecks are the Question, are Foundation Models the Answer? arXiv 2025.
>
> [4] Zheng, Xie, Zhang. Nonparametric Identification of Latent Concepts. ICML 2025.
>
> [5] Rajendran et al. From Causal to Concept-Based Representation Learning. NeurIPS 2024.

---

> > ### Comment · Reviewer_7RjS · 2025-08-04
> >
> > I thank the authors for their response.
> > I have some follow-up questions/points:
> >
> > - Assumptions: While the assumptions hold in the reasoning-shortcuts benchmark from Bortolotti et al., I think it is still important to note that these assumptions are more difficult to fulfill in real-world settings. However, I understand the point that working on relaxing these assumptions is an important direction for future work - this could be mentioned in the paper.
> >
> > - Independently trained CBNMs: I do not understand the response here - it would also be possible to train CBNMs independently when concept supervision is available? Can you please expand on this?

---

> > > ### Author Response · Authors · 2025-08-05
> > > **Reply to reviewer 7RjS**
> > >
> > > We thank the reviewer for the reply.
> > >
> > > > Assumptions
> > >
> > > We will definitely discuss relaxing these assumption in the revised manuscript.
> > >
> > > > Independent training of CBNMs
> > >
> > > Yes, if concept supervision is available, both joint training and independent training are feasible.
> > >
> > > We treat both CBMNs trained without concept supervision (experiments i Tables 1-4) and with different percentages of supervisions (the experiments in Figure 3 and in Table 4).
> > >
> > > From the experiments, we found that concept supervision is an effective mitigation strategy to learn correct concepts but it does not guarantee also learning the correct inference layer, see also Section 4 - supervised strategies.

---

> > > > ### Comment · Reviewer_7RjS · 2025-08-06
> > > >
> > > > I thank the authors for the answers and the final clarifications.
> > > > I have no further questions and will raise my score by 1.

---

> > > > > ### Author Response · Authors · 2025-08-06
> > > > > **Reply by authors**
> > > > >
> > > > > Thank you for the discussion.

---

### Official Review · Reviewer_D2Sw · 2025-06-30

**Clarity:** 2
**Significance:** 3
**Originality:** 3
**Rating:** 4
**Confidence:** 3

**Summary:**

The paper analyzes CBMs with respect to reasoning shortcuts: a CBM that learns a classifier of high accuracy without uncovering the true concepts or classifier function over the concepts. For this, the paper establishes a theory inspired by NeSy-CBMs. Using this theory the paper presents results for identifying both the true concepts and the true inference layer. In an empirical analysis, the paper demonstrated the impact of reasoning shortcuts on different datasets ans shows the impact of different mitigation strategies.

**Questions:**

1. L7: What are low-quality concepts? Isn't the definition of a concept quality always human-centric and, hence, notoriously difficult to describe mathematically?
2. The paper references causality papers. One know paper in this field is "Towards Causal Representation Learning" by Schölkopf et al. A major statement in this paper is the fact that statistical ML can only generalize in the iid and not in the ood regime. To generalize in the ood regime, it is required to identify causal connections between variables as each ood shift from the iid training data can be viewed as an intervention (in the causal sense). In this light, how conflicting are statements that CBMs can deal with ood (L20) even if they are identifying correlations (statistical machines)?
3. L46: What exactly is meant by discrete concepts?
4. How is Eq. (1) related to the usual factorization of CBMs $P(X,C,Y) = P(X)P(Y|C)P(C|X)$
5. L81: IMHO this assume that $\mathcal{G}$ is unique. How realistic is this assumption considering that in reality a set of different objects can be uniquely described by different sets of concepts (e.g., assume different level of abstractions)?
6. Table 2 results: All the methods in the table are deemed to be interpretable. Consequently, it should be possible to analyze the hypothesized simplicity bias. What rule has the SENN learned to solve the problem and how does this result relate to the established theory?
7. L341: "Our work partailly fills this gap" Why partially? Again, the **limitations** should be clearly stated.
8. L357: Why is contrastive learning helpful? Once more, using the interpretability of the methods should provide an answer.


**Some general notes not related to my assessment:**

- L24: What question? There is no question in the sentences before.
- Colored links and references are unusual and not really needed IMHO
- L32: shortcuts [...], more than reasoning shortcuts; it sounds like "more shortcuts than shortcuts"
- L44: a **proven** fact cannot fail **empirically**
- L99: A set is compared with an element. This doesn't make sense.
- L264: reporhhts
- L319: The "ood" generalization works because we induce expert knowledge pushing the assumed ood data into iid (see my comment above).
- Maybe it would be good to have one running example instead of MNIST and autonomous driving.

**Ethical Concerns:**

["NO or VERY MINOR ethics concerns only"]

**Final Justification:**

The authors addressed my comments in the rebuttal and promised to fix the mentioned issues (my mentioned points highly overlap with comments from other reviewers). Since the corrections and improvements span huge parts of the paper, I only raise my score by one point (from 3 to 4) since I cannot foresee if all the promised changes will turn the current submission in a strong one that would justify a rating of 5.

Thanks again to the authors for the good rebuttal!

**Limitations:**

The discussions of limitations must be improved: How realistic are the assumptions and what are the consequences? See further details in my comments before.

**Paper Formatting Concerns:**

Minor point: not sure if colored section links etc. are okay

**Quality:**

2

**Strengths And Weaknesses:**

**Note:** I have not checked the correctness of the proofs in detail due to the overall length of the paper+appendix.

### Strengths:

- **Originality:** The idea to analyze the learning abilities of CBMs from a mathematical perspective wrt whether the true generative process can be recovered is great. The results provide new insights about how difficult it is in theory to identify this said process. Moreover, the paper clearly motivates the existing work with respect to state of the art, and shows the limitations in learning CBMs even if respective penalties are applied.

- **Significance:** The presented theory and results are of interested for the community that works on the design of interpretable models, especially CBM researchers. Additionally, the paper addresses a difficult task, IMHO.

### Weaknesses:

- **Clarity:** As already said, the paper addresses a difficult task and, in such a situation, it is to some extent okay if the paper is not easy to understand and follow. However, the understandability of the paper should be improved. First of all, a lot of information relevant to key claims (e.g., the transferability of the results to a real task is in Appendix B) are in the appendix making the paper not self-contained. Second, the mathematical part is not easy to understand because of too short descriptions and because of using mathematical notations as isolated descriptors instead of accompanying symbols for clarity of a description (see examples below). My impression is that the paper clarity and also **quality** suffered from the page limit of the venue. I think the paper would be better suited in a journal where the page limit is not so strict. In this case, I think that the paper could become a really strong submission by having the paper better organized and content less scattered across main part and appendix. Regarding the usage of mathematical notations, ideally, mathematical notation should support the written text to add clarity but it shouldn't be a replacement for text. For instance:
   - L117: "the maps $\alpha$ and $\beta$" better "the ground-truth distribution $\alpha$ that generates the data ..."; I'm not claiming that this is a good description but I hope it makes my statement clearer.
   - L83: "$\mathcal{G} = \mathcal{C}$" better "where we assume equal discrete $\mathcal{C}$ and ground-truth concept sets $\mathcal{G}$, i.e., $\mathcal{G} = \mathcal{C}$" [...just realizing that it is not easy for me to find clear descriptive phrases for the mathematical entities, which is not a good indicator for the paper's clarity].

   The reason why I'm stressing this point is because it complicates clarity as one has to memorize all the notations in order to understand the paragraphs. Moreover, in some cases it leaves room for different interpretations if a mathematical notation is not clear (e.g., L116, $\mathbb{1}${$\mathbf{C} = \mathbf{c}$}, isn't $\mathbf{C}$ a set and $\mathbf{c}$ an element of $\mathbf{C}$? How does comparison work?). Moreover, sometimes mathematical notations are not even clear at all (e.g., cf. L46 and L63: $\mathbf{C}$ vs $\mathbf{c}$). Additionally, Figure 1 is not self-explanatory. Even after reading the related text in the corresponding section, it requires some time to understand the illustration.

- **Quality:** The paper nicely presents the assumptions related to the theory. However, when it comes to Definition 3.3 that is the backbone of all the theory, it is not rigorous in explaining the importance and the connection to the disentanglement assumption. It should be clearly stated that the definition only makes sense under this assumption. Then it should be discussed how realistic this assumption is in reality (element-wise, invertible functions and bijection from ground-truth to learned concepts); relating to the **limitations** of the paper. This discussion is crucial as it explains whether the paper analyzes an aspect of practical relevance. Hence, this should be stated clearly. The same is true for Definition 3.4. It should be discussed how realistic this definition is and what happens if it doesn't hold.

---

> ### Author Rebuttal · Authors · 2025-07-31
>
> We thank the reviewer for the positive assessment of our contribution and for pointing out elements that need improvement. We will use the **extra page** allowed for the camera-ready to move the experiments on BDD-OIA in the main text, and to clarify the mathematical notation.
>
> > **Notation is difficult**
>
> We amend this by adding a table of notation in Appendix A and complementing the text with additional explanations of the quantities in use, wherever necessary (**See detailed comments below**). We hope this improves the readability and are eager to iterate further based on the reviewer’s feedback.
>
> > **L117 and L83**
>
> Thank you, we will improve the text based on your suggestions.
>
> > **The importance of disentanglement for intended semantics**
>
> We agree with the reviewer that disentanglement is a central notion, and our definitions of intended semantics and joint reasoning shortcuts are based on it.  We will complement the text around Def 3.3 specifying that Eq. (5, left) guarantees disentanglement of the learned concepts.
>
> > **Mismatch between theory and practice**
>
> Definitions 3.3 and 3.4 in our theory are intentionally strict.
>
> _Our key insight is that the existence of JRSs implies that the learned concepts and inference layers can drastically differ from the ground-truth ones, while still yielding accurate (in fact, perfect) label predictions_. This is precisely the issue we highlight in our experiments (Table 2 and Table 3), where models achieve high label accuracy despite significantly low F1(C) and F1(β) scores. These observations reflect a substantial mismatch between the learned and intended semantics, providing evidence that models are prone to learning JRSs.
>
> In theory, when CBM concepts are even slightly entangled or do not perfectly match the ground-truth concepts, they do not satisfy Definition 3.3. Similarly, models that achieve nearly optimal label predictions but lack the intended semantics do not qualify as JRSs under Definition 3.4. However, we emphasize that our practical conclusion does not hinge on these definitions being satisfied exactly. Rather, the theory helps explain and anticipate the failure modes we observe: high task performance can mask severe representational mismatches.
> Conversely, when deterministic JRSs are mitigated (as shown in Table 1), models exhibit much closer alignment with the intended semantics, both in terms of concept quality and reasoning, which validates our theoretical framing and its practical relevance.
> Questions:
>
> > **Question: L7: What are low-quality concepts?**
>
> We completely agree that concept quality is difficult to define and human-centric in general. As we mention in the introduction, at the bare minimum, “high quality” concepts ought to be interpretable and not compromise OOD generalization. This is precisely what our definitions of intended semantics and joint RSs are meant to capture.
>
> > **Question: How conflicting are statements that CBMs can deal with ood (L20) even if they are identifying correlations (statistical machines)?**
>
> Good point! CBNMs are learned from observational data, just like other neural networks. Fitting such data is insufficient to **guarantee** any sort of OOD generalization _when the target domain is different enough_ from the source domain. This applies equally well to the label predictions and to the concept predictions. If a CBM is affected by JRSs, however, we can construct target domains in which we know it won’t generalize. For instance, imagine it has learned concepts that do not identify the ground-truth ones (e.g., in MNIST-Addition, it conflates “0” and “2”). Then we can always design a ground-truth data-generating process whose concept distribution is the same as the source domain, but the inference layer is different (say, MNIST-Multiplication) such that the learned concepts will yield poor label predictions (all products involving “2” will yield “0” as output).
>
> In short, while the absence of JRSs gives no guarantees that a CBM will generalize OOD, the presence of JRSs substantially increases the chances that it will not.
>
> We will clarify what we mean by OOD in Section 3.
>
> > **Question: L46: What exactly is meant by discrete concepts?**
>
> We mean categorical concepts, that is, random variables that can take only discrete values, as opposed to continuous concepts like angles or distances. We will be more precise in the text.
>
> > **Question: How is Eq. (1) related to the usual factorization of CBMs?**
>
> It is identical, except we marginalize over the ground-truth concepts $g$.  This enables us to reason in terms of ground-truth concepts in the derivations and it is justified by our choice of data generating process.  The usual factorization of CBMs [1] can be reobtained considering that $ p^* ( \mathbf G \mid  \mathbf X ) p^* (\mathbf X )  =  p^* ( \mathbf X \mid  \mathbf G ) p^* (\mathbf G ) $. In which case, we have:
>
> $$
> p^{\ast}(\mathbf{X}, \mathbf{Y}) = \mathbb{E}_{\mathbf{g} \sim p^{\ast}(\mathbf{G} \mid \mathbf{X})} \left[ p^{\ast}(\mathbf{X}) \, p^{\ast}(\mathbf{Y} \mid \mathbf{g}; \mathrm{K}) \right]
> $$
>
> > **Question: L81: Ground-truth concept vocabulary is assumed to be unique**
>
> Great point. We agree that there might be different choices of concept vocabulary (or domain $\mathcal G$) that all work fine for any given task, e.g., at different levels of abstraction. We do not make any assumptions about how this vocabulary is chosen.
>
> To see this, fix any such vocabulary. This, in turn, it determines the ground-truth concept extractor $\mathbf{f}^\ast$ and inference function $\mathbf{\omega}^*$.  Compatibly, our Theorem 3.9 explains when the NeSy-CBM will learn concept extractor $\mathbf f$ and an inference layer $\mathbf \omega$ equivalent to these.
>
> **Our analysis holds for any proper choice of ground-truth vocabulary** (i.e., as long as it allows to predict the ground-truth label perfectly, and assumptions 4.1 and 4.2 hold). In practice, we expect that, with more fine-grained concepts joint reasoning shortcuts can increase, whereas they can decrease with coarser concepts.
>
> We will discuss this interesting point in the paper.
>
>
> > **Question: All the methods in Table 2 are deemed interpretable. Consequently, it should be possible to analyze the hypothesized simplicity bias. What rule has the SENN learned to solve the problem, and how does this result relate to the established theory?**
>
> SENNs provide local explanations for a given prediction, i.e., they do not provide an interpretable set of rules holding in the whole domain (i.e., the inference layer $\beta$ depends on the input). In contrast, NeSy-CBMs like DSL and DPL provide global explanations that we report in our experimental evaluation (see Appendix B).
>
> We computed the concept confusion matrices for SENNs. We will report them in Appendix B, together with a comparison with the other approaches. In short, we didn’t observe any simplicity bias in SENNs. We hypothesize that this depends on both the flexibility in modelling local rules and on the reconstruction penalty applied to concepts during learning.
>
> We will clarify this in the text.
>
> > **Question: L341: "Our work partially fills this gap." Why partially? Again, the limitations should be clearly stated.**
>
> We *partially* fill the gap because we focus on NeSy-CBMs. While our results are likely to transfer to other NeSy models, doing so properly warrants further work.
>
> > **Question: L357: Why is contrastive learning helpful? Once more, using the interpretability of the methods should provide an answer.**
>
> In our experiments (see Fig 3 for MNIST-SumParity and Fig 6 in the Appendix for CLEVR), contrastive learning reduces concept collapse but does not improve the other metrics. We remarked this point in the Conclusion (L327).
>
> This is likely because contrastive losses encourage the concept extractor to assign similar (resp. dissimilar) concepts to similar (resp. dissimilar) inputs, i.e., the concept extractor can not collapse concepts together without increasing the contrastive loss.  In practice, we would expect the same to happen also when a reconstruction term is in place, but this does not work as well (Fig 3) as it is more challenging to optimize.  There are also theoretical reasons to believe contrastive learning should help with identifiability [2].
>
> > **Some general notes not related to my assessment**
>
> We thank the reviewer for the detailed comments. We plan to include all the aforementioned changes in the final version of the manuscript.
>
> ----
>
> **References**:
>
> [1] Koh, P. W., Nguyen, T., Tang, Y. S., Mussmann, S., Pierson, E., Kim, B., & Liang, P. Concept bottleneck models. In ICML 2020.
>
> [2] Zimmermann, R. S., Sharma, Y., Schneider, S., Bethge, M., & Brendel, W.  Contrastive learning inverts the data generating process. In ICML 2021.

---

> > ### Comment · Reviewer_D2Sw · 2025-08-01
> > **Response to the rebuttal**
> >
> > Thank you for providing all the feedback to my questions. I will raise my overall score by one point.
> >
> > I really hope that the authors keep their promises to improve the clarity of the paper and to address the limitations rightfully. Please don't see the statement of limitations as a weakness. Stating limitations shows scientific rigor and is much appreciated. It makes a contribution stronger and not weaker!

---

> > > ### Author Response · Authors · 2025-08-05
> > > **Reply to reviewer D2Sw**
> > >
> > > We thank the reviewer for the reply.
> > >
> > > We will address all the points raised on reviews and we are eager to know if there is anything else we can do to steer the decision to full acceptance.

---

> > > > ### Comment · Reviewer_D2Sw · 2025-08-05
> > > >
> > > > The authors addressed my comments in the rebuttal and promised to fix the mentioned issues (my mentioned points highly overlap with comments from other reviewers). Since the corrections and improvements span huge parts of the paper, I only raise my score by one point (from 3 to 4) since I cannot foresee if all the promised changes will turn the current submission in a strong one that would justify a rating of 5.
> > > >
> > > > Thanks again to the authors for the good rebuttal!

---

> > > > > ### Author Response · Authors · 2025-08-06
> > > > > **Reply by authors**
> > > > >
> > > > > Thank you for the discussion.

---

### Official Review · Reviewer_mrPP · 2025-07-02

**Clarity:** 3
**Significance:** 3
**Originality:** 3
**Rating:** 4
**Confidence:** 4

**Summary:**

This paper investigates the issue of Joint Reasoning Shortcuts (JRSs) in CBMs and NeSy-CBMs. Through theoretical analysis and empirical validation, the authors reveal how JRSs lead models to learn low-quality concepts and inference layers, thereby compromising interpretability and OOD performance. The paper introduces the notion of "Intended Semantics" and derives theoretical conditions under which JRSs can be avoided. Experiments demonstrate the limitations of existing mitigation strategies, particularly in unsupervised settings.

**Questions:**

Regarding the issue of the number of deterministic JRSs raised in the weaknesses, I suggest further analyzing how different mitigation strategies might affect this quantity.

Concerning the necessity of studying CBM shortcuts, I have several reservations:
- Intermediate layers in standard vision neural networks typically lack interpretable semantic features, and there is already extensive research on neural network shortcuts [1]. What distinguishes this work?
- Enforcing intermediate layers to output interpretable concepts may be redundant (though it does enhance interpretability). In practical applications, constructing meaningful intermediate concepts is highly challenging. However, within the NeSy framework, the presence of intermediate concepts is justified due to the support of a knowledge base. So why not use current sota vision models?

[1] Shortcut Learning in Deep Neural Networks. Geirhos, R., Jacobsen, J.-H., Michaelis, C., Zemel, R., Brendel, W., Bethge, M., & Wichmann, F. A.

**Ethical Concerns:**

["NO or VERY MINOR ethics concerns only"]

**Final Justification:**

The author's response has solved my problem, and I will keep my score unchanged.

**Limitations:**

yes

**Paper Formatting Concerns:**

I haven't found any formatting concerns.

**Quality:**

3

**Strengths And Weaknesses:**

Strengths:
1. This paper is the first to establish a theoretical framework for JRSs, which extends RS, and proposes sufficient conditions to avoid JRSs (e.g., Theorem 3.9).
2. The theoretical analysis and experimental evaluations in this paper are both highly comprehensive.

Weaknesses:
There are some questions regarding the content of Theorem 3.9. If I understand correctly, the theorem states that when the number of deterministic JRSs is zero, the system possesses the intended semantics. While this result aligns with intuition, I believe it is more important to provide a theoretical explanation of the conditions/assumptions under which the number of deterministic JRSs can be reduced.

Additionally, most of the experiments in this paper are based on variants of MNIST, which is somewhat limited. Furthermore, it would be valuable to compare with other NeSy algorithms (I noticed that only DPL was mentioned in the paper), such as LTN, among others.

---

> ### Author Rebuttal · Authors · 2025-07-31
>
> We thank the reviewer for finding our theoretical framework novel and appreciating the depth of the analysis.
> Below, we address the points raised in their review.
>
> > **Conditions under which the count of deterministic JRSs can be reduced.**
> > **Question: How do different strategies affect this quantity?**
>
> Thank you for raising this important point. We agree that reducing the number of deterministic JRSs should be the primary goal and the starting point for designing new mitigation strategies. We stress that some of the existing strategies already allow for reducing the count in Thm 3.6; we analyze the impact of multi-task learning, concept supervision, knowledge distillation, and reconstruction explicitly in Appendix D. For example, introducing a reconstruction term avoids all those JRSs in which multiple concepts are collapsed together, leading to a smaller count.
> We will make sure to clarify this before the forward pointer to Appendix D in Section 4.
>
> > **Most experiments are based on variants of MNIST**
>
> In order to assess the impact of joint reasoning shortcuts, we need concept-level annotations and prior knowledge (e.g., the rules of traffic in BDD-OIA), which are not commonly available. Our experiments rely on the most extensive existing benchmark for reasoning shortcuts [1], and, besides variants of MNIST-Addition, they also evaluate CLEVR and BDD-OIA. The former two are easy to control and evaluate, the latter is a challenging autonomous driving task with real-world images, four binary labels, and 21 binary concepts.
>
> > **Comparison with other NeSy algorithms**
>
> We consider **two** NeSy baselines: DPL and DSL.
>
> LTN is not designed for learning the knowledge from data.  [Marconato et al. 2023] have already shown that NeSy predictors with fixed prior knowledge, including regular DPL, LTN, and the Semantic Loss [2], suffer from regular Reasoning Shortcuts.
>
> Our goal is to go beyond the “fixed prior knowledge” setting and understand what happens in  CBMs and NeSy predictors that learn the knowledge (or, more generally, the inference layer) from data. This is why our choice of competitors covers mainstream baselines like CBNMs, SENNs, and DSL, which all learn the inference layer. We also include a hybrid of DPL and DSL that does the same.
>
> It is not straightforward to adapt LTN to learn the knowledge, as it would require handling the fuzzy logic in the inference layer, and designing appropriate data structures and losses for doing so is non-trivial. Doing so would be a standalone contribution.
>
> > **Question: Distinction between vanilla shortcuts and joint reasoning shortcuts.**
>
> Regular shortcuts [Geirhos et al. 2020] can be understood as unintended input-to-label assignments that achieve high performance. E.g., the model might rely on confounding features (like watermarks) to output good predictions. They compromise generalization to domains where the confounders are not present.
>
> Joint reasoning shortcuts are failure modes where either or both the concepts and the inference layer are not equivalent (in the sense we specify in Def. 3.4) to the ground-truth concepts and inference layer, and they can compromise interpretability and generalization.
>
> The two issues are related but **different**. While shortcuts can induce JRSs, the converse is not true: a model can be affected by a JRS (i.e., it might have learned “bad” concepts or a “bad” inference layer) even if it does not rely on confounders. An example is reported in Fig 2. This also means that remedies to regular shortcuts do not necessarily resolve JRSs.
>
> We will clarify this distinction in the Related Work.
>
> > **Question: Why not using SotA vision models to extract concepts?**
>
> Good point. Our work focuses on “regular” CBMs and NeSy-CBMs that do not leverage VLM concepts, but it is true that recent CBM/NeSy architectures do use them [3,4]. However, although VLMs keep improving, recent works [5,6] show that the concepts they produce are not necessarily high quality even for widely used datasets (like CUB), meaning that VLMs may be affected by JRSs or similar issues too. We plan to extend our theory to this case in future work. We will include this note in the revised manuscript.
>
> -----
>
> **References**
>
> [1] Bortolotti, Marconato, Carraro, Morettin, van Krieken, Vergari, Teso, and Passerini. A neuro-symbolic benchmark suite for concept quality and reasoning shortcuts. NeurIPS 2024.
>
> [2] Xu, Zhang, Friedman, Liang, van den Broeck. A semantic loss function for deep learning with symbolic knowledge. ICML 2018.
>
> [3] Cunnington, D., Law, M., Lobo, J., & Russo, A. The role of foundation models in neuro-symbolic learning and reasoning. In International Conference on Neural-Symbolic Learning and Reasoning 2024.
>
> [4] Oikarinen, T., Das, S., Nguyen, L., & Weng, L. Label-free Concept Bottleneck Models. ICLR 2023.
>
> [5] Srivastava, Yan, Weng.  VLG-CBM: Training Concept Bottleneck Models with Vision-Language Guidance. NeurIPS 2024.
>
> [6] Debole, Barbiero, Giannini, Passerini, Teso, Marconato. If Concept Bottlenecks are the Question, are Foundation Models the Answer? arXiv 2025.

---

### Official Review · Reviewer_d94G · 2025-07-03

**Clarity:** 2
**Significance:** 2
**Originality:** 2
**Rating:** 4
**Confidence:** 4

**Summary:**

The paper attempts to connect reasoning shortcuts from neurosymbolic AI  with concept-based models, where models can achieve high performance without learning the right semantics. It shows that maximum likelihood training alone is insufficient to ensure intended semantics and formalize the conditions for when a CBM can acquire high-quality concepts and inference layer. It also explores and empirically tests potential mitigation strategies. The results show promise on benchmark datasets.

**Questions:**

* The paper seems to give a concept-based lens to a neurosymbolic models, rather than the other way around. It claims that a CBM is a NeSyCBM with a learnable inference layer. Is my understanding correct? If so, I don’t fully agree with this, as CBMs and NeSy models are fundamentally meant for different tasks (classification and reasoning respectively). Could you please exactly state what sort of CBM is being referred to throughout?
* Is there a difference between NeSy models as NeSyCBMs? If so, could you please give an example of a NeSy model that isn’t a NeSyCBM?
* Could you explain the reasoning behind the choice of baselines? There are only two CBM-based baselines [3, 4]. All the others are NeSy ones. Also, if the CBNMs [3] have no concept supervision (L267), how are the concepts learned?
* The paper brings up desiderata for high-quality concepts and inference layer earlier in the paper. Yet these aren’t evaluated in any way or even used much in the derivations, for that matter. Could you explain why they were introduced?

**Ethical Concerns:**

["NO or VERY MINOR ethics concerns only"]

**Final Justification:**

I thank the authors for the responses; the rebuttal was helpful in clarifying most of my questions. I'd suggest that the authors also revise the manuscript with these responses to make it clearer (lack of clarity in the current manuscript on some of these fronts was an earlier concern for me).

I stay positive on the paper for the contributions; however, I still believe the empirical analysis is limited, and stay with my BA rating.

**Limitations:**

Yes

**Quality:**

3

**Strengths And Weaknesses:**

Strengths:
+ Studying concept and inference layer quality in CBMs from a reasoning shortcuts point of view is novel, to the best of my knowledge.
+ The presented theory and bounds seemed fairly complete (although I do have some concerns about them in the weaknesses) and the provided examples made things more intuitive.

Weaknesses:
- The formalization of reasoning shortcuts for CBMs is interesting, however, considering the fact that a lot of the results build on existing works like [1, 2] (which the paper also mentions in the related work), makes the degree of novelty of the work questionable. Additionally, most of the practical solutions (Sec 4), have also already been proposed in [1].
- I found the paper slightly lacking in clarity. In particular, in Sec 3, where CBMs are modeled as a pair of functions, the concept extractor is defined as mapping onto a simplex, yet in words it is said that it maps to a conditional distribution. Putting this mismatch aside, in standard CBMs [3], the concepts are predicted independently. There is no distribution over them. So it is not clear how vanilla CBM (as widely used) fits into this approach.
- I also found it difficult to differentiate between NeSyCBMs and general NeSy models - although NeSyCBMs were defined, they weren’t formalized mathematically. Several of the references and baselines were standard NeSy models, like DeepProbLog.

References:
[1] Marconato, Emanuele, et al. "Not all neuro-symbolic concepts are created equal: Analysis and mitigation of reasoning shortcuts." Advances in Neural Information Processing Systems 36. 2023.
[2] Yang, Xiao-Wen, et al. "Analysis for abductive learning and neural-symbolic reasoning shortcuts." Forty-first International Conference on Machine Learning. 2024.
[3] Koh, Pang Wei, et al. "Concept bottleneck models." International conference on machine learning. PMLR, 2020.
[4] Alvarez Melis, David, and Tommi Jaakkola. "Towards robust interpretability with self-explaining neural networks." Advances in neural information processing systems 31 (2018).

---

> ### Author Rebuttal · Authors · 2025-07-31
>
> We thank the reviewer for appreciating the novelty of our work and the thoroughness of the analysis.
>
>
> > **Novelty of the work**
>
> We acknowledge that the work of Marconato et al. (2023) inspires ours, but we go beyond it by proposing a common framework to analyze NeSy-CBMs and CBNMs. We differ in that:
>
> - **Scope** Marconato et al. study NeSy-CBMs where the inference layer is **fixed**. Lifting this constraint is technically challenging but it allows us to substantially generalize the notions of reasoning shortcuts and concept identifiability to CBNMs, a class of models that were never analyzed under this lens.
> - **Analysis** How to properly learn concepts without concept supervision is an open question. Our results answer when to expect concepts to be learned correctly by avoiding joint reasoning shortcuts.
> - **Mitigation** We test existing mitigation strategies for NeSy-CBMs with a fixed inference layer and consider two novel strategies: knowledge distillation and contrastive learning. Our results suggest that preventing JRSs is still an open question.
>
> > **Clarity**
>
> We use the functional notation for ease of manipulation (see 100-104). This formalism also captures regular CBNMs. We consider CBNMs where concepts are modelled by applying a sigmoid (or a softmax) activation on input embeddings in the bottleneck, meaning they are conditionally independent given the input, as usual. In this setting, the CBNMs’ concept extractor is precisely a function that maps inputs to a simplex with one dimension per concept: the $j$-th coordinate is the conditional probability that the $j$-th concept appears in the input. The same setup can also express NeSy-CBMs like DPL and LTN, which consider the same activations on the bottleneck.
>
>
> > **Question: NeSy is meant for reasoning, CBMs for classification**
>
> NeSy encompasses a wide family of approaches that integrate learning and reasoning, but reasoning shortcuts have so far only been studied in “NeSy predictors”, which happen to share much of the structure of CBNMs: both include a concept extractor mapping inputs to concepts and an inference layer, and both are trained via gradient descent. The main difference is the inference layer: in NeSy predictors, this is typically fixed and encodes known rules for deriving the label from the concepts (e.g., the rule of addition in MNIST-Addition); in CBNMs, it is typically a learned linear layer. This structural similarity is what enables us to extend RSs from NeSy predictors to CBNMs. At any rate, in NeSy, reasoning is always a means of computing predictions, as done in MNIST-Addition [4].
>
> > **Question: NeSy-CBMs vs general NeSy models**
>
> We distinguish between NeSy-CBMs, which combine a concept extractor and an inference layer through a bottleneck of concept activations, and general NeSy models, which can have very different architectures and, in particular, may not model a bottleneck of latent concepts. Examples include grounding-specific Markov logic networks [1], SATNet [2], and the neural theorem prover [3].
>
>
> > **Question: Why these baselines? There are only two CBMs, all others are NeSy**
>
> For both CBMs and NeSy-CBMs, we evaluated the most representative architectures:
>
> * CBNMs are the most well-studied architecture, and they also serve as a recipe for other concept-based models [see references in lines 53, 54, and 55];
> * SENNs are also well-known but leverage a slightly different architecture designed to learn concepts in an unsupervised manner.
> * DSL learns both knowledge and concepts, and implements the inference step as a lookup table.
> * a modified version of DeepProbLog that is similar to DSL but, like regular DeepProbLog, uses probabilistic logic for inference.
>
>
> > **Question: If the CBNMs have no concept supervision, how are the concepts learned?**
>
> We train CBNMs using a regular cross-entropy loss on the labels, just like regular feed-forward neural networks. The concepts are treated as latent variables: the model will learn “concepts” that allow it to achieve low training loss. CBNMs also include a cross-entropy loss on the concept bottleneck, which we don't use unless otherwise specified.
>
>
> > **Question: Why introducing the desiderata if you don’t evaluate them?**
>
> Thank you for raising this point. We want to clarify that **the metrics we use to evaluate the accuracy of learned concepts (F1(C)) and of the learned inference layer (F1(beta)) take both desiderata into account**.
>
> - **Disentanglement**:  F1(C) evaluates how close the map $\alpha$ (encoding the concept extractor) is to the identity, up to permutation and element-wise invertible transformations. To account for these symmetries, we apply the Hungarian matching algorithm to map learned to ground-truth concepts. This indirectly measures whether the learned concepts are disentangled.
> - **Generalization**: We use the permutation and element-wise transformation obtained via Hungarian matching to evaluate whether the learned inference layer $\omega$ predicts the same labels as the ground-truth knowledge $\omega^*$.  In summary, F1(\alpha) and F1(\beta) together tell us whether the two desiderata are respected.
>
> We will clarify in the text how the experimental verification accounts for the two desiderata.
>
> ----
>
> **References**
>
> [1] Lippi and Frasconi. "Prediction of protein β-residue contacts by Markov logic networks with grounding-specific weights." Bioinformatics, 2009.
>
> [2] Wang, Donti, Wilder, and Kolter. Satnet: Bridging deep learning and logical reasoning using a differentiable satisfiability solver. ICML 2019.
>
> [3] Rocktäschel and Riedel. End-to-end differentiable proving. NeurIPS, 2017
>
> [4] Manhaeve, R., Dumancic, S., Kimmig, A., Demeester, T., & De Raedt, L.. Deepproblog: Neural probabilistic logic programming. NeurIPS 2018

---

> > ### Author Response · Authors · 2025-08-06
> > **Comment by authors**
> >
> > Dear reviewer, \
> > Since the discussion period is coming soon to an end, we would like to know if our rebuttal addresses the points raised by the reviewer. Thank you.

---

> > ### Comment · Reviewer_d94G · 2025-08-07
> > **Response to author rebuttal**
> >
> > I thank the authors for the responses; the rebuttal was helpful in clarifying most of my questions. I'd suggest that the authors also revise the manuscript with these responses to make it clearer (lack of clarity in the current manuscript on some of these fronts was an earlier concern for me).
> >
> > I'd still think it'd have been ideal to compare the method against other CBM variants (there have been a variety of them in recent years including Sparse CBMs, Posthoc CBMs, Label-free CBMs, etc) -- many of these variants outperform CBMs. Even 1 or 2 other empirical comparisons would have made this more convincing empirically, as well as in terms of its generalizability too. Having said that, I agree that the work advances the understanding and adoption of reasoning shortcuts in concept-based learning, and lean positively on the work overall.

---

> > > ### Author Response · Authors · 2025-08-07
> > >
> > > We appreciate the reviewer’s thoughtful feedback and positive assessment of our work.
> > >
> > >  Regarding the suggestion to include more CBM variants (e.g., Sparse CBMs, Posthoc CBMs, Label-free CBMs), we agree that these are valuable directions. However, we believe that the core insights of our analysis--particularly those concerning joint reasoning shortcuts (JRSs)--are unlikely to change significantly across these variants, as they share the same fundamental bottleneck architecture and training dynamics as the models we study.
> > >
> > > These variants typically modify the concept supervision mechanism (e.g., Label-free CBMs) or impose architectural constraints (e.g., sparsity), but it is unclear how such changes would reduce the risk of learning JRSs. On one hand, sparsity may not always promote learning interpretable concepts [1]. On the other, a qualitatively different behavior may be observed by models that leverage **language-guided supervision**. However, the quality of the learned concepts in these models often depends heavily on the quality of the underlying vision-language model annotations, which can be noisy or unreliable in certain domains [2]. Studying their effect on JRSs is indeed a promising direction, but doing so would require a separate line of investigation and careful experimental design. We leave this to future work.
> > >
> > > We will clarify this in the revised manuscript.
> > >
> > > [1] Kantamneni et al. Are sparse autoencoders useful? A case study in sparse probing, ICML 2025 .\
> > > [2] Debole et al. If concept bottlenecks are the question, are foundation models the answer? arXiv 2025.

---

> > > > ### Comment · Reviewer_d94G · 2025-08-08
> > > >
> > > > Thank you for your clarifications.

---

### Decision · Program_Chairs · 2025-09-17

**Decision:**

Accept (poster)

**Comment:**

The paper reveal how JRSs lead models to learn low-quality concepts and inference layers, thereby compromising interpretability and OOD performance through theoretical analysis and empirical validation. It introduces the notion of "Intended Semantics" and derives theoretical conditions under which JRSs can be avoided.
All reviewers acknowledged the innovative combination of CBMs and reasoning shortcuts presented in this paper, as well as its solid theoretical contributions. However, some concerns remain regarding the sufficiency of the experimental validation and the clarity of certain theoretical sections. Overall, I recommend accepting this paper, but I strongly encourage the authors to include additional experimental results and to further improve the exposition of the theoretical parts in the camera-ready version